# Incentivizing Honesty among Competitors in Collaborative Learning and Optimization

**Florian E. Dorner**
MPI for Intelligent Systems, Tübingen
ETH Zurich
florian.dorner@tuebingen.mpg.de

**Nikola Konstantinov**
INSAIT, Sofia University
nikola.konstantinov@insait.ai

**Georgi Pashaliev**
Sofia High School of Mathematics

**Martin Vechev**
ETH Zurich
martin.vechev@inf.ethz.ch

## Abstract

Collaborative learning techniques have the potential to enable training machine learning models that are superior to models trained on a single entity's data. However, in many cases, potential participants in such collaborative schemes are competitors on a downstream task, such as firms that each aim to attract customers by providing the best recommendations. This can incentivize dishonest updates that damage other participants' models, potentially undermining the benefits of collaboration. In this work, we formulate a game that models such interactions and study two learning tasks within this framework: single-round mean estimation and multi-round SGD on strongly-convex objectives. For a natural class of player actions, we show that rational clients are incentivized to strongly manipulate their updates, preventing learning. We then propose mechanisms that incentivize honest communication and ensure learning quality comparable to full cooperation. Lastly, we empirically demonstrate the effectiveness of our incentive scheme on a standard non-convex federated learning benchmark. Our work shows that explicitly modeling the incentives and actions of dishonest clients, rather than assuming them malicious, can enable strong robustness guarantees for collaborative learning.

## 1 Introduction

Recent years have seen an increased interest in designing methods for collaborative learning, where multiple participants contribute data and train a model jointly. The premise is that the participants will then be able to obtain a better model than if they were learning in isolation. Most prominently, *federated learning (FL)* (Kairouz et al., 2021) provides a method for training models in a distributed manner, allowing data to stay with institutions, while still harvesting the benefits of collaboration.

An underlying premise for the success of collaborative learning schemes is that the participants contribute data (or gradient updates) relevant to the learning task at hand. However, when participants are in competition on some downstream task, they may have an incentive to sabotage other participants' models. For instance, firms that are competing on the same market can often improve their machine learning models by having access to their competitors's data, but at the same time will likely benefit from a gap between the quality of their models and those of other firms.

These conflicting incentives raise a concern that collaborative learning may be vulnerable to strategic updates from participants. Previous work has empirically demonstrated that irrelevant or malicious updates can negatively impact collaborative learning (Tolpegin et al., 2020; Kairouz et al., 2021). In

37th Conference on Neural Information Processing Systems (NeurIPS 2023).

particular, if a subset of participants is modeled as fully-malicious (Byzantine) agents, that collude in a worst-case manner, it is known that optimal convergence rates contain a leading-order term that is based on the fraction of Byzantine agents and is irreducible as the number of players increases (Yin et al., 2018; Alistarh et al., 2018). This suggests that collaborative learning in the presence of strategic behavior may often not provide asymptotic benefits over learning with one's own data.

**Contributions**   In this work, we study collaborative learning in the presence of strategic behavior by explicitly modeling players' competitive incentives. We consider a game between multiple players that exchange updates via a central server, where players' rewards increase both when they obtain a good model for themselves, and when other players' models perform poorly.

We study two important instantiations of this collaborative learning game: mean estimation with a very general action space and strongly-convex stochastic optimization with attacks that add gradient noise. We show that players are often incentivized to strongly manipulate their estimates, rendering collaborative learning useless. To remedy this, we suggest mechanisms inspired by peer prediction (Miller et al., 2005), that penalize cheating players using side payments. Our results on stochastic optimization rely on a novel recursive bound for the squared norm of differences in SGD-iterates between a clean trajectory and a strategically corrupted trajectory. Meanwhile, we show that side payments can be avoided in the mean estimation case, using a novel communication protocol in which the server sends noisy estimates back to players that are suspected of cheating. Our mechanisms are solely based on observable player behaviour, and recover near-optimal convergence rates at equilibrium. Furthermore, expected payments cancel out when all players are honest, so that players are incentivized to participate in the training, rather than use their own data only, despite the penalties.

Finally, we conduct experiments on the FeMNIST and Twitter datasets from LEAF (Caldas et al., 2018) and demonstrate that our mechanisms can incentivize honesty for realistic non-convex problems

## 2   Related work

**Game theory and collaborative learning**   Many works that study connections between FL and game theory focus on clients' incentives to participate in the training process at all (see Tu et al. (2021) for a recent survey, and Gradwohl & Tennenholtz (2022) for an analysis of how this relates to competition). Similarly, Karimireddy et al. (2022) study incentives for free-riding, i.e. joining collaborative learning without spending resources to contribute data. In contrast, our setting covers clients that strategically manipulate their updates in order to damage other participants' models.

Another line of work studies FL as a coalitional game theory problem, in which the players need to deal with potential issues of between-client heterogeneity by deciding with whom to collaborate Donahue & Kleinberg (2021a,b). Optimal behavior under data heterogeneity is also studied by Chayti et al. (2021), while Gupta et al. (2022) studies invariant risk minimization games in which FL participants learn predictors invariant across client distributions. In contrast, we study FL as a non-cooperative game and seek to address strategic manipulation, rather than data heterogeneity.

**Robustness in federated learning**   The robustness of FL to noisy or corrupted updates from clients has received substantial recent interest, see Shejwalkar et al. (2022) for a recent overview. One line of work studies federated learning in the context of various data corruption models, e.g. noise or bias towards protected groups (Fang & Ye, 2022; Abay et al., 2020). In contrast, we study strategic manipulation by clients as the source of data corruption.

Clients attacking the training are typically modeled as Byzantine (Blanchard et al., 2017; Yin et al., 2018; Alistarh et al., 2018), adversarially seeking to sabotage training by deviating from the FL protocol in a worst-case manner. The goal of Byzantine-robust learning is to achieve guarantees in that setting. Similar models have been studied in a statistical context, where data is stored at a single location and so communication is not a concern (Qiao & Valiant, 2018; Konstantinov et al., 2020). In contrast to these works, we model manipulation as a consequence of competitive incentives rather than maliciousness and analyze client behaviour using game theory (Osborne & Rubinstein, 1994). Rather than focussing on robustness to manipulations, we aim to *prevent manipulation* alltogether.

**Peer prediction mechanisms**   Our mechanisms for inducing honesty are closely related to *peer prediction* that aims to incentivize honest ratings on online platforms. In their seminal paper,

Miller et al. (2005) suggest paying raters based on how much their rating helps to predict other raters' ratings. While they require a common prior shared by all raters and the mechanism designer, Witkowski & Parkes (2012) extend their results relaxing that assumption. Closely related to our work, Cai et al. (2015) suggest to incentivize crowdworkers to produce accurate labels by paying them more, the better their label gets predicted by a model estimate from other workers' data. Meanwhile, Waggoner & Chen (2014) prove that peer prediction elicits common knowledge, rather than truth from participants. Lastly, while Karger et al. (2021) find that peer prediction can elicit subjective forecasts of similar accuracy as scoring based on the ground truth, Gao et al. (2014) demonstrate that human raters can end up with dishonest strategies despite the existence of honest equilibria.

# 3 Competitive federated learning

In this section, we present our framework on a high level and explain how it models competitive behaviour in FL. In this generality, however, it is impossible to analyze the problem quantitatively. We therefore define specific instantiations of the framework, which we study for the rest of the paper.

## 3.1 General framework

**Overview** Throughout the paper, we assume that there are $N \geq 2$ players, who each have a private dataset. The players participate in an FL-like procedure, which takes place over multiple rounds and requires them to send messages with information relevant to update a centrally trained model at every step. In our setup, players act strategically and competitive pressures might incentivize them to try to corrupt other players' models. This is done by manipulating updates sent to the central model, while simultaneously keeping track of an unmanipulated private (presumably more accurate) model.

To model the participants' strategic interactions, we need to define a game by specifying their action spaces and rewards. To this end, one needs to specify: their *attack strategies*, that describe whether and how they will corrupt their messages to the server; their *defense strategies*, which describes how they postprocess the server's updates to defend themselves against others' manipulations; and their *rewards*, in a way that reflects the quality of learning and the competition between them. Given these components, we are interested in the Nash equilibria of the corresponding game, in order to understand strategic behavior in FL and how it affects the quality of the players' models.

**Formal setup** We denote the samples of each player $i$ by $x^i = \{x_1^i, \ldots, x_n^i\}$, where $x \in \mathcal{X}$, and assume that $\{x_j^i : i \in [N], j \in [n]\}$ are *not necessarily independent* samples from an unknown distribution $\mathcal{D} \in \mathcal{P}(\mathcal{X})$. For simplicity, we assume that all players have an equal number of samples $n$. Players communicate via an FL-like protocol, through which a central server model $\theta^s \in \mathbb{R}^d$ is updated. The intended goal of this procedure is to find a value for $\theta^s$ that minimizes a loss function $f_{\mathcal{D}}(\theta)$. Note that because $\mathcal{D}$ is unknown to the players, they can benefit from honest collaboration.

The protocol consists of $T$ rounds and the central model is initialized at some $\theta_1^s \in \mathbb{R}^d$. At time $t = 1, 2, \ldots, T$ the server sends the model $\theta_t^s$ to all participants. Each agent $i$ is then *meant to send an update* $g_t(\theta_t^s, x^i)$ to the server, for some function $g_t : \mathbb{R}^d \times \mathcal{X}^n \to \mathbb{R}^d$. For example, in the standard FedSGD setting, $f_{\mathcal{D}}(\theta) = \mathbb{E}_{x \in \mathcal{D}}[f_x(\theta)]$ for functions $f_x : \mathbb{R}^d \to \mathbb{R}$ and $g = \frac{1}{n} \sum_j \nabla f_{x_j^i}(\theta_t^s)$ serves as an estimate of the gradient of $f$ evaluated at the data of player $i$. Each player $i$ then sends a message $m_t^i \in \mathbb{R}^d$ to the server (which may or may not be equal to $g_t(\theta_t^s, x^i)$). Finally, the server computes a new model $\theta_{t+1}^s = \text{Agg}(\theta_t^i, m_t^1, m_t^2, \ldots, m_t^N)$, via an aggregating function $\text{Agg} : \mathbb{R}^d \times \mathbb{R}^{N \times d} \to \mathbb{R}^d$ (for example, in FedSGD this will be a gradient update computed as $\bar{m}_t = \frac{1}{N} \sum_{i=1}^N m_t^i$).

**Players' strategies** At every step, the players take two decisions: how to *attack* by sending a manipulated estimate to the server, and how to *defend* themselves from unreliable estimates when updating their locally tracked model based on information received from the server. Formally, we assume that each player $i \in [N]$ chooses (potentially randomized) functions $a_1^i, a_2^i, \ldots, a_T^i$ and $d_1^i, \ldots, d_T^i$, that describe their behavior for the attack and defense stages at every time step. These functions are chosen from two respective *sets of possible actions* $\mathscr{A}$ and $\mathscr{D}$. The tuple of chosen actions $p^i = (a_1^i, \ldots, a_T^i, d_1^i, \ldots, d_T^i) \in \mathscr{A}^T \times \mathscr{D}^T$ represents the player's global strategy.

In the most general case, the attacks and defenses of each player $i$ may take into account all information available to the player, throughout the history of the optimization process. At time $t$ this

include the models $\theta_1^s, \ldots, \theta_t^s$ received by the server; the local models $\theta_1^i, \ldots, \theta_t^i$ the player kept at previous iterations; the attack strategies $a_1^i, \ldots, a_t^i$ used up to time $t$ (e.g. to correct for one's own faulty estimate at time $t$); as well as additional randomness $\xi_1^i, \ldots, \xi_t^i$ sampled at each round.

**Players' rewards**  Each player aims to obtain a final model $\theta_{T+1}^i$ that approximately minimizes $f_{\mathcal{D}}(\theta_{T+1}^i)$. Crucially, their reward also depends on other players' models. Specifically, we assume that each player $i$ has a reward function $\mathcal{R}^i : \mathbb{R}^{N \times d} \times \mathcal{P}(\mathcal{X}) \to \mathbb{R}$ and receives the reward

$$r^i = \mathcal{R}^i\left(\theta_{T+1}^1, \theta_{T+1}^2, \ldots, \theta_{T+1}^N, \mathcal{D}\right). \tag{1}$$

Note that the messages $m^i$ sent by players and thus $\theta_{T+1}^i$ and each player's reward depend not only on players' strategies, but also on the particular realization of their samples $x^i$. We thus focus on *expected rewards*, averaging out the effects of particular realizations of players' samples and randomness in their strategies. We study the *vector of expected rewards* $\mathbb{E}(r^1, \ldots, r^N)$ and its dependence on the *strategy profile*, that is on the distribution of strategies $p = (p^1, p^2, \ldots, p^N)$ chosen by each player.

**Assumptions on players' behaviour**  To analyze players' behaviour, we make two assumptions, giving rise to a classic game-theoretic setup. The first is that players seek to maximize their expected reward, as defined above, i.e. players are *rational*. In addition, since the reward of each player depends on the actions of the other players, players account for the actions of the others, which means that their behavior is *strategic*. A natural solution concept in this context is the Nash equilibrium (Nash, 1951). This describes a strategy profile in which no player can improve their reward by unilaterally changing their strategy. In our case, this classic notion translates to the following definition.

**Definition 1.**  Let $p = (p^1, p^2, \ldots, p^N) \in \left(\mathcal{P}(\mathscr{A}^T \times \mathscr{D}^T)\right)^N$ be a strategy profile. Then $p$ is a (mixed) Nash equilibrium if:

$$\forall p^* \in \mathcal{P}(\mathscr{A}^T \times \mathscr{D}^T) \text{ and } \forall i \in [N] : \mathbb{E}\left(r^i(p^1, \ldots, p^i, \ldots, p^N)\right) \geq \mathbb{E}\left(r^i(p^1, \ldots, p^*, \ldots, p^N)\right),$$

where the expectation is taken with respect to the randomness of the data and the players' strategies.

### 3.2  Specific instantiations: mean estimation and stochastic gradient descent

We study two specific cases of the game, each modeling a fundamental learning problem. The first is single-round mean estimation (Sections 4 and 5), which correspond to the general setup with $T = 1$, $f_{\mathcal{D}}(\theta) = \|\theta - \mu\|^2$, where $\mu = \mathbb{E}_{X \sim \mathcal{D}}(X)$, and the updates $g$ being the sample means of the players. The second is multi-round stochastic optimization of strongly-convex functions $f_{\mathcal{D}}(\theta)$ via SGD (Section 6), in which case the updates $g$ are stochastic gradient estimates based on players' data. In the corresponding sections, we define natural rewards that model the competing incentives.

**Strategy spaces**  To describe the *attack strategies*, in both cases we model *attacks that send a noisy update* $g_t(\theta_t^s, x^i) + \alpha_t^i \xi_t^i$, for normalized zero-mean noise $\xi_t^i$ and an attack parameter $\alpha_t^i \in \mathbb{R}$, to the server. Up to a certain magnitude of $\alpha_t^i$, these attacks have a natural interpretation as the act of *hiding a random subset of a player's data*. In addition, the $\alpha_t^i$ parameters have a natural interpretation as the *of aggressiveness of the player*. In the case of mean estimation, we are also able to analyze much more general attack strategies that can adjust $\alpha_t^i$ based on the players' samples $x^i$ and additionally allow for adding a directed bias to the communicated messages.

For the *defense strategies* in the mean estimation case, we consider a defense strategy that corrects the mean estimate received from the server for the player's own manipulation. The player then computes a weighted average of the result and the their local mean. The weighting parameter $\beta^i$ used then has a natural interpretation as the *cautiousness of the player*. In the SGD case we directly provide mechanisms that incentivize honesty at the attack stage, making potential defenses redundant.

## 4  A single-round version of the game: competitive mean estimation

In this section we analyze a single-round version of the game, in which players aim to estimate the mean of a random variable $X \in \mathcal{P}(\mathbb{R}^d)$. Specifically, we consider the game defined in Section 3, in the case of $T = 1$ rounds. Players sample from a distribution $\mathcal{D} \in \mathcal{P}(\mathbb{R}^d)$ by first independently sampling a random mean $\mu_i \sim D_\mu$ and "variance" $\sigma_i^2 := \mathbb{E}\|X^i - \mu_i\|^2 \sim D_\sigma$ (this models potential

*heterogeneity* between clients), and then receiving (conditionally) independent samples from a random variable $X^i$ with mean $\mu_i$ and "variance" $\sigma_i^2$. We call $\mu = \mathbb{E}\mu_i$, $\sigma^2 = \mathbb{E}\sigma_i^2$ and $\sigma_\star^2 = \mathbb{E}\|\mu - \mu_i\|^2$. We assume that players do not know the distributions $D_\mu$ and $D_\sigma$.

Each player wants to estimate the global mean $\mu$ as well as possible. During the single communication round, the players are meant to communicate the mean of their samples: $g_1(\theta_1^s, x^i) = \frac{1}{n}\sum_{j=1}^n x_j^i$. Instead, they send messages $m_1^i$. The server aggregates the received messages by averaging them, so that $\theta_2^s = \frac{1}{N}\sum_{i=1}^N m_1^i$. The players then receive the value of $\theta_2^s$ from the server and use their defense strategy to arrive at a final estimate $\theta_2^i = d(\theta_2^s, x^i)$ of the mean. For simplicity of notation, we ignore the dependence of all values on the time $t = 1$ in the rest of this and the next section.

**Reward functions**   To model competitive incentives, the reward of each player needs to increase as their own estimate of the mean becomes better, and as the estimates of other players become worse. Therefore, a natural reward function is:

$$\mathcal{R}^i(\theta^1, \ldots, \theta^N, \mu) = \frac{\sum_{j \neq i} \|\theta^j - \mu\|^2}{N - 1} - \lambda_i \|\theta^i - \mu\|^2, \tag{2}$$

for some $\lambda_i \geq 0$. The value of $\lambda_i$ quantifies to what extent player $i$ prioritizes the quality of their own estimate over damaging the estimates of the other players.

**Attack strategies**   We assume that players choose what estimates to communicate by deciding how to perturb the empirical mean of their data. Specifically, each player $i$ selects parameters $\alpha^i(x^i) \geq 0, b^i(x^i) \in \mathbb{R}^d$ based on their sample, in a potentially randomized manner, and communicates:

$$m^i = \bar{x}^i + \alpha^i(x^i)\xi^i + b^i(x^i), \tag{3}$$

where $\bar{x}^i = \frac{1}{n}\sum_{j=1}^n x_j^i$, $\mathbb{E}[\xi^i] = 0$ and $\mathbb{E}\|\xi^i\|^2 = 1$. Here $\bar{x}^i$ is the standard empirical mean of $x^i$, while $\alpha^i$ represents the magnitude of the noise player $i$ adds to the estimate and $b^i(x^i)$ is an additional bias term. Note that the case of $b^i(x^i) = 0$ recovers the data-hiding attack discussed in Section 3.2.

In order to prevent "non-general" strategies, such as simply setting $m^i = \mu$, that cannot be analyzed properly as their success depends on the true parameter $\mu$, we assume that players do not base their strategies on guesses about $\mu$ beyond the information they obtained from $\bar{x}^i$. Formally, we assume

$$\mathbb{E} < \bar{x}^j - \mu, b^j(x^j) >= 0. \tag{4}$$

This prevents $b^i$ from linearly encoding additional knowledge about $\mu$ and for example holds whenever $b^j(x^j)$ is independent of the residuals $\bar{x}^j - \mu$. We also assume that the noise variables $\xi^i$ are independent of each other and all $x_j^k$ and $\alpha^k(x^k)$, but make no further distributional assumptions about $\xi^i$. Indeed, all of our theorems will hold regardless of any additional assumptions about $\xi^i$.

We denote this set of attack strategies as $\mathscr{A}$. Each element in $\mathscr{A}$ can be uniquely identified via the distribution of the noise $\xi^i$ and the functions $\alpha^i(x^i)$ and $b^i(x^i)$. As $\alpha = b = 0$ can be interpreted as covering the fully collaborative case, while $\alpha, b \to \infty$ covers the fully malicious case, the (adaptive) parameters $\alpha, b$ have a natural interpretation as measures of the *aggressiveness* of a player.

We also note that these attacks are *very general*: $m^i(x^i) - \bar{x}^i$ can always be written as the sum of a deterministic component $\hat{b}(x^i)$ and zero mean noise $\hat{\xi}(x^i)$, such that *(4) and the fixed distribution of $\xi^i$ are the only assumptions separating us from the most general possible set of attacks strategies.*

**Defense strategies**   In the defense stage each player uses the received estimate $\theta^s = \bar{m} = \frac{1}{N}\sum_{i=1}^N m^i$ and their local data $x^i$ to compute a final estimate of the unknown mean. Two extreme approaches for player $i$ are being fully cautious and using their local mean $\bar{x}^i$ only, or fully trusting other players and computing the average of all sent updates, corrected for their own manipulation, that is $\bar{m}^i = \frac{1}{N}\left(N\theta^s - m^i + \bar{x}^i\right)$. We consider defense strategies that take a weighted average of these two extremes: Each player $i$ chooses a parameter $\beta^i \in [0, 1]$ and constructs a final estimate

$$\theta^i = (1 - \beta^i)\bar{m}^i + \beta^i \bar{x}^i. \tag{5}$$

Denote the described set of defenses, as $\mathscr{D}$. Each element in $\mathscr{D}$ is uniquely identified via the corresponding parameter $\beta$ with $\beta = 0$ and $\beta = 1$ covering the extreme cases from above. Since $\beta$ can be used to interpolate between these two extremes, it can be seen as a measure of *cautiousness*.

We do not cover more complicated defense strategies for two reasons: First, our proposed mechanisms will incentivize players to be honest *even without any defenses* such that more advanced defense mechanisms are not necessary. Second, as defenses can be seen as a method for mean estimation, analyzing the optimality for general classes of defenses would be fundamentally challenging for $d \geq 3$ due to Stein's Paradox (Stein, 1956), even for a single player version of our game.

## 4.1 Expected rewards and Nash equilibria

We now analyze the game with strategy set $(\mathscr{A} \times \mathscr{D})$. As the specific distribution of $\xi^i$ does not affect the players' rewards, the attack and defense strategies are for all relevant purposes uniquely determined by the functions $\alpha$, $b$, and $\beta$ respectively. We abuse notation and consider $(\alpha^i, b^i, \beta^i)$ as the strategy of player $i$. First we derive a formula for the MSE of a player, for a fixed strategy profile.

**Theorem 4.1.** *Let $\mathcal{D}$ be as described above. Then the expected mean squared error (MSE) of player $i \in [N]$ for any strategy profile $((\alpha^1, b^i, \beta^1), \ldots, (\alpha^N, b^N, \beta^N)) \in (\mathscr{A} \times \mathscr{D})^N$ is:*

$$
\mathbb{E}\left(\|\theta^i - \mu\|^2\right) = \left(1 - \beta^i\right)^2 \left( \frac{\sigma^2}{Nn} + \frac{\sigma_\star^2}{N} + \frac{1}{N^2}\sum_{j\neq i}\mathbb{E}(a^j(x^j)^2) + \frac{1}{N^2}\mathbb{E}\|\sum_{j\neq i}b^j(x^j)\|^2 \right)
$$
$$
+ (\beta^i)^2(\frac{\sigma^2}{n} + \sigma_\star^2) + 2\left(1 - \beta^i\right)\beta^i(\frac{\sigma^2}{Nn} + \frac{\sigma_\star^2}{N})
$$

This is proven in Appendix C similar to the bias-variance decomposition. This result allows us to analyze the expected rewards defined by equation (2) of the players for any strategy profile.

One can immediately see that there is no incentive for players to cooperate as long as $\beta^i < 1$: other players $j$ can always increase their reward by increasing $\mathbb{E}(a^j(x^j)^2)$ (unless it is already infinite). But for finite $\mathbb{E}(a^j(x^j)^2)$ and $\mathbb{E}\|\sum_{j\neq i}b^j(x^j)\|^2$, the optimal $\beta^i$ can be shown to never equal one, such that equilibria are only possible "at infinity":

**Corollary 4.2.** *The game defined by the reward in equation (2) and the set of strategies $\mathscr{A} \times \mathscr{D}$ does not have any (pure or mixed) Nash equilibrium for which $\mathbb{E}(\alpha^j(x^j)^2)$ and $\mathbb{E}\|b^j(x^j)\|^2$ are finite for all players.*

For details, see Appendix C. This shows that our defenses are unable to prevent maximal dishonesty by at least some players and formalizes a simple intuitive observation: as long as a player considers other players' updates at all, others are incentivized to reduce the information their updates convey about their samples. As at equilibrium at least one other player will infinitely distort the server estimate $\bar{m}$, *no player can benefit from collaborative learning without modifications to the protocol.*

# 5 Mechanisms for incentivizing honesty

Given the impossibility of successful learning with rational competing agents in the simple mean estimation setting, we shift our focus to modifications of the protocol that allow for honest Nash equilibria (that is, equilibria where $\alpha^j(x_j) = b^j(x^j) = 0, \forall j$). To this end, we design two mechanisms that seek to penalize dishonest players proportionally to the magnitude of their manipulations (and, thus, the damage caused to other players). Note that this is *complimentary* to robust estimation methods (Diakonikolas et al., 2019) that can reduce but not eliminate the impact of manipulations.

The first relies on explicit side payments and requires transferable utility (that is, the existence of an outside resource $R$ such as money, that is valued equally and on the same scale as the reward $\mathcal{R}$ by all players). The second is a modification of the FL protocol, in which the server adds noise to the estimates it sends to players that have sent suspicious updates. Importantly, for both mechanisms, the penalties can be computed by the server without the need for knowing $\alpha^i, b^i$ or other additional information beyond the players' updates, and without prior knowledge of the true distribution $\mathcal{D}$.

## 5.1 Efficient solution for fully transferable utility

We first consider the case of transferable utility. In this case, we introduce a more general *penalized reward* for player $i$, which is given by

$$
\mathcal{R}_p^i = \mathcal{R}^i(\theta^1, \ldots, \theta^N, \mu) - p^i(m^1, \ldots, m^N).
$$

Here $p^i(m^1, \ldots, m^N)$ denotes a penalty paid by player $i$ to the server, measured in terms of the resource $R$ and depending on the messages that the clients sent. As players value the reward and resource equally, they optimize for $\mathcal{R}_p^i$ instead of $\mathcal{R}^i$.

Inspired by peer prediction (Miller et al., 2005) we consider a penalty for player $i$ proportional to the squared difference between that player's update and the average update sent by all players:

$$p^i(m^1, \ldots, m^N) = C\|m^i - \bar{m}\|^2, \tag{6}$$

for some constant $C \geq 0$. This is a natural measure of the "suspiciousness" of the client's update. In order to prevent excessive payments for honest players, we redistribute the penalties as

$$p'^i(m^1, \ldots, m^N) = C\|m^i - \bar{m}\|^2 - \sum_{j \neq i} \frac{C\|m^j - \bar{m}\|^2}{N - 1}$$

This redistribution also makes it possible to implement our mechanism in a decentrally with messages sent publically, if players are able to credibly commit to the implied payments to other players.

Theorem 5.1 establishes that this penalty can incentivize full honesty for the right choice of $C$:

**Theorem 5.1.** *In the setting of Theorem 4.1 for the penalized game with rewards*

$$\mathcal{R}_{p'}^i = \frac{\sum_{j \neq i} \|\theta^j - \mu\|^2}{N - 1} - \lambda_i \|\theta^i - \mu\|^2 - p'^i(m^1, \ldots, m^N)$$

*the strategy profile $\alpha^j = b^j = \beta^j = 0$ for all $j$ is a Nash equilibrium, whenever $C > \frac{1}{(N-1)^2 - 1}$ and maximizes the sum of all players' rewards among equilibria whenever $\lambda \geq 1$ for all players.*

*At this equilibrium, the expected penalty $p^i(m^1, \ldots, m^N)$ paid by each player $i$ is equal to $0$. Each player is incentivized to participate in the penalized game rather than relying on their own estimate, whenever $N > 2$, the other $N - 1$ players participate at the honest equilibrium, and $\lambda_i > \frac{N}{(N-1)^2}$.*

Intuitively, our incentive mechanism is effective because $\alpha^i$ and $b^i$ only affect the first term of the original reward of player $i$ (equation 2), as well as the penalty, to which they contribute at most as $\frac{1}{N^2}$ and $-\frac{(N-1)^2-1}{N^2}C$ respectively. At the honest equilibrium every player's MSE is in $O(\frac{1}{N})$, such that for large $N$ a player can strongly improve their own error by joining the collaboration, while barely affecting others' errors. For a complete proof consider Appendix D.1.

Theorem 5.1 shows that our mechanism fulfills two desirable properties: *(budget) balance* and *(ex-ante) individual rationality/voluntary participation* (Jackson, 2014). The first property means that the server neither makes a profit nor a loss. The second holds as long as $\lambda_i > \frac{N}{(N-1)^2}$ and the other players take part in the optimization, and means that a player will receive better reward at the game's equilibrium, than when learning with their own data, despite the penalties assigned by the server. While non-honest equilbria exist, these are difficult to coordinate on (as they lack the natural symmetric Schelling point of honesty), while also yielding less total reward summed across players than honesty, such that there is no strong incentive for such coordination.

## 5.2 Non-transferable utility

We now discuss a way to achieve similar results in the case of non-transferable utility, where players' rewards are not translatable to monetary terms. Instead of modifying players' reward function, we modify the FL protocol, altering the server's messages to players. This effectively results in a robust learning algorithm that the server can implement. We do so by letting the server send noisier versions of its mean estimate to players whose messages suspiciously deviate from the average of all players' updates. The penalization scheme is designed in a way such that players receive expected rewards $\mathbb{E}(\mathcal{R}^i)$ similar to the expected penalized rewards $\mathbb{E}(\mathcal{R}_{p'}^i)$ in the previous section. This is conceptually similar to methods against free-riding (e.g. Karimireddy et al. (2022)), which often tie the accuracy of the model a client receives in an FL setting to the client's overall contribution to model training.

**Theorem 5.2.** *Consider the modified game with reward $\mathcal{R}^i = \frac{\sum_{j \neq i} \|\theta^j - \mu\|^2}{N-1} - \lambda_i \|\theta^i - \mu\|^2$, where player $i$ receives an estimate $\bar{m} + \sqrt{C}\epsilon^i \|m^i - \bar{m}\|$ for independent noise $\epsilon^i$ with mean $\mathbb{E}\epsilon^i = 0$ and "variance" $\mathbb{E}\|\epsilon^i\|^2 = 1$, instead of the empirical mean $\bar{m}$, from the server. Then honesty*

$(\alpha^i = 0, b^i = 0, \beta^i = \frac{C}{C+1})$ *is a Nash equilibrium, as long as* $C > \frac{1}{\lambda_i(N-1)^2-1}$ *and* $\lambda_i > \frac{1}{(N-1)^2}$. *Furthermore, honesty maximizes the sum of all players' rewards among equilibria whenever* $\lambda_i \geq 1$ *for all players.*

*For fixed* $\lambda_i = \lambda, k > 1$ *and* $C = \frac{k}{\lambda(N-1)^2-1}$, $\mathbb{E}\left(\|\theta^i - \mu\|^2\right) = O\left(\frac{\sigma^2}{Nn} + \frac{\sigma_*^2}{N}\right)$ *and players are incentivized to participate in the penalized game rather than relying on their own estimate, whenever* $N \geq 2$, *the other* $N-1$ *players participate at the honest equilibrium and* $\lambda \geq 1$.

Essentially, the noise added by the server increases the expected MSE for player $i$ by $C\|m^i - \bar{m}\|^2$, producing similar incentives as in Theorem 5.1. Theorem 5.2 is proven in Appendix D.2. We obtain voluntary participation if at least two other players participate at the honest equilibrium and $\lambda > 1$.

**The benefits of modeling clients' rationality**   Note that at the equilibrium, players' MSEs are of the same order as if all clients honestly communicated their sample means. In particular, when $\sigma^* = 0$ (i.e. homogeneous clients), this is in contrast to known negative results for worst-case poisoning attacks Qiao & Valiant (2018) and single-round Byzantine robustness Alistarh et al. (2018). In this sense, our modified protocol acts as a robust and efficient collaborative learning algorithm. This is possible because our data corruption model is derived by explicitly modeling clients' incentives.

# 6   Beyond mean estimation: stochastic gradient descent

We now extend the ideas from the last section to multi-round collaborative Stochastic Gradient Descent (SGD) in the FL setting. We show that under stronger assumptions, mechanisms similar to those described in Section 5.1 can still provide arbitrary bounds on manipulations by rational players. Again, this is *complimentary* to methods for robust federated learning such as median-based gradient aggregation that can reduce but not eliminate the impact of existing manipulations.

## 6.1   The game and rewards

We consider a $T$-round version of the game described in Section 3. The FL protocol is designed to minimize a loss function $f_{\mathcal{D}}(\theta)$ over a closed and convex set of model parameters $\theta \in W \subset \mathbb{R}^d$ that contains the global minimizer of $f_{\mathcal{D}}(\theta)$. At every time step $t$, each player is meant to communicate an update $g_t(\theta_t^s, x^i)$ with expectation $\nabla f(\theta_t^s)$. We denote by $e_t^i(\theta_t) = g_t(\theta_t^s, x^i) - \nabla f(\theta_t)$ the difference between the gradient and the estimate, which is a deterministic function of $\theta_t$ for fixed data $x^i$, but is assumed to fulfill $\mathbb{E}_{x^i} e_t^i(\theta) = 0$ for all $\theta$, and to be independent across time and players. Intuitively, $f_{\mathcal{D}}(\theta)$ can be thought of as an expected loss $\mathbb{E}_{x \sim \mathcal{D}} f_x(\theta)$, for which player $i$ computes approximate stochastic gradients as $\nabla f_{x_t^i}(\theta_{t-1}^s)$ using their $t$-th sample $x_t^i$. The message sent by player $i$ is termed $m_t^i$. The server averages the received messages to compute a gradient estimate $\bar{m}_t = \frac{1}{N}\sum_i m_t^i$ and updates the parameter $\theta$ via $\theta_{t+1}^s = \Pi_W(\theta_t^s - \gamma_t \bar{m}_t)$, using a fixed learning rate schedule $\gamma_t$ and the projection $\Pi_W$ onto $W$. Finally, the server sends the updated $\theta_{t+1}^s$ to all players.

**Strategies and rewards**   We consider attacks of the form $m_t^i = g_t(\theta_t^s, x^i) + \alpha_t^i \xi_t^i$, that is, the true gradient estimate plus random noise of the form $\alpha_t^i \xi_t^i$, where $\mathbb{E}\xi_t^i = 0$, $\mathbb{E}\|\xi_t^i\|^2 = 1$ and $\xi_t^i$ is sampled independent from the other $\xi_{t'}^j$ and the algorithm's trajectory. The set of attack strategies $\mathcal{A}^g$ is then described by the sequence of aggressiveness parameters $\alpha_t^i > 0$, which we assume to be selected in advance, independent of the optimization trajectory. Since defenses were already ineffective for mean estimation, we directly focus on mechanisms and only consider adjustments to the server's final estimate for the noise player $i$ added themselves in the final step $T$: $\theta_{T+1}^i = \theta_T - \frac{\alpha_T^i}{N}\xi_T^i$. The assumption of non-adaptive strategies is needed to avoid complicated dependencies between successive SGD rounds and more sophisticated attack strategies are beyond the scope of our analysis.

We do not consider a bias term for these attacks. Our unbiased attacks have natural interpretations, both in terms of *hiding randomly selected data points* and *differential privacy defenses*, and unlike in the mean estimation case, the effects of a fixed-direction attack on the loss $f_{\mathcal{D}}(\theta)$ can strongly depend on the current estimate $\theta_t^s$ and the attack's precise direction, making it substantially harder to analyze such attacks. That said, it is easy to see that if the server aims to optimize $f_{\mathcal{D}}(\theta_{T+1})$ and is allowed to shift its estimate $\theta_t^s$ to defend against fixed-direction attacks at every step $t$, the fixed direction attacks would be neutralized by the server at any equilbrium, unless they inadvertently improved $f_{\mathcal{D}}(\theta_{T+1})$.

Given these strategies and a Lipschitz function $U_i : \mathbb{R}^N \to \mathbb{R}$, player $i$ aims to maximize the reward

$$\mathcal{R}_U^i = U^i(f(\theta_{T+1}^1), ..., f(\theta_{T+1}^N)). \tag{7}$$

It is easy to see that this game does not always have an equilibrium and players are often incentivized to lie aggressively. In particular, we recover the mean estimation setting with $\beta^i = 0$ when setting $U^i(x) = \sum_{j \neq i} \|\theta - \theta^j\|^2 - \lambda_i \|\theta - \theta^i\|^2$, $T = 1$, $\theta_1 = 0$ and $\gamma_1 = 0.5$, as $\frac{d}{d\theta} \|\theta - x_i^j\|^2 = 2(\theta - x_i^j)$.

Our next result establishes that players can be incentivized to be arbitrarily honest, using a penalty scheme similar to the one in Section 5.1. Again, penalties are redistributed such that players' penalties have zero expectation whenever $\alpha_t^i = \alpha_t^j$ for all $i, j, t$. We set $\bar{m}_t = \frac{1}{N} \sum_j m_t^j$ and

$$\mathcal{R}_{U_p}^i = U^i(f(\theta_{T+1}^1), ..., f(\theta_{T+1}^N)) - \sum_{t=1}^T C_t \|m_t^i - \bar{m}_t\|^2 + \frac{1}{N-1} \sum_{k \neq i} \sum_t C_t \|m_t^k - \bar{m}_t\|^2$$

for constants $C_t$. With this penalized reward we prove:

**Theorem 6.1.** *Assume $f$ is $B$-smooth and $L$-Lipschitz on $W$ and $m$-strongly convex on $\mathbb{R}^d$ (See Appendix 2 for definitions of these properties). Also assume that for all $i$ and $t$ the gradient noise $e_t^i$ is $B'$-Lipschitz with probability one and that there exist scalars $M \geq 0$ and $M_V \geq 0$, such that for all $t$:*

$$\mathbb{E}_{s_i}(\|(e_t^i(\theta_t^s)\|^2) \leq M + M_V \|\nabla f(\theta_t^s))\|_2^2. \tag{8}$$

*Set the learning $\gamma_t = \frac{4}{\eta m + tm}$ for an integer constant $\eta > 1$, such that $\frac{4}{\eta m + m} \leq \frac{1}{B(M_V/N+1)}$.*

*Then if $U^i$ is $l_1$-Lipschitz with constant $L_U$ for all players $i$, all player's best response strategy fulfill $\alpha_i^t \leq \frac{8LL_U N}{C^t(N-2)m\sqrt{T+\eta}}$ independent of other players' strategies. If $\alpha_t^i = \alpha_t^j, \forall i, j, t$, each player's expected penalty is 0. For $\epsilon > 0$, $C^t \geq \frac{8LL_U N}{\epsilon(N-2)m\sqrt{T+\eta}}$ yields $\alpha_i^t \leq \epsilon$ for rational players. In that case, if in addition $W$ is bounded and we have that $P(\exists t \leq T : \Pi_W(\theta_t^s - \gamma_t \bar{m}_t) \neq \theta_t^s - \gamma_t \bar{m}_t) \in O(\frac{1}{NT})$, we get $\mathbb{E}(f(\theta_{T+1}) - f(\theta^*)) \in O(\frac{1+M+\epsilon^2}{NT}) + O(\frac{1}{T^2})$.*

Intuitively, small perturbations of gradient estimates sent to the server should only have a small effect on the final learnt model. Formally, our assumptions allow us to inductively prove bounds on the difference between the values of $\mathbb{E}f(\theta_{T+1}^j)$ in a clean ($\alpha_t^i = 0$) scenario and a scenario with other values of $\alpha_t^i$. By setting $C^t$ large enough, we can then ensure that the penalties paid by a dishonest player always outweigh their effect on the final model. The condition on $\Pi_W$ ensures that SGD is not slowed down by projections and holds for "sufficiently large" distances between the boundary of $W$ and $\theta^1$. See Appendix E.2 for more details on this and the theorem proof. Theorem 6.1 implies that for sufficiently large $C^t$, despite *all players acting strategically*, the model converges at speed comparable to when all clients share clean updates (where the convergence rate is $O(\frac{1+M}{NT}) + O(\frac{1}{T^2})$), thereby ensuring full learning benefits from the collaboration. Moreover, as long as all players are equally honest, this is achieved with zero expected penalties for players and thus with budget balance.

## 7 Experimental results

To verify that our mechanisms can work for SGD in the non-convex case we simulate FedSGD McMahan et al. (2017) with clients corrupting their messages to different degrees, and record how players' rewards and penalties are affected by their aggressiveness $\alpha$ for different penalty constants $C$. The $\alpha_t^i$ that empirically maximizes a player's reward is an approximate best response for a given $C$ and fixed $\alpha_t^j$ for $j \neq i$, and should thus be small for a successful mechanism.

First, we simulate FedSGD, treating each writer as a client, to train a CNN-classifier using the architecture provided by Caldas et al. (2018) for the FeMNIST dataset that consists of characters and numbers written by different writers. Second, we train a two-layer linear classifier with 384 hidden neurons on top of frozen "bert-base-uncased" embeddings on the Twitter Sentiment Analysis dataset from Caldas et al. (2018). In both cases, we randomly select $m = 3$ clients and compute a gradient estimate $g_t^i$ for the cross entropy loss $f$ using a single batch containing 90% of the data provided by the corresponding writer at time step $t < T = 10650$ [1]. We test on the remaining 10% of the data.

---

[1]$T$ was selected as 3 times the number of writers in the FeMNIST dataset, as reported by Caldas et al. (2018).

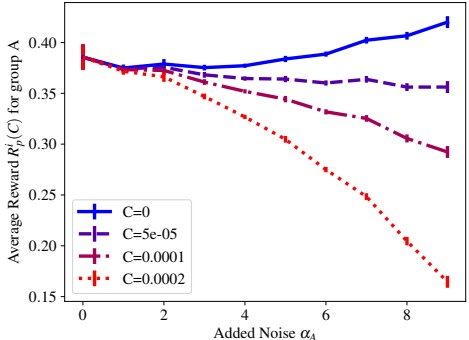

Figure 1: FeMNIST Dataset

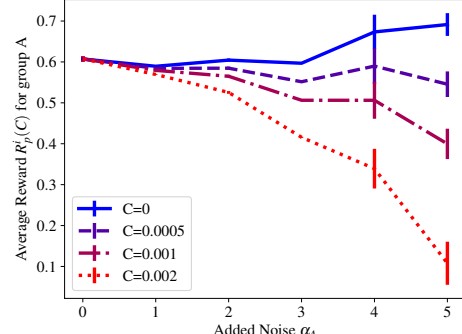

Figure 2: Twitter Dataset

Figure 3: Average reward $\mathcal{R}_p^i(C)$ received by players in group $A$ for $\alpha_B = 0$ and varying $\alpha_A$. Different colors represent different penalty weights $C$. Results are averaged over 10 runs and error bars show the standard error.

For both experiments, we randomly split writers into two groups $A$ and $B$ containing one and two thirds of the writers respectively, and corrupt the gradient estimates $(m_t^i)_l = (g_t^i)_l + \alpha_A(\xi_t^i)_l$ sent by players in group $A$ for each weight tensor and bias vector $l$ separately, by adding isotropic normal noise $(\xi_t^i)_l$ with "variance" $E\|(\xi_t^i)_l\|^2 = 1$. We do the same with $\alpha_B$ for group $B$. At each step, the three corrupted gradients are then averaged (weighted by the corresponding writers' datasets' sizes) to $\bar{m}_t$, which is used to update our neural network's parameters $\theta_t$ with learning rate 0.06, i.e. $\theta_{t+1} = \theta_t - 0.06\bar{m}_t$. Unlike in our theorems, writers reuse the same data points whenever they calculate gradients. In the clean case ($\alpha_A = \alpha_B = 0$), our final models achieve a test accuracy of 86% on FeMNIST [2] and 63% on Twitter. We record both the cross-entropy loss $f$ achieved by the final model $\theta_T$ on the test set, as well as the sum of the squared deviations $\|m_t^i - \bar{m}_t\|^2$ incurred by each individual client $i$ across all steps. This allows us to calculate the estimated reward $\mathcal{R}_p^i(C) = f(\theta_T) - \sum_{t=0}^{T-1} \mathbb{I}_t(i)C\|m_t^i - \bar{m}_t\|^2 + \frac{1}{2}\sum_{k \neq i}\sum_{t=0}^{T-1} \mathbb{I}_t(i)\mathbb{I}_t(k)C\|m_t^k - \bar{m}_t\|^2$ received by every player for penalty weights $C$ held constant over time, where $\mathbb{I}_t(j)$ is a binary indicator equal to 1 whenever player $i$ provided an update at time $t$.

Figure 3 shows the average reward $\mathcal{R}_p^i(C)$ received by players in group $A$ for $\alpha_B = 0$ and $\alpha_A$ varying on the x-axis for different penalty weights $C$. It clearly shows that penalization decreases players' gains from adding noise even in the non-convex case, and that near-zero noise is optimal for players given sufficiently large penalty weights $C$. At the same time, despite client heterogeneity, the penalties paid by honest players are small: In the FeMNIST experiments, if all players are honest, an overwhelming majority (98%) of players end up paying less than 0.0031 on average (over 10 rounds), even at $C = 0.0002$. This is an order of magnitude under the increase in loss from moving from $\alpha_A = 0$ to $\alpha_A = 9$, which is already disincentivized for the substantially smaller penalty constant $C = 5e-5$. On the Twitter dataset, noise has a larger effect on the loss (the loss is degraded by 0.084 already at noise level $a_A = 5$ compared to 0.034 at noise level $a_A = 9$ for FeMNIST). This means that larger penalty weights are necessary to incentivize honesty. Correspondingly, the 98th percentile of penalties paid at the largest considered penalty level $C = 0.002$ is also larger (0.0243) than on FeMNIST. Additional experimental results can be found in Appendix B.2.

## 8    Conclusion

In this paper, we studied a framework for FL with strategic agents, where players compete and aim to not only improve their model but also damage others' models. We analyzed both the single- and multi-round version of the problem for a natural class of strategies and goals and showed how to design mechanisms that incentivize rational players to behave honestly. In addition, we empirically demonstrated that our approach works on realistic data outside the bounds of our theoretical results.

---

[2]For comparison, Caldas et al. (2018) aim for an accuracy threshold of 75% using 5% of the training data.

## Acknowledgments

This research was partially funded by the Ministry of Education and Science of Bulgaria (support for INSAIT, part of the Bulgarian National Roadmap for Research Infrastructure). Florian Dorner is grateful for financial support from the Max Planck ETH Center for Learning Systems (CLS). The authors would like to thank Mark Vero and Nikita Tsoy for their helpful feedback on earlier versions of this manuscript.

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

# A   Ethical considerations

Our theoretical results establish that penalties average out to zero in expectation, but substantial client heterogeneity can still cause large payments for individual honest participants. Despite our experimental results indicating that variance can be manageable even for real-world data, this is problematic for two reasons: The first is about fairness: Paying some honest participants large amounts, while demanding large amounts from others, based on what essentially amounts to luck is problematic, especially when the participants are individuals rather than firms, and when the nature of the problem might make it difficult for participants to gauge the order of magnitude of payments in advance. The second is about diversity: Clients that expect their data to deviate strongly from the mean of the overall data distribution might opt to not participate in FL with our mechanisms, even though underrepresented types of clients can provide data that is crucial to a model's broad performance and generalization. In our formalism, this problem is obscured by the assumption that all participants sample their data independently from the same distribution, and are unable to predict whether or not their data represents outliers.

Correspondingly, it is important to keep penalty weights $C$ as low as possible to reduce the likelihood of overwhelmingly large penalties, only apply our framework in the context of firms rather than individuals (for whom competitive incentives might often play less of a direct role either way), as well as ensure that the our assumptions about a common data distributions are plausible for the problem at hand. The former can be particularly challenging for non-convex problems, or convex problems with unknown problem parameters, for which no strong candidate for $C$ can be established theoretically.

Furthermore, our results do not establish collusion-proofness of our mechanism. While we expect our mechanism to be collusion-proof against small coalitions, there is a problem once the colluding coalition is large enough to significantly affect the mean estimate, as the shifted mean would reduce the penalty paid by each member of the coalition.

# B   Additional results

## B.1   Nash equilibria in the mean estimation game under bounded attacks

The conclusion that no Nash equilibria exist in the mean estimation game described in 4 is rather intuitive, since all participants have an incentive to send as modified an update as possible, therefore damaging the other players' estimates. In practice, however, attacking in an unbounded manner, that is, sending updates very far from the true mean, may not be plausible. Indeed, if most players send their true mean, one expects the variance of the estimates that the server receives to be of order $\mathcal{O}\left(\frac{\sigma^2}{n} + \sigma_\star^2\right)$. Therefore, players might in practice be reluctant to send estimates that are further than $A\sqrt{\frac{\sigma^2}{n} + \sigma_\star^2}$ away from there true local mean, for some constant A.

We therefore consider the same game as before, in the case when an upper bound A on the parameters $\alpha^i$ is given. Denote the resulting set of attack strategies by $\mathscr{A}_A$. Since $\mathscr{A}_A \subset \mathscr{A}^m$, Theorem 4.1 holds for the joint set of strategies $(\mathscr{A}_A^m \times \mathscr{D})$. Then we have the following

**Corollary B.1.** *In the setup of Theorem 4.1, if the set of available strategies is $\mathscr{A}_A \times \mathscr{D}$ for some constant $A > 0$, the only Nash equilibria of the game with $b^i(x^i) = 0$ fixed for all players $i$ are the strategy profiles for which:*

$$|\alpha_i| = A \quad and \quad \beta^i = \frac{A^2}{(\frac{\sigma^2}{n} + \sigma_\star^2)N + A^2} \quad \forall i \in [N]. \tag{9}$$

*Furthermore, at each of these equilibria the value of mean squared error of the estimate of each player $i$ is*

$$\mathbb{E}\left(\|\theta^i - \mu\|^2\right) = (\frac{\sigma^2}{n} + \sigma_\star^2)\frac{(1 + \frac{1}{\frac{\sigma^2}{n} + \sigma_\star^2}A^2)}{(N + \frac{1}{\frac{\sigma^2}{n} + \sigma_\star^2}A^2)}$$

As a result, whenever $A = \mathcal{O}(1)$, each player's estimate at any Nash equilibrium achieves a mean squared error of $\mathcal{O}\left(\frac{\sigma^2}{Nn} + \frac{\sigma_\star^2}{N}\right)$, which is of the same order as the MSE of the estimates that would have been obtained in a fully collaborative setting.

## B.2 Additional details on experiments

The network we train is based on the network used in the LEAF repository [3] but implemented in pytorch Paszke et al. (2019). It consists of two convolutional layers with relu activations, kernel size $5$, $(2, 2)$ padding and $32$ and $64$ filters, respectively, each followed by max pooling with kernel size and stride 2. After the convolutional layers, there is a single hidden dense layer with $2048$ neurons and a relu activation, and a dense output layer. All experiments were conducted using a single GPU each[4] per run.

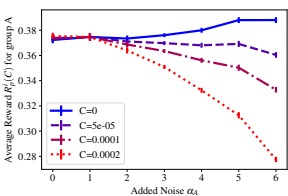
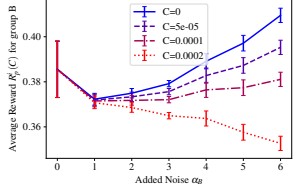
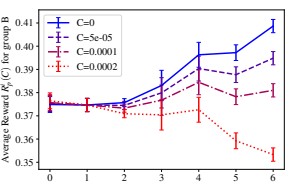

Figure 4: $\alpha_B = 1$, $\alpha_A$ varying.

Figure 5: $\alpha_A = 0$, $\alpha_B$ varying.

Figure 6: $\alpha_A = 1$, $\alpha_B$ varying.

Figure 7: Average reward $\mathcal{R}_p^i(C)$ received by players in a group for different values of $\alpha_A$ and $\alpha_B$. Different colors represent different penalty weights $C$. Results are averaged over 10 runs and error bars show the standard error.

We downloaded the FeMNIST and Twitter datasets using the code provided at https://github.com/TalwalkarLab/leaf/tree/master/data/femnist, opting not to filter writers that only have produced a small amount of samples. Correspondingly, our FeMNIST dataset contains 817851 examples of handwritten digits and characters written by a total of 3597 writers rather than the 805263 samples from 3550 writers reported in Caldas et al. (2018).

Figures 4, 5, 6 show results similar to 2 for $\alpha_B$ fixed to 1 instead of 0 (Figure 4), or $\alpha_B$ varying while $\alpha_A$ is fixed to 0 (Figure 5), or 1 (Figure 6), respectively. As payments are redistributed, the average payments for players in group $B$ increase slower with $\alpha_B$, as each individual's increase in payment is partially balanced out by an increase in received payments from encountering other members of group $B$ (Figures 5, 6). Meanwhile, figures 4 and 6 hint at honest players receiving money when others are adding noise: In both cases, players of one group receive slightly higher reward for larger penalties when they are honest ($\alpha = 0$) while the other group slightly cheats ($\alpha = 1$).

It is worth noting, that we do not perform a projection step in our experiments, such that numerical instabilities become an issue for large values of $\alpha$. In particular, for $\alpha_A > 6$ or $\alpha_B > 6$ we regularly observed NaN gradients on FeMNIST for one or more of our 10 runs.

Figure 10 show histograms over the total penalty for $C = 0.0002$ (FeMNIST) and $C = 0.002$ (Twitter) paid by each individual client over the whole 10650 steps for the honest case of $\alpha_A = \alpha_B = 0$, averaged over 10 runs. Clearly, most penalties are on the order of $0.01$ (FeMNIST) or $0.1$ (Twitter), which is substantially smaller (FeMNIST) / comparable (Twitter) to the negative effects of larger noise values on the order of $0.1$, which are strongly disincentivized by the considered penalty values.

Figure 13 shows additional results using SGD with Median-based rather than Mean-based aggregation as a baseline that is more robust to noisy gradients. We can see that while using Median-based aggregation helps, players can affect the loss as much as before by adding even more noise (see Figure 11), such that adding noise is still incentivized. At the same time, our mechanism still works: As can be seen in Figure 12, rewards still decrease with increased noise for sufficiently large $C$. Correspondingly, players remain disincentivized to add noise. This empirically suggests that our mechanism can also be applied to more advanced federated learning protocols that go beyond simple SGD with Mean-based aggregation.

---

[3]https://github.com/TalwalkarLab/leaf/blob/master/models/femnist/cnn.py
[4]We used assigned GPUs from a cluster that employs mostly, but not exclusively Nvidia V100 GPUs

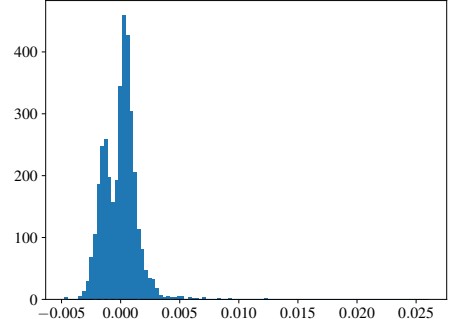 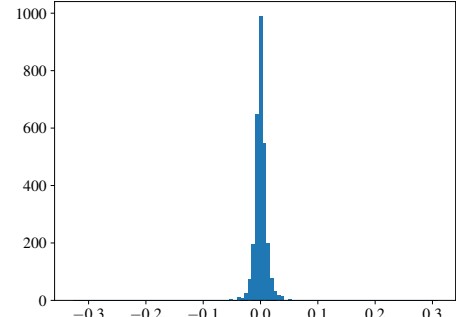

Figure 8: FeMNIST with penalty $C = 0.0002$     Figure 9: Twitter with penalty $C = 0.002$

Figure 10: Histogram of average (over 10 runs) total penalties paid by players for $\alpha_A = \alpha_B = 0$

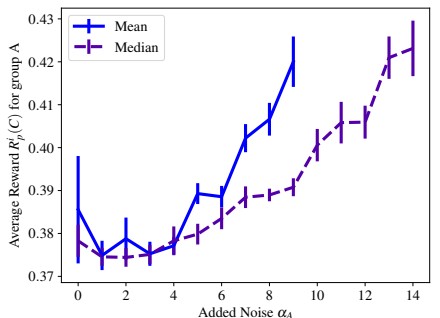

Figure 11: Mean vs Median-based aggregation in the unpenalized setting ($C = 0$).

Figure 12: Median-based aggregation with different penalty weights $C$.

Figure 13: Results on FeMNIST with median-based aggregation: Average reward $\mathcal{R}_p^i(C)$ (with penalties calculated in terms of the distance to the median rather than the mean for Median-based aggregation) received by players in group $A$ for $\alpha_B = 0$ and varying $\alpha_A$. Results are averaged over 10 runs and error bars show the standard error.

## C    Proofs on mean estimation

First we prove Theorem 4.1.

**Theorem C.1.** *Let $\mathcal{D}$ be as described in 4. Then, for any strategy profile $((\alpha^1, b^1, \beta^1), \ldots, (\alpha^N, b^N, \beta^N)) \in (\mathscr{A} \times \mathscr{D})^N$ and for any player $i \in [N]$, the expected mean squared error of player $i$ is:*

$$
\mathbb{E}\left(\|\theta^i - \mu\|^2\right)
$$

$$
= \left(1 - \beta^i\right)^2 \left( \frac{\sigma^2}{Nn} + \frac{\sigma_\star^2}{N} + \frac{1}{N^2} \sum_{j \neq i} \mathbb{E}(a^j(x^j)^2) + \frac{1}{N^2} \mathbb{E}\| \sum_{j \neq i} b^j(x^j)\|^2 \right)
$$

$$
+ (\beta^i)^2 (\frac{\sigma^2}{n} + \sigma_\star^2) + 2 \left(1 - \beta^i\right) \beta^i (\frac{\sigma^2}{Nn} + \frac{\sigma_\star^2}{N})
$$

*Proof.* Recall that $\theta^i = (1 - \beta^i)\bar{m}^i + \beta^i \bar{x}^i = (1 - \beta^i)\frac{1}{N}\left(N\theta^s - m^i + \bar{x}^i\right) + \beta^i \bar{x}^i$. Therefore, $\theta^i - \mu = (1 - \beta^i)\left(\frac{1}{N}\left(N\theta^s - m^i + \bar{x}^i\right) - \mu\right) + \beta^i\left(\bar{x}^i - \mu\right)$.

Denote
$$Y = \frac{1}{N}\left(N\theta^s - m^i + \bar{x}^i\right) - \mu, \quad Z = \bar{x}^i - \mu$$

Then we have:
$$\mathbb{E}\left(\|\theta^i - \mu\|^2\right) = \mathbb{E}\left(\|\left(1 - \beta^i\right)Y + \beta^i Z\|^2\right) \tag{10}$$
$$= \left(1 - \beta^i\right)^2 \mathbb{E}\|Y\|^2 + (\beta^i)^2 \mathbb{E}\|Z^2\| + 2\left(1 - \beta^i\right)\beta^i \mathbb{E}(Y^t Z)$$

The second term is the mean squared error of the (clean) estimate based on the local data of player $i$ and so:

$$\mathbb{E}(\|Z\|^2) = \mathbb{E}\left(\|\bar{x}^i - \mu\|^2\right)$$
$$= \mathbb{E}\left(\|\frac{1}{n}\sum_j x_j^i - \mu\|^2\right)$$
$$= \mathbb{E}\left(\|\frac{1}{n}\sum_j (x_j^i - \mu)\|^2\right)$$
$$= \frac{1}{n^2}\mathbb{E}\left(\|\sum_j (x_j^i - \mu)\|^2\right)$$
$$= \frac{1}{n^2}\mathbb{E}\left(\|\sum_j (x_j^i - \mu^i)\|^2 + \|\sum_j (\mu^i - \mu)\|^2\right)$$
$$= \frac{1}{n^2}\left(\sum_j \mathbb{E}\left(\|(x_j^i - \mu^i)\|^2\right) + \mathbb{E}\left(\|n(\mu^i - \mu)\|^2\right)\right)$$
$$= \frac{n}{n^2}\mathbb{E}\left(\|(X^i - \mu^i)\|^2\right) + \mathbb{E}\|(\mu^i - \mu)\|^2$$
$$= \mathbb{E}\left(\frac{\sigma_i^2}{n}\right) + \sigma_\star^2$$
$$= \frac{\sigma^2}{n} + \sigma_\star^2$$
$$\tag{11}$$

Where we can split $\mathbb{E}\left(\|\sum_j (x_j^i - \mu)\|^2\right)$ because of the tower law of expectation and using that $\mathbb{E}(x_j^i) = \mu^i$ conditional on the value of $\mu^i$; and the squared norm of the last sum factors, as all $x_j^i - \mu^i$ terms are independent and have zero mean such that

$$\mathbb{E}\|\sum_j (x_j^i - \mu)\|^2$$

$$= \mathbb{E}\|\sum_{j>1}(x_j^i - \mu)\|^2 + \mathbb{E}\|x_1^i - \mu\|^2 + 2\mathbb{E}\left((x_1^i - \mu)^t \left(\sum_{j>1}(x_j^i - \mu)\right)\right)$$

$$= \mathbb{E}\|\sum_{j>1}(x_j^i - \mu)\|^2 + \mathbb{E}\|x_1^i - \mu\|^2 + 2\mathbb{E}\left(\sum_d (x_1^i - \mu)_d \left(\sum_{j>1}(x_j^i - \mu)_d\right)\right)$$

$$= \mathbb{E}\|\sum_{j>1}(x_j^i - \mu)\|^2 + \mathbb{E}\|x_1^i - \mu\|^2 + 2\sum_d \mathbb{E}\left((x_1^i - \mu)_d \left(\sum_{j>1}(x_j^i - \mu)_d\right)\right)$$

$$= \mathbb{E}\|\sum_{j>1}(x_j^i - \mu)\|^2 + \mathbb{E}\|x_1^i - \mu\|^2 + 2\sum_d \mathbb{E}(x_1^i - \mu)_d \mathbb{E}\left(\sum_{j>1}(x_j^i - \mu)_d\right)$$

$$= \mathbb{E}\| \sum_{j>1}(x_j^i - \mu)\|^2 + \mathbb{E}\|x_1^i - \mu\|^2 + 2\sum_d 0\mathbb{E}\left(\sum_{j>1}(x_j^i - \mu)_d\right)$$

$$= \mathbb{E}\| \sum_{j>1}(x_j^i - \mu)\|^2 + \mathbb{E}\|x_1^i - \mu\|^2$$

and so on, allowing us to inductively factor the sum.

Now, to compute the terms that depend on $Y$, note that by definition $\theta^s = \frac{1}{N}\sum_{i=1}^N m^i$ and therefore:

$$Y = \frac{1}{N}\left(N\theta^s - m^i + \bar{x}^i\right) - \mu$$

$$= \frac{1}{N}\left(\sum_{j=1}^N m^j - m^i + \bar{x}^i\right) - \mu$$

$$= \frac{1}{N}\sum_{j\neq i} m^j + \frac{1}{N}\bar{x}^i - \mu$$

Define random variables:

$$Y_1 = \frac{1}{N}\sum_{j\neq i}^N m^j, \quad Y_2 = \frac{1}{N}\bar{x}^i - \mu$$

so that $Y = Y_1 + Y_2$. This implies that:

$$\mathbb{E}(Y^t Z) = \mathbb{E}\left(Z^t(Y_1 + Y_2)\right) = \mathbb{E}(Z^t Y_1) + \mathbb{E}(Z^t Y_2).$$

Since the value of $Y_1$ only depends on the data of all players $j \in \{1, 2, \ldots, N\}/\{i\}$ and the value of $Z$ depends only on the data of player $i$, it follows that $Z$ and $Y_1$ are independent and therefore $\mathbb{E}(Z^t Y_1) = \mathbb{E}(Z)^t\mathbb{E}(Y_1)$.

Therefore,

$$\mathbb{E}(Z^t Y) = \mathbb{E}\left(Z^t(Y_1 + Y_2)\right)$$

$$= \mathbb{E}(Z)^t\mathbb{E}(Y_1) + \mathbb{E}(Z^t Y_2)$$

$$= \mathbb{E}\left(\bar{x}^i - \mu\right)^t \mathbb{E}\left(\frac{1}{N}\sum_{j\neq i} m^j\right) + \mathbb{E}\left(\left(\bar{x}^i - \mu\right)^t\left(\frac{1}{N}\bar{x}^i - \mu\right)\right)$$

$$= 0 + \frac{1}{N}\mathbb{E}\left(\|\bar{x}^i\|^2\right) - \frac{1}{N}\mu^t\mathbb{E}\left(\bar{x}^i\right) - \mu^t\mathbb{E}\left(\bar{x}^i\right) + \|\mu\|^2$$

$$= \frac{1}{N}\mathbb{E}\left(\|\bar{x}^i\|^2\right) - \frac{1}{N}\mu^t\mu - \mu^t\mu + \|\mu\|^2$$

$$= \frac{1}{N}\mathbb{E}\left(\|\bar{x}^i\|^2\right) - \frac{1}{N}\|\mu\|^2$$

$$= \frac{1}{N}\mathbb{E}\left(\|\frac{1}{n}\sum_j x_j^i\|^2\right) - \frac{1}{N}\|\mu\|^2$$

$$= \frac{1}{N}\mathbb{E}\left(\|(\frac{1}{n}\sum_j x_j^i - \mu) + \mu\|^2\right) - \frac{1}{N}\|\mu\|^2$$

$$= \frac{1}{N}\mathbb{E}\left(\|(\frac{1}{n}\sum_j x_j^i - \mu)\|^2 + \|\mu\|^2\right) - \frac{1}{N}\|\mu\|^2$$

$$= \frac{1}{N}\mathbb{E}\left(\|(\frac{1}{n}\sum_j x_j^i - \mu)\|^2\right)$$

$$= \frac{\sigma^2}{Nn} + \frac{\sigma_\star^2}{N} \tag{12}$$

Where the squared norm of the sum factors as all $x_i^j - \mu$ terms have zero mean, while $\mu$ is a constant and the last equality follows from the computations in (11).

Now, by definition $m^j = \bar{x}^j + \alpha^j(x^j)\xi^j + b^j(x^j)$ for every $j \in [N]$, where the $\xi^j$ are random variables with zero mean and unit "variance" $\mathbb{E}(\|\xi^j\|^2)$. Note that because $Y_1$ and $Y_2$ depend only on the data of all players $j \in \{1, 2, \ldots, N\}/\{i\}$ and on the data of player $i$ respectively, they are independent and we get:

$$\mathbb{E}\left(\|Y^2\|\right) = \mathbb{E}\left(\|Y_1 + Y_2\|^2\right)$$
$$= \mathbb{E}\left(\|Y_1\|^2\right) + \mathbb{E}\left(\|Y_2\|^2\right) + 2\mathbb{E}(Y_1)^t\,\mathbb{E}(Y_2) \tag{13}$$

In order to calculate the value of $\mathbb{E}\left(Y^2\right)$ we have to compute $\mathbb{E}(Y_1), \mathbb{E}(Y_2), \mathbb{E}\left(\|Y_1\|^2\right)$ and $\mathbb{E}\left(\|Y_2\|^2\right)$. We have:

$$\mathbb{E}(Y_1) = \mathbb{E}\left(\frac{1}{N}\sum_{j\neq i}\left(\bar{x}^j + \alpha^j(x^j)\xi^j + b^j(x^j)\right)\right)$$
$$= \frac{1}{N}\sum_{j\neq i}\mathbb{E}\left(\bar{x}^j\right) + \frac{1}{N}\sum_{j\neq i}\mathbb{E}(\alpha^j(x^j))\mathbb{E}\left(\xi_j\right) + \frac{1}{N}\sum_{j\neq i}\mathbb{E}(b^j(x^j))$$
$$= \frac{N-1}{N}(\mu) + \frac{1}{N}\sum_{j\neq i}\mathbb{E}b^j(x^j),$$

since $\mathbb{E}(\xi^j) = 0$ and $\xi^j$ is independent from all other variables. In addition,

$$\mathbb{E}(Y_2) = \mathbb{E}\left(\frac{1}{N}\bar{x}^i - \mu\right) = \frac{1}{N}\mathbb{E}(\bar{x}^i) - \mu = \frac{(1-N)}{N}\mu$$

Next,

$$\mathbb{E}\left(\|Y_1\|^2\right) = \mathbb{E}\left(\|\frac{1}{N}\sum_{j\neq i}\left(\bar{x}^j + \alpha^j(x^j)\xi^j + b^j(x^j)\right)\|^2\right)$$

$$= \mathbb{E}\left(\|\frac{1}{N}\sum_{j\neq i}\bar{x}^j + b^j(x^j)\|^2 + \|\frac{1}{N}\sum_{j\neq i}\alpha^j(x^j)\xi^j\|^2\right)$$

$$= \frac{1}{N^2}\mathbb{E}\left(\|\sum_{j\neq i}\bar{x}^j + \sum_{j\neq i}b^j(x^j)\|^2\right) + \frac{1}{N^2}\sum_{j\neq i}\mathbb{E}(\alpha^j(x^j)^2)$$

$$= \frac{1}{N^2}\mathbb{E}\left(\|\sum_{j\neq i}(\bar{x}^j - \mu + b^j(x^j)) + (N-1)\mu\|^2\right) + \frac{1}{N^2}\sum_{j\neq i}\mathbb{E}(\alpha^j(x^j)^2)$$

$$= \frac{1}{N^2}\mathbb{E}\left(\|\sum_{j\neq i}\bar{x}^j - \mu\|^2 + \|\sum_{j\neq i}b^j(x^j) + (N-1)\mu\|^2\right) + \frac{1}{N^2}\sum_{j\neq i}\mathbb{E}(\alpha^j(x^j)^2)$$

$$= \frac{1}{N^2}\mathbb{E}\left(\|\sum_{j\neq i}\sum_k(\frac{1}{n}x_k^j - \mu)\|^2\right) + \frac{1}{N^2}\mathbb{E}\|(N-1)\mu + \sum_{j\neq i}b^j(x^j)\|^2 + \frac{1}{N^2}\sum_{j\neq i}\mathbb{E}(\alpha^j(x^j)^2)$$

$$= \frac{1}{N^2}\mathbb{E}\left(\|\frac{1}{n}(\sum_{j\neq i}\sum_k x_k^j - \mu)\|^2\right) + \frac{1}{N^2}\mathbb{E}\|(N-1)\mu + \sum_{j\neq i}b^j(x^j)\|^2 + \frac{1}{N^2}\sum_{j\neq i}\mathbb{E}(\alpha^j(x^j)^2)$$

$$= \frac{1}{N^2}\mathbb{E}\left(\sum_{j\neq i}\|\frac{1}{n}(\sum_k x_k^j - \mu)\|^2\right) + \frac{1}{N^2}\mathbb{E}\|(N-1)\mu + \sum_{j\neq i}b^j(x^j)\|^2 + \frac{1}{N^2}\sum_{j\neq i}\mathbb{E}(\alpha^j(x^j)^2)$$

$$= \frac{(N-1)}{N^2}\left(\frac{\sigma^2}{n}+\sigma_\star^2\right) + \frac{1}{N^2}\mathbb{E}\|(N-1)\mu + \sum_{j\neq i} b^j(x^j)\|^2 + \frac{1}{N^2}\sum_{j\neq i}\mathbb{E}(\alpha^j(x^j)^2)$$

Where the first squared norms factors because

$$\mathbb{E}[< \frac{1}{N}\sum_{j\neq i}\bar{x}^j + b^j(x^j), \alpha^j(x^j)\xi^j >] = \mathbb{E}[< \alpha^j(x^j)\frac{1}{N}\sum_{j\neq i}(\bar{x}^j + b^j(x^j)), \xi^j >] = 0$$

as the $\xi_j$ are independent of all other variables and have zero mean, the $\mathbb{E}(\|\sum_{j\neq i}\alpha^j(x^j)\xi^j\|^2)$ term factors because all $\alpha^j(x^j)$ and $\xi_j$ are independent (and the $\xi_j$ have zero mean), the $b^j(x^j)$ terms factor because $\mathbb{E} < \bar{x}^j - \mu, b^j(x^j) >= 0$ by assumption and because the $\bar{x}^j - \mu$ have zero mean and are independent of $b^i(x^i)$ for $j \neq i$. The last squared norm factors because all $\sum_k x_j^k - \mu$ terms are independent and have zero mean.

Finally,

$$\mathbb{E}\left(\|Y_2\|^2\right) = \mathbb{E}\left(\|\frac{1}{N}\bar{x}^i - \mu\|^2\right)$$

$$= \mathbb{E}\left(\|\frac{1}{Nn}\sum_j x_j^i - \mu\|^2\right)$$

$$= \mathbb{E}\left(\|\frac{1}{Nn}\sum_j (x_j^i - \mu) + \frac{1-N}{N}\mu\|^2\right)$$

$$= \mathbb{E}\left(\|\frac{1}{Nn}\sum_j (x_j^i - \mu)\|^2 + \|\frac{1-N}{N}\mu\|^2\right)$$

$$= \frac{1}{N^2}\mathbb{E}\left(\|\frac{1}{n}\sum_j (x_j^i - \mu)\|^2\right) + \frac{(1-N)^2}{N^2}\|\mu\|^2$$

$$= \frac{1}{N^2}(\frac{\sigma}{n}+\sigma_\star^2) + \frac{(1-N)^2}{N^2}\|\mu\|^2$$

$$= \frac{1}{N^2 n}\sigma^2 + \frac{1}{N^2}\sigma_\star^2 + \frac{(1-N)^2}{N^2}\|\mu\|^2$$

with the sums factoring for the same reasons as above. Substituting in (13):

$$\mathbb{E}\left(\|Y^2\|\right) = \mathbb{E}\left(\|Y_1\|^2\right) + \mathbb{E}\left(\|Y_2\|^2\right) + 2\mathbb{E}(Y_1)^t\mathbb{E}(Y_2)$$

$$= \frac{(N-1)}{N^2}(\frac{\sigma^2}{n}+\sigma_\star^2) + \frac{1}{N^2}\mathbb{E}\|(N-1)\mu + \sum_{j\neq i} b^j(x^j)\|^2 + \frac{1}{N^2}\sum_{j\neq i}\mathbb{E}(\alpha^j(x^j)^2)$$

$$+ \frac{1}{N^2 n}\sigma^2 + \frac{1}{N^2}\sigma_\star^2 + \frac{(1-N)^2}{N^2}\|\mu\|^2 + 2(\frac{N-1}{N}\mu + \frac{1}{N}\sum_{j\neq i}\mathbb{E}b^j(x^j))^t\frac{(1-N)}{N}\mu$$

$$= \frac{\sigma^2}{Nn} + \frac{\sigma_\star^2}{N} + \frac{1}{N^2}\sum_{j\neq i}\mathbb{E}(\alpha^j(x^j)^2)$$

$$+ \frac{(N-1)^2}{N^2}\|\mu\|^2 + \frac{1}{N^2}\mathbb{E}\|\sum_{j\neq i} b^j(x^j)\|^2 + \frac{2(N-1)}{N^2}\sum_{j\neq i}\mathbb{E}(b^j(x^j)^t\mu)$$

$$+ \frac{(N-1)^2}{N^2}\|\mu\|^2 + 2\frac{(N-1)(1-N)}{N^2}\|\mu\|^2 + \frac{2(1-N)}{N^2}\sum_{j\neq i}\mathbb{E}b^j(x^j)^t\mu$$

$$= \frac{\sigma^2}{Nn} + \frac{\sigma_\star^2}{N} + \frac{1}{N^2}\sum_{j\neq i}\mathbb{E}(\alpha^j(x^j)^2) + \frac{1}{N^2}\mathbb{E}\|\sum_{j\neq i} b^j(x^j)\|^2 \qquad (14)$$

Substituting (11), (12) and (14) into (10), we have:

$$
\mathbb{E}\left(\|\theta^i - \mu\|^2\right) = \left(1 - \beta^i\right)^2 \mathbb{E}\left(\|Y\|^2\right) + (\beta^i)^2 \mathbb{E}\left(\|Z\|^2\right) + 2\left(1 - \beta^i\right)\beta^i \mathbb{E}\left(Z^t Y\right)
$$

$$
= \left(1 - \beta^i\right)^2 \left( \frac{\sigma^2}{Nn} + \frac{\sigma_\star^2}{N} + \frac{1}{N^2}\sum_{j \neq i} \mathbb{E}(\alpha^j(x^j)^2) + \frac{1}{N^2}\mathbb{E}\|\sum_{j \neq i} b^j(x^j)\|^2 \right)
$$

$$
+ (\beta^i)^2 \left(\frac{\sigma^2}{n} + \sigma_\star^2\right) + 2\left(1 - \beta^i\right)\beta^i\left(\frac{\sigma^2}{Nn} + \frac{\sigma_\star^2}{N}\right)
$$

$\square$

Next we prove a lemma that gives the optimal value of the defense parameter $\beta$ of a player, assuming that the attack parameters $\alpha$ of all players are fixed.

**Lemma C.1.** *For a fixed set of values $\alpha_1, \ldots, \alpha_N \in [0, \infty)$ and $b^1, \ldots, b^N \in \mathbb{R}^d$, the value of $\beta^i$ that minimizes the mean squared error of the estimate of player $i$ is given by:*

$$
(\beta^i)^* = \frac{\frac{1}{N^2}\sum_{j \neq i}\mathbb{E}(\alpha^j(x^j)^2) + \frac{1}{N^2}\mathbb{E}\|\sum_{j \neq i} b^j(x^j)\|^2}{\left(\frac{\sigma^2}{n} + \sigma_\star^2 - \frac{\sigma^2}{Nn} - \frac{\sigma_\star^2}{N} + \frac{1}{N^2}\sum_{j \neq i}\mathbb{E}(\alpha^j(x^j)^2) + \frac{1}{N^2}\mathbb{E}\|\sum_{j \neq i} b^j(x^j)\|^2\right)} \tag{15}
$$

*Proof.* Re-writing the statement of Theorem C.1,

$$
\mathbb{E}\left(\|\theta^i - \mu\|^2\right) = (\beta^i)^2 \left( \frac{\sigma^2}{n} + \sigma_\star^2 - \frac{\sigma^2}{Nn} - \frac{\sigma_\star^2}{N} + \frac{1}{N^2}\sum_{j \neq i}\mathbb{E}(\alpha^j(x^j)^2) + \frac{1}{N^2}\mathbb{E}\|\sum_{j \neq i} b^j(x^j)\|^2 \right)
$$

$$
- 2\frac{\beta^i}{N^2}\left( \sum_{j \neq i}\mathbb{E}(\alpha^j(x^j)^2) + \mathbb{E}\|\sum_{j \neq i} b^j(x^j)\|^2 \right)
$$

$$
+ \left( \frac{\sigma^2}{Nn} + \frac{\sigma_\star^2}{N} + \frac{1}{N^2}\sum_{j \neq i}\mathbb{E}(\alpha^j(x^j)^2) + \frac{1}{N^2}\mathbb{E}\|\sum_{j \neq i} b^j(x^j)\|^2 \right)
$$

This is a quadratic function of $\beta^i$ with a positive coefficient in front of the square term. Therefore, the function is minimized over $(-\infty, \infty)$ at the point:

$$
(\beta^i)^* = \frac{\frac{1}{N^2}\sum_{j \neq i}\mathbb{E}(\alpha^j(x^j)^2) + \frac{1}{N^2}\mathbb{E}\|\sum_{j \neq i} b^j(x^j)\|^2}{\left(\frac{\sigma^2}{n} + \sigma_\star^2 - \frac{\sigma^2}{Nn} - \frac{\sigma_\star^2}{N} + \frac{1}{N^2}\sum_{j \neq i}\mathbb{E}(\alpha^j(x^j)^2) + \frac{1}{N^2}\mathbb{E}\|\sum_{j \neq i} b^j(x^j)\|^2\right)}
$$

$\square$

This result is closely related to the work of Grimberg et al. (2021), which studies the optimal way of averaging two sample sets from two different distributions, with the goal of minimizing the mean squared error of the estimate on one of these distributions. However, in our case the estimate from the server is an average of manipulated samples coming from *multiple distributions*, rather than i.i.d. samples from a single one. Now, C.1 allows us to prove Corollary 4.2:

**Corollary C.2.** *The game defined by the expected reward*

$$
\mathbb{E}\left(\mathcal{R}^i(\theta^1, \ldots, \theta^N, \mu)\right) = \frac{\sum_{j \neq i}\mathbb{E}\left(\|\theta^j - \mu\|^2\right)}{N - 1} - \lambda_i \mathbb{E}\left(\|\theta^i - \mu\|^2\right)
$$

*and the set of strategies $\mathscr{A} \times \mathscr{D}$ does not have a (pure or mixed) Nash equilibrium at which $\mathbb{E}((\alpha^j)^2)$ and $\mathbb{E}\|b^j(x^j)\|^2$ are finite for all players.*

*Proof.* Assume that a strategy profile $((\alpha^1, b^1, \beta^1), \ldots, (\alpha^N, b^N, \beta^N)) \in \mathcal{P}\left(\mathscr{A} \times \mathscr{D}\right)^N$ is a (potentially mixed) Nash equilibrium for which $\mathbb{E}((\alpha^j)^2)$ and $\mathbb{E}\|b^j(x^j)\|^2$ are finite for player $j \neq i$

(with the expectations taken both over the randomness of the strategy profile and $x$). Note that in the definition of the (expected) reward function, the value of the defense parameters of the $i$-th player $\beta^i$ only affects the MSE of the estimate $\theta_i$ of that player. Therefore, it follows from Lemma C.1 that each $\beta^i$ must be non-random and defined by equation (15). In particular, it is easy to see that each $\beta^i \in [0, 1)$. In that case, it follows from Theorem C.1 that the expected reward of each player $j$ is strictly monotonically increasing in $\mathbb{E}(\alpha^j(x^j)^2)$, so that player $j$ can increase their reward by increasing $\mathbb{E}((\alpha^j)^2)$. This contradicts the assumption that the strategy profile is a Nash equilibrium. $\square$

Finally, we prove corollary B.1. For this, we consider the same game as before, in the case when an upper bound A on the parameters $\alpha^i$ is given. Recall that we denote the resulting set of attack strategies by $\mathscr{A}_A$. Since $\mathscr{A}_A \subset \mathscr{A}^m$, Theorem 4.1 holds for the joint set of strategies $(\mathscr{A}_A^m \times \mathscr{D})$. Then we have the following

**Corollary C.3.** *In the setup of Theorem 4.1, if the set of available strategies is $\mathscr{A}_A \times \mathscr{D}$ for some constant $A > 0$, the only Nash equilibria of the game with $b^i(x^i) = 0$ fixed for all players $i$ are the strategy profiles for which:*

$$|\alpha^i(x^i)| = A \quad \text{and} \quad \beta^i = \frac{A^2}{(\frac{\sigma^2}{n} + \sigma_\star^2)N + A^2} \quad \forall i \in [N]. \tag{16}$$

*Furthermore, at each of these equilibria the value of mean squared error of the estimate of each player $i$ is*

$$\mathbb{E}\left(\|\theta^i - \mu\|^2\right) = (\frac{\sigma^2}{n} + \sigma_\star^2)\frac{(1 + \frac{1}{\frac{\sigma^2}{n} + \sigma_\star^2}A^2)}{(N + \frac{1}{\frac{\sigma^2}{n} + \sigma_\star^2}A^2)}$$

*Proof.* Assume that a strategy profile $((\alpha^1, b^1, \beta^1), \ldots, (\alpha^N, b^N, \beta^N))$ is a Nash equilibrium. As in Corollary C.2, it follows that each $\beta^i$ is given by equation (15) and is therefore in the interval $[0, 1)$. Therefore, the reward of each player $i$ is increasing with $\mathbb{E}(\alpha^i(x^i)^2)$. It follows that $\alpha^i(x^i) = A$ for all $i \in [N]$. Substituting for the value of $(\beta^i)^*$ we get that for every $i \in [N]$:

$$\beta^i = (\beta^i)^* = \frac{\frac{N-1}{N^2}A^2}{\frac{\sigma^2}{n} + \sigma_\star^2 - \frac{\sigma_\star^2}{Nn} - \frac{\sigma_\star^2}{N} + \frac{N-1}{N^2}A^2} = \frac{A^2}{(\frac{\sigma^2}{n} + \sigma_\star^2)N + A^2}.$$

Substituting into Theorem C.1 and setting $\hat{\sigma}^2 = \sigma^2 + n\sigma_\star^2$ we get:

$$\mathbb{E}\left(\|\theta^i - \mu\|^2\right) = \left(1 - \beta^i\right)^2 \left(\frac{\hat{\sigma}^2}{Nn} + \frac{1}{N^2}\sum_{j\neq i}(\alpha^j)^2\right) + (\beta^i)^2\frac{\hat{\sigma}^2}{n} + 2\left(1 - \beta^i\right)\beta^i\frac{\hat{\sigma}^2}{Nn}$$

$$= \frac{\frac{N^2\hat{\sigma}^4}{n^2}}{(\frac{N\hat{\sigma}^2}{n} + A^2)^2}\left(\frac{\hat{\sigma}^2}{Nn} + \frac{N-1}{N^2}A^2\right) + \frac{A^4}{(\frac{N\hat{\sigma}^2}{n} + A^2)^2}\frac{\hat{\sigma}^2}{n} + 2\frac{\frac{A^2N\hat{\sigma}^2}{n}}{(\frac{N\hat{\sigma}^2}{n} + A^2)^2}\frac{\hat{\sigma}^2}{Nn}$$

$$= \frac{\frac{N\hat{\sigma}^6}{n^3} + \frac{\hat{\sigma}^4}{n^2}(N-1)A^2 + \frac{A^4\hat{\sigma}^2}{n} + 2\frac{A^2\hat{\sigma}^4}{n^2}}{(\frac{N\hat{\sigma}^2}{n} + A^2)^2}$$

$$= \frac{\hat{\sigma}^2}{n}\frac{\frac{N\hat{\sigma}^4}{n^2} + \frac{\hat{\sigma}^2}{n}(N-1)A^2 + A^4 + 2\frac{\hat{\sigma}^2}{n}A^2}{(\frac{N\hat{\sigma}^2}{n} + A^2)^2}$$

$$= \frac{\hat{\sigma}^2}{n}\frac{\frac{\hat{\sigma}^4}{n^2}N + \frac{\hat{\sigma}^2}{n}NA^2 + A^4 + \frac{\hat{\sigma}^2}{n}A^2}{(\frac{\hat{\sigma}^2}{n}N + A^2)^2}$$

$$= \frac{\hat{\sigma}^2}{n}\frac{N + \frac{n}{\hat{\sigma}^2}NA^2 + \frac{n^2}{\hat{\sigma}^4}A^4 + \frac{n}{\hat{\sigma}^2}A^2}{(N + \frac{n}{\hat{\sigma}^2}A^2)^2}$$

$$= \frac{\hat{\sigma}^2}{n}\frac{(1 + \frac{n}{\hat{\sigma}^2}A^2)(N + \frac{n}{\hat{\sigma}^2}A^2)}{(N + \frac{n}{\hat{\sigma}^2}A^2)^2}$$

$$= \frac{\hat{\sigma}^2}{n}\frac{(1 + \frac{n}{\hat{\sigma}^2}A^2)}{(N + \frac{n}{\hat{\sigma}^2}A^2)}$$

$$= (\frac{\sigma^2}{n} + \sigma_\star^2) \frac{(1 + \frac{1}{\frac{\sigma^2}{n} + \sigma_\star^2} A^2)}{(N + \frac{1}{\frac{\sigma^2}{n} + \sigma_\star^2} A^2)}$$

$\square$

# D   Proofs on mechanisms for mean estimation

Next, we proof a version of Theorem 5.1 without redistribution.

**Proposition D.1.** *In the setting of 4.1, the penalized game with rewards*

$$\mathcal{R}_p^i = \frac{\sum_{j \neq i} \|\theta^j - \mu\|^2}{N - 1} - \lambda_i \|\theta^i - \mu\|^2 - C\|m^i - \theta^s\|^2$$

*has a Nash equilbrium consisting of the strategies $\alpha^j = b^j = \beta^j = 0$ for all $j$ whenever $C > \frac{1}{(N-1)^2}$.*

*At this equilibrium, the expected penalty $p^i(m^1, \ldots, m^N)$ paid by each player $i$ is equal to $C\frac{(N-1)}{Nn}\sigma^2$, thus a player is incentivized to participate in the penalized game rather than relying on their own estimate, whenever $N > 2$, the other $N - 1$ players participate at the honest equilibrium and $\lambda_i > C + \frac{N}{(N-1)^2}$ or $N = 2$ and $\lambda_i > C + 1$.*

*Proof.* We begin by inserting the equality $\mathbb{E}\left(\|\theta^i - \mu\|^2\right) = \left(1 - \beta^i\right)^2 \left(\frac{\sigma^2}{Nn} + \frac{\sigma_\star^2}{N} + \frac{1}{N^2}\sum_{j \neq i}\mathbb{E}(\alpha^j(x^j)^2) + \frac{1}{N^2}\mathbb{E}\|\sum_{j \neq i} b^j(x^j)\|^2\right) + (\beta^i)^2(\frac{\sigma^2}{n} + \sigma_\star^2) + 2\left(1 - \beta^i\right)\beta^i(\frac{\sigma^2}{Nn} + \frac{\sigma_\star^2}{N})$ from 4.1 in the first two terms to obtain

$$\mathbb{E}\mathcal{R}_p^i = \mathbb{E}\left(\frac{\sum_{j \neq i}\|\theta^j - \mu\|^2}{N - 1} - \lambda_i\|\theta^i - \mu\|^2 - C\|m^i - \frac{1}{N-1}\sum_{j \neq i} m^j\|^2\right)$$

$$= \mathbb{E}\frac{\sum_{j \neq i}\left(1 - \beta^j\right)^2\left(\frac{\sigma^2}{Nn} + \frac{\sigma_\star^2}{N} + \frac{1}{N^2}\sum_{k \neq j}\mathbb{E}(\alpha^k(x^k)^2) + \frac{1}{N^2}\mathbb{E}\|\sum_{k \neq j} b^k(x^k)\|^2\right)}{N - 1}$$

$$+ \mathbb{E}\frac{\sum_{j \neq i}\left((\beta^j)^2(\frac{\sigma^2}{n} + \sigma_\star^2) + 2\left(1 - \beta^j\right)\beta^j(\frac{\sigma^2}{Nn} + \frac{\sigma_\star^2}{N})\right)}{N - 1}$$

$$- \lambda_i\mathbb{E}\left(1 - \beta^i\right)^2\left(\frac{\sigma^2}{Nn} + \frac{\sigma_\star^2}{N} + \frac{1}{N^2}\sum_{j \neq i}\mathbb{E}(\alpha^j(x^j)^2) + \frac{1}{N^2}\mathbb{E}\|\sum_{j \neq i} b^j(x^j)\|^2\right)$$

$$- \lambda_i\left((\beta^i)^2(\frac{\sigma^2}{n} + \sigma_\star^2) + 2\left(1 - \beta^i\right)\beta^i(\frac{\sigma^2}{Nn} + \frac{\sigma_\star^2}{N})\right)$$

$$- \mathbb{E}\left(C\|m^i - \theta^s\|^2\right)$$

Correspondingly, we get

$$\frac{d}{d\mathbb{E}\alpha^j(x^j)^2}\mathbb{E}\mathcal{R}_p^i = \sum_{j \neq i}\frac{(1 - \beta^i)^2}{(N-1)N^2} - C\frac{d}{d\alpha^j(x^j)^2}\mathbb{E}\|m^i - \theta^s\|^2 \tag{17}$$

To analyze the second term, we calculate:

$$\mathbb{E}\|m^i - \theta^s\|^2 = \|m^i - \theta^s\|^2$$

$$= \mathbb{E}\|m^i - \frac{1}{N}\sum_j m^j\|^2$$

$$= \mathbb{E}\|\alpha^i(x^i)\xi^i + b^i(x^i) + \frac{1}{n}\sum_k x_k^i - \frac{1}{N}\sum_j(\alpha^j(x^j)\xi^j + b^j(x^j) + \frac{1}{n}\sum_k x_k^j)\|^2$$

$$= \mathbb{E}\|\alpha^i(x^i)\xi^i + \frac{1}{n}(\sum_k x_k^i - \mu)$$

$$-\frac{1}{N}\sum_j \left(\alpha^j(x^j)\xi^j + \frac{1}{n}\sum_k(x_k^j - \mu)\right) + \mu - \mu + d_b^i\|^2$$

$$= \mathbb{E}\|\frac{N-1}{N}\left(\alpha^i(x^i)\xi^i + \frac{1}{n}(\sum_k x_k^i - \mu)\right)$$

$$-\frac{1}{N}\sum_{j\neq i}\left(\alpha^j(x^j)\xi^j + \frac{1}{n}\sum_k(x_k^j - \mu)\right) + d_b^i\|^2$$

$$= (\frac{N-1}{N})^2\mathbb{E}\|\alpha^i(x^i)\xi^i\|^2 + (\frac{N-1}{N})^2\mathbb{E}\|\frac{1}{n}\sum_k x_k^i - \mu\|^2$$

$$+ \frac{1}{N^2}\sum_{j\neq i}\mathbb{E}\|\alpha^j(x^j)\xi^j\|^2 + \frac{1}{N^2}\sum_{j\neq i}\mathbb{E}\|\frac{1}{n}\sum_k x_k^j - \mu\|^2 + \mathbb{E}\|d_b^i\|^2$$

$$= (\frac{N-1}{N})^2\mathbb{E}(\alpha^i(x^i)^2) + \frac{(N-1)^2}{N^2}(\frac{\sigma^2}{n} + \sigma_\star^2)$$

$$+ \frac{1}{N^2}\sum_{j\neq i}\mathbb{E}(\alpha^j(x^j)^2) + \frac{(N-1)}{N^2}(\frac{\sigma^2}{n} + \sigma_\star^2) + \mathbb{E}\|d_b^i\|^2$$

$$= (\frac{N-1}{N})^2\mathbb{E}(\alpha^i(x^i)^2) + \frac{1}{N^2}\sum_{j\neq i}\mathbb{E}(\alpha^j(x^j)^2)$$

$$+ \frac{(N-1)}{Nn}\sigma^2 + \frac{N-1}{N}\sigma_\star^2 + \mathbb{E}\|d_b^i\|^2 \tag{18}$$

setting $d_b^i = \frac{N-1}{N}b^i(x^i) - \sum_{j\neq i}\frac{b^j(x^j)}{N}$. The squared norm again factors because of the independence and zero means of both $\xi^i$ and $x_k^i - \mu$ and because $\mathbb{E} < \bar{x}^j - \mu, b^j(x^j) >= 0$, while $b^j(x^j)$ is independent of $\bar{x}^i - \mu$ for $i \neq j$. Now, inserting 18 in 17, we obtain

$$\frac{d}{d\mathbb{E}\alpha^i(x^i)^2}\mathbb{E}\mathcal{R}_p^i(\theta^1,\ldots,\theta^N,\mu) = \sum_{j\neq i}\frac{(1-\beta^i)^2}{(N-1)N^2} - C(\frac{N-1}{N})^2$$

$$\leq \frac{1}{N^2} - C\frac{(N-1)^2}{N^2}$$

$$= \frac{1}{N^2}\left((1 - C(N-1)^2)\right) \tag{19}$$

As $\beta^i \in [0,1]$. Thus $\mathbb{E}\frac{d}{d\alpha^i}\mathcal{R}_p^i$ is negative whenever $C(N-1)^2 > 1$ or $C > \frac{1}{(N-1)^2}$. Correspondingly, for such $C$, player $i$ is incentivized to set $\alpha^i = 0$, independent of other players' strategies.

Now, assuming $b^j(x^j) = 0$ for all other players $j \neq i$, we also get

$$\frac{d}{d\mathbb{E}\|b^i(x^i)\|^2}\mathbb{E}\mathcal{R}_p^i(\theta^1,\ldots,\theta^N,\mu) = \sum_{j\neq i}\frac{(1-\beta^i)^2}{(N-1)N^2} - C(\frac{N-1}{N})^2$$

$$\leq \frac{1}{N^2} - C\frac{(N-1)^2}{N^2}$$

$$= \frac{1}{N^2}\left((1 - C(N-1)^2)\right). \tag{20}$$

Correspondingly, $b^i = 0$ for all players is a Nash Equilbrium. As the penalty $p^i(m^1,...,m^N)$ does not dependend on the defense strategies $\beta^i$, the optimal $\beta^i$ at the equilibrium $\alpha^i = 0$ and $b^i = 0$ can still be calculated using 15 and is equal to zero as well.

The average penalty honest players pay at the Nash equilibrium is then given by $C\frac{(N-1)}{N}(\frac{\sigma^2}{n} + \sigma_\star^2)$, according to 18.

To understand participation incentives, we compare the equilibrium reward $\mathcal{R}^i(\theta^1,\ldots,\theta^N,\mu)$ player $i$ receives if they do not participate while all other players do, to the penalized reward $\mathcal{R}_p^i$ player $i$

would obtain when participating. We first calculate $\mathcal{R}^i(\theta^1,\dots,\theta^N,\mu)$ assuming player $i$ only uses their own estimate and does not send an update to the server, while the other $N-1$ players participate in the penalized game at equilbrium:

$$\mathbb{E}\mathcal{R}^i(\theta^1(N-1),\dots,\theta^i(1),\dots,\theta^N(N-1),\mu) = \mathbb{E}\left(\frac{\sum_{j\neq i}\|\theta^j(N-1)-\mu\|^2}{N-1} - \lambda\|\theta^i(1)-\mu\|^2\right)$$

$$= \frac{\sigma^2}{(N-1)n} + \frac{\sigma_\star^2}{(N-1)} - \lambda_i(\frac{\sigma^2}{n} + \sigma_\star^2) \qquad (21)$$

using 4.1 with $\alpha^j = b^j = \beta^j = 0$ and substituting $N-1$ and $1$ for $N$ respectively. We then calculate $\mathbb{E}\mathcal{R}_p^i$:

$$\mathbb{E}\mathcal{R}_p^i = \mathbb{E}\left(\frac{\sum_{j\neq i}\|\theta^j-\mu\|^2}{N-1} - \lambda_i\|\theta^i-\mu\|^2 - C\|m^i-\theta^s\|^2\right)$$

$$= (1-\lambda_i)(\frac{\sigma^2}{Nn} + \frac{\sigma_\star^2}{N}) - C(\frac{(N-1)}{Nn}\sigma^2 + \frac{(N-1)}{N}\sigma_\star^2) \qquad (22)$$

using 4.1 with $\alpha^j = b^j = \beta^j = 0$ and D.1. The difference between these can the be calculated as

$$\mathbb{E}\mathcal{R}_p^i - \mathbb{E}\mathcal{R}^i(\theta^1(N-1),\dots,\theta^i(1),\dots,\theta^N(N-1),\mu)$$

$$= (1-\lambda_i)(\frac{\sigma^2}{Nn} + \frac{\sigma_\star^2}{N}) - C\frac{(N-1)}{N}(\frac{\sigma^2}{n} + \sigma_\star^2) - \frac{1}{N-1}(\frac{\sigma^2}{n} + \sigma_\star^2) + \lambda_i(\frac{\sigma^2}{n} + \sigma_\star^2)$$

$$= (\frac{\sigma^2}{n} + \sigma_\star^2)\left((1-\lambda_i)\frac{1}{N} - C\frac{N-1}{N} - \frac{1}{N-1} + \lambda_i\right)$$

$$= (\frac{\sigma^2}{n} + \sigma_\star^2)\left(\frac{1}{N} - C\frac{N-1}{N} - \frac{1}{N-1} + \frac{N-1}{N}\lambda_i\right)$$

$$= (\frac{\sigma^2}{n} + \sigma_\star^2)\left(\frac{1}{N} - \frac{1}{N-1} + \frac{N-1}{N}(\lambda_i - C)\right)$$

$$\geq (\frac{\sigma^2}{n} + \sigma_\star^2)\left(-\frac{1}{N-1} + \frac{N-1}{N}(\lambda_i - C)\right)$$

$$= (\frac{\sigma^2}{n} + \sigma_\star^2)\left(\frac{N-1}{N}(\lambda_i - C - \frac{N}{(N-1)^2})\right)$$

and is positive whenever $\lambda_i > C + \frac{N}{(N-1)^2}$, such that in these cases player $i$ is better off participating in data sharing, despite the penalties. If $N=2$, we instead obtain

$$\mathbb{E}\mathcal{R}_p^i - \mathbb{E}\mathcal{R}^i(\theta^1(N-1),\dots,\theta^i(1),\dots,\theta^N(N-1),\mu) = \frac{\sigma^2}{n}\left(-\frac{1}{2} + \frac{1}{2}(\lambda_i - C)\right)$$

which is positive whenever $\lambda_i > C + 1$. $\qquad\square$

We now prove 5.1:

**Theorem D.1.** *In the setting of 4.1, the penalized game with rewards*

$$\mathcal{R}_{p'}^i = \frac{\sum_{j\neq i}\|\theta^j-\mu\|^2}{N-1} - \lambda_i\|\theta^i-\mu\|^2 - C\|m^i-\theta^s\|^2 + \frac{1}{N-1}\sum_{j\neq i}C\|m^j-\bar{m}\|^2$$

*has a Nash equilibrium consisting of the strategies $\alpha^j = b^j = \beta^j = 0$ for all $j$ whenever $C > \frac{1}{(N-1)^2-1}$. Furthermore, this equilibrium maximizes the sum of all players' rewards among equilibria whenever $\lambda_i \geq 1$ for all players.*

*At this equilibrium, the expected penalty $p^i(m^1,\dots,m^N)$ paid by each player $i$ is equal to $0$, such that player $i$ is incentivized to participate in the penalized game rather than relying on their own estimate, whenever $N > 2$, the other $N-1$ players participate at the honest equilibrium, and $\lambda_i > \frac{N}{(N-1)^2}$.*

*Proof.* Analogous to the proof of D.1 We have that

$$\frac{d}{d\mathbb{E}\alpha^i(x^i)^2}\mathbb{E}\mathcal{R}^i_{p'} = \frac{d}{d\mathbb{E}\alpha^i(x^i)^2}\mathbb{E}\mathcal{R}^i_p + \frac{1}{N-1}\sum_{j\neq i}C\frac{d}{d\mathbb{E}\alpha^i(x^i)^2}\mathbb{E}\|m^j - \theta^s\|^2$$

$$\leq \frac{1}{N^2}\left((1 - C(N-1)^2)\right) + C\frac{1}{N^2}$$

$$= \frac{1}{N^2}\left((1 - C\left((N-1)^2 - 1\right))\right)$$

by inserting 18 and 19.

Similarly, assuming $b^j(x^j) = 0$ for all players $j \neq i$ we get

$$\frac{d}{d\mathbb{E}\|b^i(x^i)\|^2}\mathbb{E}\mathcal{R}^i_{p'} = \frac{d}{d\mathbb{E}\|b^i(x^i)\|^2}\mathbb{E}\mathcal{R}^i_p + \frac{1}{N-1}\sum_{j\neq i}C\frac{d}{d\mathbb{E}\|b^i(x^i)\|^2}\mathbb{E}\|m^j - \theta^s\|^2$$

$$\leq \frac{1}{N^2}\left((1 - C(N-1)^2)\right) + C\frac{1}{N^2}$$

$$= \frac{1}{N^2}\left((1 - C\left((N-1)^2 - 1\right))\right)$$

by inserting 18 and 20

Thus $\mathbb{E}\frac{d}{d\mathbb{E}\alpha^i(x^i)^2}\mathcal{R}^i_{p'}$ is negative whenever $C((N-1)^2 - 1) > 1$ or $C > \frac{1}{(N-1)^2-1}$ and $\frac{d}{d\mathbb{E}\|b^i(x^i)\|^2}\mathbb{E}\mathcal{R}^i_{p'}$ under the same conditions as long as $b^j(x^j) = 0$ for all other players $j$.

The expected penalty paid by each player is zero by symmetry, as every players' paid penalty gets redistributed equally among all other players, such that the payments cancel out in expectation.

Similarly, the calculations for participation incentives are exactly as in D.1, but $\mathbb{E}\mathcal{R}^i_{p'}$ is now equal to $(1 - \lambda_i)(\frac{\sigma^2}{Nn} + \frac{\sigma^2_\star}{N})$ rather than $(1 - \lambda_i)(\frac{\sigma^2}{Nn} + \frac{\sigma^2_\star}{N}) - C(\frac{(N-1)}{Nn}\sigma^2 + \frac{(N-1)}{N}\sigma^2_\star)$ because the expected penalty paid by players is equal to zero at equilbrium.

Lastly, it is easy to see that $\mathbb{E}\|b^i(x^i)\|^2$ has to be nonzero for at least two players at any other equilibrium. Compared to the honest equilibrium, that increases every players' MSE. But the (unpenalized) expected reward for player $i$ equals $\frac{\sum_{j\neq i}\|\theta^j - \mu\|^2}{N-1} - \lambda_i\|\theta^i - \mu\|^2$, such that the sum of all players' (unpenalized) rewards equals $\sum_j(1 - \lambda_j)\|\theta^j - \mu\|^2$, which is strictly monotonically decreasing in all players' MSEs when $\lambda_i \geq 1$ for all players. As all players' penalties add up to zero in expectation, the sum of penalized rewards is equally strictly monotonically decreasing in all players' MSEs.

$\square$

Next, we prove 5.2

**Theorem D.2.** *Consider the modified game with reward*

$$\mathcal{R}^i = \frac{\sum_{j\neq i}\|\theta^j - \mu\|^2}{N-1} - \lambda_i\|\theta^i - \mu\|^2,$$

*where player $i$ receives an estimate $\bar{m} + \sqrt{C}\epsilon^i\|m^i - \bar{m}\|$ for independent noise $\epsilon^i$ with mean $\mathbb{E}\epsilon^i = 0$ and "variance" $\mathbb{E}\|\epsilon^i\|^2 = 1$, instead of the empirical mean $\bar{m}$, from the server. Then honesty $(\alpha^i = 0, b^i = 0, \beta^i = \frac{C}{C+1})$ is a Nash equilibrium, as long as $C > \frac{1}{\lambda_i(N-1)^2-1}$ and $\lambda_i > \frac{1}{(N-1)^2}$. Furthermore, honesty maximizes the sum of all players' rewards among equilibria whenever $\lambda_i \geq 1$ for all players.*

*Furthermore, for fixed constant $\lambda_i = \lambda$, $\mathbb{E}\left(\|\theta^i - \mu\|^2\right) \in O\left(\frac{\sigma^2}{Nn} + \frac{\sigma^2_\star}{N}\right)$ whenever $C = \frac{k}{\lambda(N-1)^2-1}$ for any constant $k > 1$, such that players are incentivized to participate in the penalized game rather than relying on their own estimate, whenever $N > 2$, the other $N - 1$ players participate at the honest equilibrium, and $\lambda \geq 1$.*

*Proof.* In order to calculate the reward $\mathcal{R}^i(\theta^1, \ldots, \theta^N, \mu) = \frac{\sum_{j\neq i}\|\theta^j - \mu\|^2}{N-1} - \lambda_i\|\theta^i - \mu\|^2$ for player $i$, we have a closer look at the mean squared error incurred by players in the modified game. For

convenience, we set $\bar{\sigma}^2 := \frac{\sigma^2}{n} + \sigma_\star^2$

$$\mathbb{E}\|\theta^i - \mu\|^2 = \mathbb{E}\|(1 - \beta^i)\left(\bar{m}^i + \sqrt{C}\|m^i - \theta^s\|\epsilon^i\right) + \beta^i \bar{x}^i - \mu\|^2$$

$$= \mathbb{E}\|(1 - \beta^i)\bar{m}^i + \beta^i \bar{x}^i - \mu\|^2 + \mathbb{E}\|(1 - \beta^i)\sqrt{C}\|m^i - \theta^s\|\epsilon^i\|^2$$

$$+ 2\mathbb{E} < \left((1 - \beta^i)\sqrt{C}\|m^i - \theta^s\|\epsilon^i, \left((1 - \beta^i)\bar{m}^i + \beta^i \bar{x}^i - \mu\right)\right) >$$

$$= (1 - \beta^i)^2 \left(\frac{\bar{\sigma}^2}{N} + \frac{1}{N^2}\sum_{j \neq i}\mathbb{E}(\alpha^j(x^j)^2) + \frac{1}{N^2}\mathbb{E}\|\sum_{j \neq i} b^j(x^j)\|^2\right)$$

$$+ (\beta^i)^2\bar{\sigma}^2 + 2(1 - \beta^i)\beta^i \frac{\bar{\sigma}^2}{N}$$

$$+ (1 - \beta^i)^2 C \left((\frac{N-1}{N})^2\mathbb{E}(\alpha^i(x^i)^2) + \frac{1}{N^2}\sum_{j \neq i}\mathbb{E}(\alpha^j(x^j)^2) + \mathbb{E}\|d_b^i\|^2 + \frac{(N-1)}{N}\bar{\sigma}^2\right)$$

$$+ 2\mathbb{E} < \left((1 - \beta^i)\sqrt{C}\|m^i - \theta^s\|\epsilon^i, \left((1 - \beta^i)\bar{m}^i + \beta^i \bar{x}^i - \mu\right)\right) >$$

using the calculations from the proof of C.1 for the first term and D.1 for the second. The last term is equal to zero because:

$$\mathbb{E}\left(< (1 - \beta^i)\sqrt{C}\|m^i - \theta^s\|\epsilon^i, \left((1 - \beta^i)\bar{m}^i + \beta^i \bar{x}^i - \mu\right) >\right)$$

$$= \mathbb{E}\left(< \epsilon^i, (1 - \beta^i)\sqrt{C}\|m^i - \theta^s\| \left((1 - \beta^i)\bar{m}^i + \beta^i \bar{x}^i - \mu\right) >\right)$$

$$= \mathbb{E}(< \epsilon^i, \mathbb{E}\left((1 - \beta^i)\sqrt{C}\|m^i - \theta^s\| \left((1 - \beta^i)\bar{m}^i + \beta^i \bar{x}^i - \mu\right) >\right)$$

$$= 0$$

as $\epsilon^i$ is independent of all the other terms and has mean zero. Simplifying, we obtain

$$\mathbb{E}\|\theta^i - \mu\|^2$$

$$= (1 - \beta^i)^2 \left(\frac{\bar{\sigma}^2}{N} + \frac{1}{N^2}\sum_{j \neq i}\mathbb{E}(\alpha^j(x^j)^2) + \frac{1}{N^2}\mathbb{E}\|\sum_{j \neq i} b^j(x^j)\|^2\right) + (\beta^i)^2\bar{\sigma}^2 + 2(1 - \beta^i)\beta^i \frac{\bar{\sigma}^2}{N}$$

$$+ (1 - \beta^i)^2 C \left((\frac{N-1}{N})^2\mathbb{E}(\alpha^i(x^i)^2) + \mathbb{E}\|d_b^i\|^2 + \frac{1}{N^2}\sum_{j \neq i}\mathbb{E}(\alpha^j(x^j)^2) + \frac{(N-1)}{N}\bar{\sigma}^2\right)$$

$$= (\beta^i)^2 \left(\bar{\sigma}^2 + \frac{C(N-1)-1}{N}\bar{\sigma}^2 + \frac{1+C}{N^2}\sum_{j \neq i}\mathbb{E}(\alpha^j(x^j)^2) + \frac{1}{N^2}\mathbb{E}\|\sum_{j \neq i} b^j(x^j)\|^2\right.$$

$$\left. + C(\frac{N-1}{N})^2\mathbb{E}(\alpha^i(x^i)^2) + C\mathbb{E}\|d_b^i\|^2\right)$$

$$- 2\beta^i \left(\frac{1+C}{N^2}\sum_{j \neq i}\mathbb{E}(\alpha^j(x^j)^2) + \frac{1}{N^2}\mathbb{E}\|\sum_{j \neq i} b^j(x^j)\|^2\right.$$

$$\left. + C(\frac{N-1}{N})^2\mathbb{E}(\alpha^i(x^i)^2) + C\mathbb{E}\|d_b^i\|^2 + C\frac{(N-1)}{N}\bar{\sigma}^2\right)$$

$$+ \frac{1+C}{N^2}\sum_{j \neq i}\mathbb{E}(\alpha^j(x^j)^2) + \frac{1}{N^2}\mathbb{E}\|\sum_{j \neq i} b^j(x^j)\|^2 + \frac{1 - C + CN}{N}\bar{\sigma}^2$$

$$+ C(\frac{N-1}{N})^2\mathbb{E}(\alpha^i(x^i)^2 + C\mathbb{E}\|d_b^i\|^2)$$

As in C.1, this yields an optimal value for $\beta^i$ of

$$(\beta^i)^* = \frac{\left(H_i(\alpha, b) + C\frac{(N-1)}{N}\bar{\sigma}^2\right)}{\left(\bar{\sigma}^2 + \frac{C(N-1)-1}{N}\bar{\sigma}^2 + H_i(\alpha, b)\right)}$$

for $H_i(\alpha, b) = \frac{1+C}{N^2}\sum_{j\neq i}\mathbb{E}(\alpha^j(x^j)^2) + \frac{1}{N^2}\mathbb{E}\|\sum_{j\neq i}b^j(x^j)\|^2 + C(\frac{N-1}{N})^2\mathbb{E}(\alpha^i(x^i)^2) + C\mathbb{E}\|d_b^i\|^2$
Now, calculating the derivative of $\mathbb{E}\mathcal{R}^i(\theta^1, \ldots, \theta^N, \mu)$ with respect to $\mathbb{E}(\alpha^i(x^i)^2)$ yields:

$$\frac{d}{d\mathbb{E}(\alpha^i(x^i)^2)}\mathbb{E}\mathcal{R}^i(\theta^1, \ldots, \theta^N, \mu) = \frac{d}{d\mathbb{E}(\alpha^i(x^i)^2)}\frac{\sum_{j\neq i}\|\theta^j - \mu\|^2}{N-1} - \frac{d}{d\mathbb{E}(\alpha^i(x^i)^2)}\lambda_i\|\theta^i - \mu\|^2$$

$$= \sum_{j\neq i}\frac{(1-\beta^j)^2}{(N-1)N^2}((1+C)) - (1-\beta^i)^2(\lambda_i C\frac{(N-1)^2}{N^2})$$

Similarly, assuming $b^j(x^j) = 0$ for all other players $j \neq i$, we get

$$\frac{d}{d\mathbb{E}\|b^i(x^i)\|^2}\mathbb{E}\mathcal{R}^i(\theta^1, \ldots, \theta^N, \mu) = \frac{d}{d\mathbb{E}\|b^i(x^i)\|^2}\frac{\sum_{j\neq i}\|\theta^j - \mu\|^2}{N-1} - \frac{d}{d\mathbb{E}\|b^i(x^i)\|^2}\lambda_i\|\theta^i - \mu\|^2$$

$$= \sum_{j\neq i}\frac{(1-\beta^j)^2}{(N-1)N^2}((1+C)) - (1-\beta^i)^2(\lambda_i C\frac{(N-1)^2}{N^2})$$

Both are negative, whenever $\beta^j = \beta^i \neq 1$ and $\lambda_i C(N-1)^2 > 1 + C$ or $C(\lambda_i(N-1)^2 - 1) > 1$, which is true whenever $C > \frac{1}{\lambda_i(N-1)^2-1}$ and $\lambda_i > \frac{1}{(N-1)^2}$.

But for $\alpha^j = b^j = 0$ for all players $j$, the formula for $(\beta^i)^*$ simplifies to

$$(\beta^i)^* = \frac{C\frac{(N-1)}{N}\bar{\sigma}^2}{\bar{\sigma}^2 + \frac{C(N-1)-1}{N}\bar{\sigma}^2} = \frac{C\frac{(N-1)}{N}}{1 + \frac{C(N-1)-1}{N}} = \frac{C\frac{(N-1)}{N}}{\frac{N-1}{N} + \frac{C(N-1)}{N}} = \frac{C}{1+C},$$

for all players. In particular, we have $(\beta^i)^* < 1$ for $C > 0$, such that $\beta^j = \beta^i \neq 1$ and the derivatives with respect to both $\alpha$ and $b$ are negative for all players, turning $\alpha^j = b^j = 0$, $(\beta^i) = \frac{C}{1+C}$ into a Nash equilibrium.

We can now upper bound $\mathbb{E}\|\theta^i - \mu\|^2$ at the honest equilbrium by considering it at the suboptimal $\beta^i = 0$:

$$\mathbb{E}\|\theta^i - \mu\|^2 \leq \frac{\bar{\sigma}^2}{N} + C\frac{N-1}{N}\bar{\sigma}^2,$$

which is is $O(\frac{\bar{\sigma}^2}{N})$ as long as $C$ is in $O(\frac{1}{N})$, which is the case for constant $\lambda_i = \lambda$, $C = \frac{k}{\lambda(N-1)^2-1}$ and any $k > 1$.

In terms of participation incentives, we again look at the difference in rewards obtained by player $i$ in both cases:

$$\mathbb{E}\mathcal{R}^i(\theta^1, \ldots, \theta^N, \mu) - \mathbb{E}\mathcal{R}^i(\theta^1(N-1), \ldots, \theta^i(1), \ldots, \theta^N(N-1), \mu) \tag{23}$$

$$= \mathbb{E}\left(\frac{\sum_{j\neq i}\|\theta^j(N) - \mu\|^2}{N-1} - \lambda_i\|\theta^i(N) - \mu\|^2\right) \tag{24}$$

$$- \mathbb{E}\left(\frac{\sum_{j\neq i}\|\theta^j(N-1) - \mu\|^2}{N-1} - \lambda_i\|\theta^i(1) - \mu\|^2\right)$$

$$= (1-\lambda_i)\mathbb{E}\|\theta^i(N) - \mu\|^2 - \mathbb{E}\left(\frac{\sum_{j\neq i}\|\theta^j(N-1) - \mu\|^2}{N-1} - \lambda_i\|\theta^i(1) - \mu\|^2\right)$$

It is obvious that $\mathbb{E}\|\theta^i(N) - \mu\|^2 < \mathbb{E}\|\theta^j(N-1) - \mu\|^2 \leq \mathbb{E}\|\theta^i(1) - \mu\|^2$ at the honest equilibrium because of symmetry and as players could otherwise improve their reward by setting $\beta^i = 1$, which

was shown to be suboptimal in the proof of 5.2. This has two implications: First, 23 is positive for $\lambda_i = 1$. Second, the derivative of 23 with respect to $\lambda_i$ is always positive. Combined, this implies that 23 is positive for all $\lambda_i \geq 1$.

Lastly, it is easy to see that $\mathbb{E}\|b^i(x^i)\|^2$ has to be nonzero for at least two players at any other equilibrium. Compared to the honest equilibrium, that increases every players' MSE. But the expected reward for player $i$ equals $\frac{\sum_{j \neq i}\|\theta^j - \mu\|^2}{N-1} - \lambda_i\|\theta^i - \mu\|^2$, such that the sum of all players' rewards equals $\sum_j(1 - \lambda_j)\|\theta^j - \mu\|^2$, which is strictly monotonically decreasing in all players' MSEs when $\lambda_i \geq 1$ for all players. $\qquad\square$

# E   Proofs on stochastic gradient descent

**Outlook**   In this section we prove Theorem 6.1. To this end, we first present the formal definitions of the assumptions on $f$ that we make. Next, we prove two results which bound the difference between the performance of a model resulting from a corrupted optimization scheme (in which players send corrupted estimates) and the performance of a model resulting from honest participation (i.e. from vanilla SGD). In particular, Theorem E.1 provides such a result for a simple penalization scheme, which then easily extends to the penalties presented in Section 6 (Theorem E.2). Next, we combine these results with classic bounds on the distance of the plain SGD trajectory to the minimum value of $f$ (Lemma E.1), to provide an upper bound on the difference between the performance of the corrupted trajectory and the minimum value of $f$ in Theorem E.1. Finally, we show how this last result can be extended to general Lipschitz utilities in Theorem E.2, thereby proving the result from the main text.

**Definitions**   First we formally state our assumptions on the function $f$.

**Definition 2.** A function $f : W \subset \mathbb{R}^n \to \mathbb{R}^d$ is called $L$-Lipschitz with respect to given norms $\|\cdot\|_n$ and $\|\cdot\|_d$ if for all $x, y \in W$
$$\|f(x) - f(y)\|_d \leq L\|x - y\|_n.$$

**Definition 3.** A continously differentiable function $f : W \subset \mathbb{R}^n \to \mathbb{R}$ is called $B$-smooth if its gradient $\nabla f : W \to \mathbb{R}^n$ is $B-$Lipschitz with respect to the euclidean norm.

**Definition 4.** A differentiable function $f : W \subset \mathbb{R}^n \to \mathbb{R}$ is called $m$-strongly convex if for all $x, y \in W$
$$f(x) \geq f(y) + \nabla f(y)^t(x - y) + \frac{m}{2}\|x - y\|^2,$$
where $\|\cdot\|$ denotes the euclidean norm.

We start with proving a weaker version of 6.1 with constant learning rates in which player's paid penalties do not get redistributed:

**Proposition E.1.** *Assume $f$ is $B$-smooth and $L$-Lipschitz with respect to the euclidean norm on $W$ and $m$-strongly convex on $\mathbb{R}^d$. Also assume that for all $i, t$ the gradient noise $e_t^i$ is $B'$-Lipschitz with respect to the euclidean norm with probabiltiy one and that the constant learning rate $\gamma$ fulfills $0 < \gamma < \frac{2m}{B^2 + B'^2}$. Then for the penalized game with reward*

$$\mathcal{R}_p^i(\theta_{T+1}^1, \dots, \theta_{T+1}^N, f) = \left(\frac{1}{N-1}\sum_{j \neq i} f(\theta_{T+1}^j)\right) - \sum_{t=1}^T C_t\|m_t^i - \frac{1}{N}\sum_j m_t^j\|^2,$$

*any player's best response strategy fulfills $\alpha_t^i \leq \frac{LNc^{\frac{T-t}{2}}\gamma}{C_t(N-1)^2} \leq \frac{LN\gamma}{C_t(N-1)^2}$ for $c = (1 + \gamma^2(B^2 + B'^2) - 2\gamma m)$, independent of other players' strategies.*

*Given $\epsilon > 0$, the expected absolute change in function values $f(\theta_T^j)$ due to noise added by players playing best responses compared to full honesty can be bounded by $\frac{1}{1 - \sqrt{c}}L\gamma\epsilon$ by choosing $C_t \geq \frac{LNc^{\frac{T-t}{2}}\gamma}{\epsilon(N-1)^2} < \frac{LN\gamma}{\epsilon(N-1)^2}$. The total penalties paid by player $i$ can then be bounded by $\frac{1}{1 - \sqrt{c}}\frac{L\gamma}{(N-1)}(\frac{G^2}{\epsilon} + \epsilon) \leq$ for a global bound on the "variance" of the gradient estimates $\|e_t^i(\theta)\|^2 \leq G^2$.*

*Proof.* We compare two trajectories $\theta_t$ and $\theta'_t$ starting at the same $\theta_0$ and sharing the same realizations for the noise variables $e_t^j$ and $\xi_t^j$ in which player $i$ employs different agressiveness schedules $\alpha_t^i$ and $(\alpha_t^i)'$ with squared difference $\delta_t^i = (\alpha_t^i - (\alpha_t^i)')^2$. We define $\bar{e}_t = \frac{1}{N}\sum_i e_t^i$. Then:

$$\mathbb{E}\|\theta_{t+1} - \theta'_{t+1}\|^2 = \mathbb{E}\|\Pi_W(\theta_t - \gamma_t(\nabla f(\theta_t) + \bar{e}_t(\theta_t) + \frac{1}{N}\sum_{j\neq i}\alpha_t^j\xi_t^j + \frac{1}{N}\alpha_t^i\xi_t^i))$$

$$- \Pi_W(\theta'_t - \gamma_t(\nabla f(\theta'_t) + \bar{e}_t(\theta'_t) + \frac{1}{N}\sum_{j\neq i}\alpha_t^j\xi_t^j + \frac{1}{N}(\alpha_t^i)'\xi_t^i))\|^2$$

$$\leq \mathbb{E}\|\theta_t - \gamma_t(\nabla f(\theta_t) + \bar{e}_t(\theta_t) + \frac{1}{N}\sum_{j\neq i}\alpha_t^j\xi_t^j + \frac{1}{N}\alpha_t^i\xi_t^i)$$

$$- \theta'_t - \gamma_t(\nabla f(\theta'_t) + \bar{e}_t(\theta'_t) + \frac{1}{N}\sum_{j\neq i}\alpha_t^j\xi_t^j + \frac{1}{N}(\alpha_t^i)'\xi_t^i)\|^2$$

$$= \mathbb{E}\|\theta_t - \theta'_t\|^2 + \gamma_t^2\mathbb{E}\|\nabla f(\theta_t) - \nabla f(\theta'_t)\|^2 + \gamma_t^2\mathbb{E}\|\bar{e}_t(\theta_t) - \bar{e}_t(\theta'_t)\|^2$$

$$+ 2\gamma_t\mathbb{E}|<\theta_t - \theta'_t, \nabla f(\theta'_t) - \nabla f(\theta_t)> + \frac{\gamma_t^2}{N^2}\mathbb{E}\|(\alpha_t^i - (\alpha_t^i)')\xi_t^i\|^2$$

$$- 2\gamma_t\mathbb{E} <\bar{e}_t(\theta_t) - \bar{e}_t(\theta'_t), \theta_t - \theta'_t - \gamma_t(\nabla f(\theta_t) - \nabla f(\theta'_t))>$$

Where the first inequality follows from the well-known $1-$Lipschitzness of projections onto convex closed sets with respect to the euclidean norm (Balashov & Golubev (2012)) and the $\xi^i$ terms factor because of their zero mean and independence of the other variables. Similarly, the last term turns out to equal zero because:

$$\mathbb{E} <\bar{e}_t(\theta_t) - \bar{e}_t(\theta'_t), \theta_t - \theta'_t - \gamma_t(\nabla f(\theta_t) - \nabla f(\theta'_t))>$$
$$= \mathbb{E}[\mathbb{E}[<\bar{e}_t(\theta_t) - \bar{e}_t(\theta'_t), \theta_t - \theta'_t - \gamma_t(\nabla f(\theta_t) - \nabla f(\theta'_t))> |\theta_t, \theta'_t]]$$
$$= \mathbb{E}[< \mathbb{E}[\bar{e}_t(\theta_t) - \bar{e}_t(\theta'_t)|\theta_t, \theta'_t], \theta_t - \theta'_t - \gamma_t(\nabla f(\theta_t) - \nabla f(\theta'_t))>]$$
$$= \mathbb{E}[< 0, \theta_t - \theta'_t - \gamma_t(\nabla f(\theta_t) - \nabla f(\theta'_t))>] = 0$$

as $\mathbb{E}\bar{e}_t(\theta) = 0$ for any fixed $\theta$. Correspondingly,

$$\mathbb{E}\|\theta_{t+1} - \theta'_{t+1}\|^2 = \mathbb{E}\|\theta_t - \theta'_t\|^2 + \gamma_t^2\mathbb{E}\|\nabla f(\theta_t) - \nabla f(\theta'_t)\|^2 + \gamma_t^2\mathbb{E}\|\bar{e}_t(\theta_t) - \bar{e}_t(\theta'_t)\|^2$$

$$+ \frac{\gamma_t^2}{N^2}\mathbb{E}\|(\alpha_t^i - (\alpha_t^i)')\xi_t^i\|^2 + 2\gamma_t\mathbb{E} <\theta_t - \theta'_t, \nabla f(\theta'_t) - \nabla f(\theta_t)>$$

$$= \mathbb{E}\|\theta_t - \theta'_t\|^2 + \gamma_t^2\mathbb{E}\|\nabla f(\theta_t) - \nabla f(\theta'_t)\|^2 + \gamma_t^2\mathbb{E}\|\bar{e}_t(\theta_t) - \bar{e}_t(\theta'_t)\|^2 + \frac{\gamma_t^2}{N^2}\delta_t^i$$

$$+ 2\gamma_t\mathbb{E} <\theta_t - \theta'_t, \nabla f(\theta'_t)> + 2\gamma_t\mathbb{E} <\theta'_t - \theta_t, \nabla f(\theta_t)>$$

$$\leq \mathbb{E}\|\theta_t - \theta'_t\|^2 + \gamma_t^2\mathbb{E}\|\nabla f(\theta_t) - \nabla f(\theta'_t)\|^2 + \gamma_t^2\mathbb{E}\|\bar{e}_t(\theta_t) - \bar{e}_t(\theta'_t)\|^2 + \frac{\gamma_t^2}{N^2}\delta_t^i$$

$$+ 2\gamma_t\mathbb{E}(f(\theta_t) - f(\theta'_t) - \frac{m}{2}\|\theta_t - \theta'_t\|^2 + f(\theta'_t) - f(\theta_t) - \frac{m}{2}\|\theta_t - \theta'_t\|^2)$$

$$= \mathbb{E}\|\theta_t - \theta'_t\|^2 + \gamma_t^2\mathbb{E}\|\nabla f(\theta_t) - \nabla f(\theta'_t)\|^2 + \gamma_t^2\mathbb{E}\|\bar{e}_t(\theta_t) - \bar{e}_t(\theta'_t)\|^2 + \frac{\gamma_t^2}{N^2}\delta_t^i$$

$$- 2\gamma_t m\mathbb{E}\|\theta_t - \theta'_t\|^2$$

$$\leq \mathbb{E}\|\theta_t - \theta'_t\|^2 + \gamma_t^2 B^2\mathbb{E}\|\theta_t - \theta'_t\|^2 + \gamma_t^2(B')^2\mathbb{E}\|\theta_t - \theta'_t\|^2 + \frac{\gamma_t^2}{N^2}\delta_t^i$$

$$- 2\gamma_t m\mathbb{E}\|\theta_t - \theta'_t\|^2$$

$$= (1 + \gamma_t^2(B^2 + B'^2) - 2\gamma_t m)\mathbb{E}\|\theta_t - \theta'_t\|^2 + \frac{\gamma_t^2}{N^2}\delta_t^i \tag{25}$$

Where the first inequality follows from strong convexity.

Now for a constant learning rate $\gamma = \gamma_t$ and $c = (1 + \gamma^2(B^2 + B'^2) - 2\gamma m)$:

$$(\mathbb{E}\frac{1}{L}|f(\theta_{T+1}) - f(\theta'_{T+1})|)^2 \leq (\mathbb{E}\|\theta_{T+1} - \theta'_{T+1}\|)^2$$

$$\leq \mathbb{E}\|\theta_{T+1} - \theta'_{T+1}\|^2$$

$$\leq c^T \mathbb{E}\|\theta_1 - \theta'_1\|^2 + \sum_{t=1}^{T} c^{T-t}\frac{\gamma^2}{N^2}\delta_t^i$$

$$= \sum_{t=1}^{T} c^{T-t}\frac{\gamma^2}{N^2}\delta_t^i$$

so that

$$\mathbb{E}|f(\theta_{T+1}) - f(\theta'_{T+1})| \leq L\sqrt{\sum_{t=1}^{T} c^{T-t}\frac{\gamma^2}{N^2}\delta_t^i} \leq L\sum_{t=1}^{T}\frac{\gamma}{N}c^{\frac{T-t}{2}}\sqrt{\delta_t^i} = \frac{L\gamma}{N}\sum_{t=1}^{T} c^{\frac{T-t}{2}}|\alpha_t^i - (\alpha_t^i)'|.$$

Where the last inequality follows from the general inequality $\sqrt{\sum_i x_i} \leq \sum_i \sqrt{x_i}$ for $x_i \geq 0$, which inductively follows from

$$\sqrt{x + y} = \sqrt{(\sqrt{x} + \sqrt{y})^2 - 2\sqrt{xy}} \leq \sqrt{(\sqrt{x} + \sqrt{y})^2} = \sqrt{x} + \sqrt{y}.$$

In particular if we set $\theta'_t$ to the trajectory in which player $i$ is honest $((\alpha_t^i)' = 0)$, we obtain

$$\mathbb{E}|f(\theta_{T+1}) - f(\theta'_{T+1})| \leq \frac{L\gamma}{N}\sum_{t=1}^{T} c^{\frac{T-t}{2}}\alpha_t^i. \tag{26}$$

The same inequalities holds for players' final estimates $\theta_{T+1}^j$ and $(\theta_{T+1}^j)'$, as the noise correction step $\theta_{T+1}^i = \theta_{T+1} - \frac{\alpha_T^i}{N}\xi^i$ is the same in both cases, so that

$$\mathbb{E}\|\theta_{T+1}^i - (\theta_{T+1}^i)'\|^2 = \mathbb{E}\|\theta_{T+1} - (\theta_{T+1})'\|^2.$$

It is worth noting, that the contribution of noise at early time steps to the sum diminishes exponentially as long as $c < 1$ which is true for $\gamma^2(B^2 + B'^2) - 2\gamma m < 0$, i.e. $\gamma < \frac{2m}{B^2+B'^2}$.

Next, we consider the expected difference in penalties received by player $i$ at time $t$ if they use $(\alpha_t^i)'$ rather than $\alpha_t^i$ and thus send message $(m_t^i)'$ rather than $m_t^i$:

$$\mathbb{E}p_i^t(m_t^1, \ldots, (m_t^i)', \ldots, m_t^N) - p_i^t(m_t^1, \ldots, m_t^i, \ldots, m_t^N)$$

$$= \mathbb{E}\|\frac{N-1}{N}(m_t^i)' - \frac{1}{N}\sum_{j\neq i}m_t^j\|^2 - \mathbb{E}\|\frac{N-1}{N}m_t^i - \frac{1}{N}\sum_{j\neq i}m_t^j\|^2$$

$$= \mathbb{E}\|\frac{N-1}{N}((\alpha_t^i)'\xi_t^i + g_t^i) - \frac{1}{N}\sum_{j\neq i}m_t^j\|^2 - \mathbb{E}\|\frac{N-1}{N}(\alpha_t^i\xi_t^i + g_t^i) - \frac{1}{N}\sum_{j\neq i}m_t^j\|^2$$

$$= \mathbb{E}\|\frac{N-1}{N}(\alpha_t^i)'\xi_t^i\|^2 - \mathbb{E}\|\frac{N-1}{N}(\alpha_t^i)'\xi_t^i\|^2$$

$$+ \mathbb{E}\|\frac{N-1}{N}(g_t^i) - \frac{1}{N}\sum_{j\neq i}m_t^j\|^2 - \mathbb{E}\|\frac{N-1}{N}(g_t^i) - \frac{1}{N}\sum_{j\neq i}m_t^j\|^2$$

$$= (\frac{N-1}{N}(\alpha_t^i)')^2 - (\frac{N-1}{N}\alpha_t^i)^2$$

for $g_t^i = g_t(\theta_{t-1}^s, x^i)$.

In particular, we can bound the difference between expected penalized rewards for two trajectories with all $\alpha_k^i$ fixed but two different values $\alpha_t^i$ and $(\alpha_t^i)'$ varying for $k = t$ as follows:

$$\mathbb{E}\left(\mathcal{R}^i(\theta_{T+1}) - \sum_k C_k p_k^i(m_k^1, \ldots, m_k^i, \ldots, m_k^N)\right.$$

$$\left. - \mathcal{R}^i((\theta_{T+1})') + \sum_k C_k p_k^i(m_k^1, \ldots, (m_k^i)', \ldots, m_k^N)\right)$$

$$= \frac{1}{N-1}\sum_{j \neq i}(f(\theta_{T+1}^j) - f((\theta_{T+1}^j)')) + C_t((\frac{N-1}{N}(\alpha_t^i)')^2 - (\frac{N-1}{N}\alpha_t^i)^2)$$

$$\leq L\frac{\gamma}{N}c^{\frac{T-t}{2}}|\alpha_t^i - (\alpha_t^i)'| + C_t((\frac{N-1}{N}(\alpha_t^i)')^2 - (\frac{N-1}{N}\alpha_t^i)^2)$$

In particular, for $(\alpha_i^t)' = 0$, we obtain

$$\mathbb{E}\left(\mathcal{R}^i(\theta_T) - \sum_k C_k p_i^t(m_k^1, \ldots, m_k^i, \ldots, m_k^N) - \mathcal{R}^i((\theta_T)') + \sum_t C_k p_k^i(m_k^1, \ldots, (m_k^i)', \ldots, m_k^N)\right)$$

$$\leq L\frac{\gamma}{N}c^{\frac{T-t}{2}}\alpha_t^i - C_t((\frac{N-1}{N}\alpha_t^i)^2)$$

By the quadratic formula, this is zero at zero and at

$$\alpha_t^i = \frac{-2L\frac{\gamma}{N}c^{\frac{T-t}{2}}}{-2C_t(\frac{N-1}{N})^2} = \frac{LNc^{\frac{T-t}{2}}\gamma}{C_t(N-1)^2} \tag{27}$$

and because of the negative quadratic term negative whenever $\alpha_t^i > \frac{LNc^{\frac{T-t}{2}}\gamma}{C_t(N-1)^2}$. Correspondingly, in terms of penalized reward players are always better off by not adding any noise at all $\alpha_t^i = 0$ compared to adding large noise, such that rational players will never choose $\alpha_t^i > \frac{LNc^{\frac{T-t}{2}}\gamma}{C_t(N-1)^2}$. Therefore, the noise $\alpha_t^i$ added by any player $i$ at a given time step $t$ can be limited to any fixed constant $\epsilon > 0$ by choosing $C_t$ such that $\frac{LNc^{\frac{T-t}{2}}\gamma}{C_t(N-1)^2} \leq \epsilon$, i.e. $C_t \geq \frac{LNc^{\frac{T-t}{2}}\gamma}{\epsilon(N-1)^2}$.

Applying this observation to each player, substituting into Equation (26) and using the triangle inequality, the overall damage caused by all $N$ players compared to full honesty can then be bounded by

$$\mathbb{E}|f(\theta_{T+1}) - f(\theta_{T+1}')| \leq L\gamma \sum_{t=1}^T c^{\frac{T-t}{2}}\epsilon$$

where $\theta_t'$ represents the fully honest strategy and $\theta_t$ represents a strategy in which all players act rationally given the penalty magnitude $C_t$. Using a geometric series bound, we obtain $\mathbb{E}|f(\theta_{T+1}) - f(\theta_{T+1}')| \leq \frac{1}{1-\sqrt{c}}L\gamma\epsilon$.

Lastly, for a global bound on the "variance" of the gradients $\|e_t^i(\theta)\|^2 \leq G^2$ we get,

$$\mathbb{E}p_i^t(m_t^1, \ldots, m_t^i, \ldots, m_t^N)$$

$$= \mathbb{E}\|\frac{N-1}{N}m_t^i - \frac{1}{N}\sum_{j \neq i}m_t^j\|^2$$

$$= \mathbb{E}\|\frac{N-1}{N}(g_t^i + \alpha_t^i\xi_t^i) - \frac{1}{N}\sum_{j \neq i}(g_t^j + \alpha_t^j\xi_t^j)\|^2$$

$$= \mathbb{E}\|\frac{N-1}{N}g_t^i - \frac{1}{N}\sum_{j \neq i}g_t^j\|^2 + (\frac{N-1}{N})^2(\alpha_t^i)^2 + \frac{1}{N^2}\sum_{j \neq i}(\alpha_t^j)^2$$

$$\leq \mathbb{E}\|\frac{N-1}{N}(g_t^i(\theta) - \nabla f(\theta)) - \frac{1}{N}\sum_{j \neq i}(g_t^j(\theta) - \nabla f(\theta)) + (\frac{N-1}{N} - \frac{N-1}{N})\nabla f(\theta)\|^2$$

$$+ (\frac{N-1}{N})\epsilon^2$$

$$= \mathbb{E}[\mathbb{E}[\|\frac{N-1}{N}e_t^i(\theta) - \frac{1}{N}\sum_{j\neq i}e_t^j(\theta)\|^2|\theta]] + (\frac{N-1}{N})\epsilon^2$$

$$= (\frac{N-1}{N})^2\mathbb{E}[\mathbb{E}[\|e_t^i(\theta)\|^2|\theta] + \frac{1}{N^2}\sum_{j\neq i}\mathbb{E}[\mathbb{E}[\|(e_t^j(\theta)\|^2|\theta]] + (\frac{N-1}{N})\epsilon^2$$

$$\leq (\frac{N-1}{N})(G^2 + \epsilon^2)$$

as the $\xi_t^i$ are independent with zero mean, and the gradient noise $e_t^i(\theta)$ are independent with zero mean, given $\theta$. Correspondingly, for $C_t = \frac{LNc^{\frac{T-t}{2}}\gamma}{\epsilon(N-1)^2}$ the total expected penalties paid by player $i$ can be bounded as

$$\mathbb{E}\sum_t^T C_t p_i^t(m_t^1, \ldots, m_t^i, \ldots, m_t^N)$$

$$\leq \sum_t^T \frac{Lc^{\frac{T-t}{2}}\gamma}{\epsilon(N-1)}(K^2 + \epsilon^2)$$

$$\leq \frac{1}{1-\sqrt{c}}\frac{L\gamma}{(N-1)}(\frac{K^2}{\epsilon} + \epsilon)$$

□

Next, we prove the budget-balanced version of E.1

**Proposition E.2.** *Under the assumptions of E.1, in the balanced penalized game with reward*

$$\mathcal{R}_p^i(\theta_{T+1}^1, \ldots, \theta_{T+1}^N, f) = \left(\frac{1}{N-1}\sum_{j\neq i}f(\theta_{T+1}^j)\right) - \sum_{t=1}^T C_t\|m_t^i - \frac{1}{N}\sum_j m_t^j\|^2$$

$$+ \frac{1}{N-1}\sum_{k\neq i}\sum_{t=0}^T C_t\|m_t^k - \frac{1}{N}\sum_j m_t^j\|^2,$$

*any player's best response strategy fulfills $\alpha_t^i \leq \frac{Lc^{\frac{T-t}{2}}\gamma}{C_t(N-2)} \leq \frac{L\gamma}{C_t(N-2)}$ for $c = (1 + \gamma^2(B^2 + B'^2) - 2\gamma m)$, independent of other players' strategies.*

*Given $\epsilon > 0$, the expected absolute change in function values $f(\theta_T^j)$ due to noise added by players playing best responses compared to full honesty can be bounded by $\frac{1}{1-\sqrt{c}}L\gamma\epsilon$ by choosing $C_t \geq \frac{Lc^{\frac{T-t}{2}}\gamma}{\epsilon(N-2)} \leq \frac{L\gamma}{\epsilon(N-2)}$. As long as all players $i$ choose the same strategy ($\alpha_t^i = \alpha_t^j \ \forall i, j, t$), the expected total penalty paid by each player equals zero.*

*Proof.* We begin by considering the expected difference in penalties received by player $l$ at time $t$ if player $i$ uses $(\alpha_t^i)'$ rather than $\alpha_t^i$ and thus send message $(m_t^i)'$ rather than $m_t^i$:

$$\mathbb{E}p_t^l(m_t^1, \ldots, (m_t^i)', \ldots, m_t^N) - p_t^l(m_t^1, \ldots, m_t^i, \ldots, m_t^N)$$

$$= \mathbb{E}\|m_t^l - \frac{1}{N}\sum_{j\neq i}m_t^j - \frac{1}{N}(m_t^i)'\|^2 - \mathbb{E}\|m_t^l - \frac{1}{N}\sum_{j\neq i}m_t^j - \frac{1}{N}m_t^i\|^2$$

$$= \mathbb{E}\|m_t^l - \frac{1}{N}\sum_{j\neq i}m_t^j - \frac{1}{N}((\alpha_t^i)'\xi_t^i + g_t^i)\|^2 - \mathbb{E}\|m_t^l - \frac{1}{N}\sum_{j\neq i}m_t^j - \frac{1}{N}(\alpha_t^i\xi_t^i + g_t^i)\|^2$$

$$= \mathbb{E}\|m_t^l - \frac{1}{N}\sum_{j\neq i}m_t^j - \frac{1}{N}g_t^i\|^2 + \mathbb{E}\frac{1}{N^2}\|(\alpha_t^i)'\xi_t^i\|^2$$

$$- \mathbb{E}\|m_t^l - \frac{1}{N}\sum_{j\neq i} m_t^j - \frac{1}{N}g_t^i\|^2 - \mathbb{E}\frac{1}{N^2}\|\alpha_t^i \xi_t^i\|^2$$

$$= (\frac{(\alpha_t^i)'}{N})^2 - (\frac{\alpha_t^i}{N})^2$$

Again, we can bound the difference between expected penalized rewards for two trajectories with all $\alpha_k^i$ fixed but two different values $\alpha_t^i$ and $(\alpha_t^i)'$ varying for $k = t$ as follows:

$$\mathbb{E}(\mathcal{R}^i(\theta_{T+1}) - \sum_k C_k p_k^i(m_k^1,\ldots,m_k^i,\ldots,m_k^N)$$

$$+ \frac{1}{N-1}\sum_{j\neq i}\sum_k C_k p_k^j(m_k^1,\ldots,m_k^i,\ldots,m_k^N)$$

$$- \mathcal{R}^i((\theta_{T+1})') + \sum_t C_k p_k^i(m_k^1,\ldots,(m_k^i)',\ldots,m_k^N)$$

$$- \frac{1}{N-1}\sum_{j\neq i}\sum_t C_k p_k^j(m_k^1,\ldots,(m_k^i)',\ldots,m_k^N))$$

$$= \frac{1}{N-1}\sum_{j\neq i}(f(\theta_{T+1}^j) - f((\theta_{T+1}^j)'))$$

$$+ C_t((\frac{N-1}{N}(\alpha_t^i)')^2 - (\frac{1}{N}(\alpha_t^i)')^2 - (\frac{N-1}{N}\alpha_t^i)^2) + (\frac{1}{N}\alpha_t^i)^2)$$

$$= \frac{1}{N-1}\sum_{j\neq i}(f(\theta_{T+1}^j) - f((\theta_{T+1}^j)'))$$

$$+ C_t((\frac{N-1}{N}(\alpha_t^i)')^2 - (\frac{1}{N}(\alpha_t^i)')^2 - (\frac{N-1}{N}\alpha_t^i)^2) + (\frac{1}{N}\alpha_t^i)^2)$$

$$\leq L\frac{\gamma}{N}c^{\frac{T-t}{2}}|\alpha_t^i - (\alpha_t^i)'| + C_t(\frac{N-2}{N}((\alpha_t^i)')^2 - \frac{N-2}{N}(\alpha_t^i)^2)$$

Again, for $(\alpha_t^i)' = 0$ we obtain

$$\mathbb{E}(\mathcal{R}^i(\theta_{T+1}) - \sum_k C_k p_k^i(m_k^1,\ldots,m_k^i,\ldots,m_k^N)$$

$$+ \frac{1}{N-1}\sum_{j\neq i}\sum_k C_k p_k^j(m_k^1,\ldots,m_k^i,\ldots,m_k^N)$$

$$- \mathcal{R}^i((\theta T+1)') + \sum_k C_k p_k^i(m_k^1,\ldots,(m_k^i)',\ldots,m_k^N)$$

$$- \frac{1}{N-1}\sum_{j\neq i}\sum_k C_k p_k^j(m_k^1,\ldots,(m_k^i)',\ldots,m_k^N))$$

$$\leq L\frac{\gamma}{N}c^{\frac{T-t}{2}}\alpha_t^i - C_t(\frac{(N-2)}{N}(\alpha_t^i)^2)$$

which is zero at zero and at

$$\alpha_t^i = \frac{-2L\frac{\gamma}{N}c^{\frac{T-t}{2}}}{-2C_t\frac{N-2}{N}} = \frac{Lc^{\frac{T-t}{2}}\gamma}{C_t(N-2)} \tag{28}$$

and negative for $\alpha_t^i > \frac{Lc^{\frac{T-t}{2}}\gamma}{C_t(N-2)}$ as long as $N > 2$. Players are thus again incentivized to select $\alpha_t^i$ that do not fulfill that inequality.

As in E.1, the overall damage caused by all $N$ players compared to full honesty can then be bounded by $\mathbb{E}|f(\theta_{T+1}) - f(\theta'_{T+1})| \leq \frac{1}{1-\sqrt{c}}L\gamma\epsilon$ by choosing $C_t \geq \frac{Lc^{\frac{T-t}{2}}\gamma}{\epsilon(N-2)}$ where $\theta'_T$ represents the fully

honest strategy and $\theta_T$ represents a strategy in which all players act rationally given the penalty magnitude $C_t$, and the same is true for player's estimates $\theta^j_{T+1}$ and $(\theta^j_{T+1})'$. By symmetry, as long as all players $i$ choose the same strategy ($\alpha^i_t = \alpha^j_t \; \forall i, j, t$), the expected penalties paid by each player equal 0. $\qquad\square$

To prove 6.1, we adapt a classic result from convex optimization to give convergence rates for SGD with bounded perturbations from the clients and with a linearly decaying learning rate.

**Lemma E.1.** *In the settings of E.2, assume that all players use bounded attacks, so that $(\alpha^i_t)^2 \leq \epsilon^2$ for all $i, t$. Also, assume that there exist scalars $M \geq 0$ and $M_V \geq 0$, such that for all $t$:*

$$\mathbb{E}_{s_i}(\|e^i_t(\theta^s_t)\|^2) = \mathbb{E}_{x^i}(\|g_t(\theta^s_t, x^i)\|^2) - \|\mathbb{E}_{x^i} g_t(\theta^s_t, x^i)\|^2 \leq M + M_V \|\nabla f(\theta^s_t)\|^2_2. \tag{29}$$

*Assume that for some integer constant $\eta > 0$, such that $\frac{4}{\eta m + m} \leq \frac{1}{B(M_V/N+1)}$, the learning rate is set as $\gamma_t = \frac{4}{\eta m + tm}$. In that case, if $P(\exists t \leq T : \Pi_W(\theta^s_t - \gamma_t \bar{m}_t) \neq \theta^s_t - \gamma_t \bar{m}_t) \in O(\frac{1}{NT})$ we get $\mathbb{E}(f(\theta_t) - f(\theta^*)) \in O(\frac{1+M+\epsilon^2}{Nt}) + O(\frac{1}{t^2})$, we have:*

$$\mathbb{E}(f(\theta_T) - f(\theta^*)) \leq \frac{8B(M+\epsilon^2)}{3m^2 NT} + \mathcal{O}\left(\frac{1}{NT}\right) + \mathcal{O}\left(\frac{1}{T^2}\right) \tag{30}$$

*for any $T \geq \eta$.*

*Proof.* We first condition on the case in which there is no $t \leq T$ with $\Pi_W(\theta^s_t - \gamma_t \bar{m}_t) \neq \theta^s_t - \gamma_t \bar{m}_t$, so that we do not have to worry about projections. Then, for a random vector $g$, denote $\mathbb{V}(g) = \mathbb{E}(\|g\|^2) - \|\mathbb{E}(g)\|^2$. Note that, by the independence of the stochastic gradients and players' noise, it follows that:

$$\mathbb{V}\left(\frac{1}{N}\sum_{i=1}^{N} m^t_i\right) = \mathbb{E}\|\frac{1}{N}\sum_{i=1}^{N}(\nabla f(\theta_t) + e^i_t(\theta^s_t) + \alpha^i_t \xi^i_t)\|^2 - \|\mathbb{E}(\frac{1}{N}\sum_{i=1}^{N}(\nabla f(\theta_t) + e^i_t(\theta^s_t) + \alpha^i_t \xi^i_t))\|^2$$

$$= \frac{1}{N^2}\mathbb{E}\|\sum_{i=1}^{N}(\nabla f(\theta_t) + e^i_t(\theta^s_t) + \alpha^i_t \xi^i_t)\|^2 - \frac{1}{N^2}\|\nabla f(\theta_t)\|^2$$

$$= \frac{1}{N^2}(\mathbb{E}\|\sum_{i=1}^{N} e^i_t(\theta^s_t)\|^2 + \mathbb{E}\|\sum_{i=1}^{N} \alpha^i_t \xi^i_t\|^2 + \|\nabla f(\theta_t)\|^2) - \frac{1}{N^2}\|\nabla f(\theta_t)\|^2$$

$$= \frac{1}{N^2}(\sum_{i=1}^{N}\mathbb{E}\|e^i_t(\theta^s_t)\|^2 + \sum_{i=1}^{N}(\alpha^i_t)^2 \mathbb{E}\|\xi^i_t\|^2)$$

$$\leq \frac{1}{N^2}(\sum_{i=1}^{N} M + M_V\|\nabla f(\theta^s_t))\|^2_2 + \sum_{i=1}^{N}(\alpha^i_t)^2)$$

$$\leq \frac{M+\epsilon^2}{N} + \frac{M_V}{N}\|\nabla F(\theta^s_t)\|^2_2$$

Additionally, since $f$ is strongly convex, it has a unique minimizer $\theta^* \in \mathbb{R}^d$. Now since the learning rate is of the form $\gamma_t = \frac{4/m}{\eta+t}$, with $\frac{4}{m} > \frac{1}{m}$, $\eta > 0$ and $\gamma_1 = \frac{\beta}{\eta+1} \leq \frac{1}{B(M_V/N+1)}$, the conditions of Theorem 4.7 in Bottou et al. (2018) hold with $\mu = 1, M_V = M_V/N, M = (M+\epsilon^2)/N, M_G = M_V/N + 1$. Using equation 4.23 in their proof gives:

$$\mathbb{E}(f(\theta_{t+1}) - f(\theta^*)) \leq \left(1 - \frac{4}{\eta+t}\right)\mathbb{E}(f(\theta_t) - f(\theta^*)) + \frac{8B(M+\epsilon^2)}{Nm^2(\eta+t)^2} \tag{31}$$

for any $t \geq 1$. We now use a classic result by Chung:

**Lemma E.2** (Chung (1954)). *Let $\{b_n\}_{n\geq 1}$ be a sequence of real numbers, such that for some $n_0 \in \mathbb{N}$, it holds that for all $n \geq n_0$,*

$$b_{n+1} \leq \left(1 - \frac{d}{n}\right) b_n + \frac{c}{n^2},$$

*where $c > 1, c_1 > 0$. Then*

$$b_n \leq \frac{c}{d-1}\frac{1}{n} + \mathcal{O}\left(\frac{1}{n^2} + \frac{1}{n^d}\right).$$

We set $x_{t+\eta} := \mathbb{E}\left(f(\theta_t) - f(\theta^*)\right)$ for $t \geq 1$ and $x_k = \mathbb{E}\left(f(\theta_1) - f(\theta^*)\right)$ for $k \leq \eta$. Using 31 and $k = t + \eta$ we get

$$x_{k+1} \leq (1 - \frac{4}{k})x_k + \frac{8B(M + \epsilon^2)}{Nm^2 k^2}$$

Now using $d = 4$ and $c = \frac{8B(M+\epsilon^2)}{Nm^2}$, we have $x_{k+1} \leq (1 - \frac{d}{k})x_k + \frac{c}{k^2}$ such that E.2 yields

$$x_t \leq \frac{8B(M + \epsilon^2)}{3Nm^2 t} + \mathcal{O}\left(\frac{1}{t^2} + \frac{1}{t^d}\right)$$

and thus

$$\mathbb{E}\left(f(\theta_t) - f(\theta^*)\right) \leq \frac{8B(M + \epsilon^2)}{3Nm^2(t + \eta)} + \mathcal{O}\left(\frac{1}{t^2} + \frac{1}{t^d}\right) \leq \frac{8B(M + \epsilon^2)}{3Nm^2 t} + \mathcal{O}\left(\frac{1}{t^2} + \frac{1}{t^4}\right)$$

Now, if there is a $t$ with $\Pi_W(\theta_t^s - \gamma_t \bar{m}_t) \neq \theta_t^s - \gamma_t \bar{m}_t$, we can still bound $\mathbb{E}\left(f(\theta_t) - f(\theta^*)\right)$ by some constant because $W$ is bounded and $f$ is Lipschitz. Correspondingly, as $P(\exists t \leq T : \Pi_W(\theta_t^s - \gamma_t \bar{m}_t) \neq \theta_t^s - \gamma_t \bar{m}_t) \in O(\frac{1}{NT})$, the total expectation for both cases combined is bounded by $\frac{8B(M+\epsilon^2)}{3m^2 NT} + \mathcal{O}\left(\frac{1}{NT}\right) + \mathcal{O}\left(\frac{1}{T^2}\right)$.

$\square$

To prove theorem 6.1, we use a non-asymptotic version of Chung's Lemma Chung (1954) similar to the one used in the proof of Lemma 1 in Rakhlin et al. (2012):

**Lemma E.3.** *For constants $c > 0$ and $d > 1$, whenever $t + 1 \geq d$ and the recursive inequality*

$$x_{t+1} \leq (1 - \frac{d}{t+1})x_t + \frac{c}{(t+1)^2},$$

*holds we get that if*

$$x_t \leq \frac{2d^2 c}{t(d^3 - d^2)}$$

*for $t = k$ the same is true for $t = k + 1$.*

*Proof.* Using the condition on $x_t$, we obtain

$$x_{t+1} \leq (1 - \frac{d}{t+1})x_t + \frac{c}{(t+1)^2}$$
$$\leq \frac{2d^2 c}{t(d^3 - d^2)} - \frac{2d^3 c}{t(t+1)(d^3 - d^2)} + \frac{c}{(t+1)^2}$$

as by assumption $\frac{d}{t+1} \leq 1$. As $d > 1$, $\frac{2d^2 c}{(t+1)(d^3 - d^2)}$ is positive and we can divide the equation above by it to obtain

$$\frac{x_{t+1}(t+1)(d^3 - d^2)}{2d^2 c} \leq \frac{t+1}{t} - \frac{d}{t} + \frac{(d^3 - d^2)}{(t+1)2d^2}$$

for which we want to show that it is bounded above by 1. Multiplying the equation

$$\frac{t+1}{t} - \frac{d}{t} + \frac{(d^3 - d^2)}{(t+1)2d^2} \leq 1$$

by $t(t+1)$ for $t \geq 1$ we get

$$t^2 + 2t + 1 - dt - d + t\frac{(d^3 - d^2)}{2d^2} \leq t^2 + t$$

which is equivalent to

$$t + 1 - dt - d + \frac{t}{2}(d-1) \leq 0,$$

i.e.

$$\frac{1}{2}(1-d)t + 1 - d \leq 0,$$

and

$$(1-d)t \leq 2(d-1),$$

which is true whenever

$$t \geq -2.$$

$\square$

**Theorem E.1.** *In the settings of E.2, assume that there exist scalars $M \geq 0$ and $M_V \geq 0$, such that for all $t$:*

$$\mathbb{E}_{s_i}(\|(e_t^i(\theta_t^s)\|^2) \leq M + M_V \|\nabla f(\theta_t^s))\|_2^2. \tag{32}$$

*Assume that for some integer constant $\eta > 1$, such that $\frac{4}{\eta m + m} \leq \frac{1}{B(M_V/N+1)}$, the learning rate is set as $\gamma_t = \frac{4}{\eta m + tm}$.*

*Then any player's best response strategy fulfills $\alpha_i^t \leq \frac{8L}{C^t(N-2)m\sqrt{T1+\eta}}$ independent of other players' strategies, such that for any given $\epsilon > 0$, $C^t \geq \frac{8L}{\epsilon(N-2)m\sqrt{T+\eta}}$ yields $\alpha_i^t \leq \epsilon$ for rational players. In that case, as long as $W$ is bounded and we have that $P(\exists t \leq T : \Pi_W(\theta_t^s - \gamma_t\bar{m}_t) \neq \theta_t^s - \gamma_t\bar{m}_t) \in O(\frac{1}{NT})$ we get $\mathbb{E}(f(\theta_t) - f(\theta^*)) \in O(\frac{1+M+\epsilon^2}{Nt}) + O(\frac{1}{t^2})$.*

*Proof.* We make use of inequality 25 from the proof of E.1 to analyse the difference between two trajectories that are identical except for the actions of player $i$.

$$\mathbb{E}\|\theta_{t+1} - \theta'_{t+1}\|^2$$

$$\leq (1 + \gamma_t^2(B^2 + B'^2) - 2\gamma_t m)\mathbb{E}\|\theta_t - \theta'_t\|^2 + \frac{\gamma_t^2}{N^2}\delta_i^t$$

$$= (1 + \frac{16}{m^2(\eta+t)^2}(B^2 + B'^2) - \frac{8}{\eta+t})\mathbb{E}\|\theta_t - \theta'_t\|^2 + \frac{16}{(\eta+t)^2N^2m^2}\delta_i^t.$$

$$\leq (1 + \frac{16}{m^2(\eta+t)^2}(B^2 + B'^2) - \frac{8}{\eta+t})\mathbb{E}\|\theta_t - \theta'_t\|^2 + \frac{16}{(\eta+t)^2N^2m^2}\delta_i^t.$$

For $t \geq 0$ and $\eta \geq \max\{\frac{32(B^2+B'^2)}{13m^2}, 1\}$ we get

$$16\frac{B^2 + B'^2}{m^2(t+\eta)^2} - \frac{8}{t+\eta} \leq -\frac{1.5}{t+\eta} \tag{33}$$

by calculating

$$\eta \geq \frac{32(B^2 + B'^2)}{13m^2}$$

$$\implies 6.5\eta + 6.5t \geq \frac{16(B^2 + B'^2)}{m^2}$$

$$\implies 8\eta + 8t - \frac{16(B^2 + B'^2)}{m^2} \geq 1.5\eta + 1.5t$$

$$\implies 8\eta + 8t - \frac{16(B^2 + B'^2)}{m^2} \geq 1.5\frac{(t+\eta)^2}{t+\eta}$$

$$\implies \frac{8}{\eta + t} - \frac{16(B^2 + B'^2)}{(\eta + t)^2 m^2} \geq \frac{1.5}{t + \eta}$$

$$\implies \frac{16(B^2 + B'^2)}{(\eta + t)^2 m^2} - \frac{8}{\eta + t} \leq -\frac{1.5}{t + \eta}$$

We now set $x_{t+\eta} := \mathbb{E}\|\theta_{t+1} - \theta'_{t+1}\|^2$ for $t \geq 0$ and $x_k = 0$ for $k \leq \eta$. These definitions are consistent for $t = 0$ because $\mathbb{E}\|\theta_1 - \theta'_1\|^2 = 0$.

Using 33 and $k = t - 1 + \eta$, we get

$$x_{k+1} \leq (1 - \frac{1.5}{k+1})x_k + \frac{16}{(k+1)^2 N^2 m^2}\delta_i^t.$$

At the same time, $x_\eta = \mathbb{E}\|\theta_1 - \theta'_1\|^2 = 0 \leq \frac{4c}{\eta - 1}$ for any $c > 0$. Correspondingly, E.3 with $d = 1.5$ and $c = \frac{16}{N^2 m^2} \max_t\{\delta_i^t\}$ implies that for $k \geq \eta$ and $k + 1 \geq 1.5$ we get

$$x_k \leq \frac{4c}{k}$$

and thus

$$\mathbb{E}\|\theta_t - \theta'_t\|^2 \leq \frac{4c}{t - 1 + \eta}$$

for $t \geq 1$. This yields

$$\mathbb{E}\|\theta_{T+1} - \theta'_{T+1}\|^2 \leq \frac{64 \max_t\{\delta_i^t\}}{(T + \eta)N^2 m^2}.$$

Consequentially, we obtain

$$\mathbb{E}|f(\theta_{T+1}) - f(\theta'_{T+1})| \leq L\sqrt{\frac{64 \max_t\{\delta_i^t\}}{(T + \eta)N^2 m^2}}.$$

Again, we can bound the difference between expected penalized rewards for two trajectories with all $\alpha_k^i$ fixed but two different values $\alpha_t^i$ and $(\alpha_t^i)'$ varying for $k = t$ by considering $\delta_k = 0$ for all $k \neq t$. This yields

$$\mathbb{E}|f(\theta_{T+1}) - f(\theta'_{T+1})| \leq L\sqrt{\frac{64\delta_i^t}{(T + \eta)N^2 m^2}} = L\sqrt{\frac{64|\alpha_t^i - (\alpha_t^i)'|^2}{(T + \eta)N^2 m^2}}.$$

As in E.2, this allows us to upper bound the gains in penalized reward from changing $\alpha_t^i$ to $(\alpha_t^i)'$ by

$$\frac{8L}{Nm\sqrt{T+\eta}}|\alpha_t^i - (\alpha_t^i)'| + C_t(\frac{N-2}{N}((\alpha_t^i)')^2 - \frac{N-2}{N}(\alpha_t^i)^2).$$

In particular, for $(\alpha_t^i)' = 0$ this bound becomes

$$\frac{8L}{Nm\sqrt{T+\eta}}\alpha_t^i - C_t(\frac{N-2}{N}(\alpha_t^i)^2)$$

which is zero at $\alpha_t^i = 0$ and at

$$\alpha_t^i = \frac{-2\frac{8L}{Nm\sqrt{T+\eta}}}{-2C_t\frac{N-2}{N}} = \frac{8L}{C_t(N-2)m\sqrt{T+\eta}} \tag{34}$$

and negative for $\alpha_t^i$ larger than that as long as $N > 2$.

Again the noise $\alpha_t^i$ added by any rational player $i$ at a given time step $t$ can therefore be limited to any fixed constant $\epsilon > 0$ by choosing $C_t$ such that $\frac{8L}{C_t(N-2)m\sqrt{T+\eta}} \leq \epsilon$, i.e. $C_t \geq \frac{8L}{\epsilon(N-2)m\sqrt{T+\eta}}$.

We conclude by using E.1 to obtain the convergence rate. $\qquad\square$

As a final step, we show that a version of E.1 also holds for more general reward functions, thus proving 6.1.

**Theorem E.2.** *Up to constants, theorems E.2 and E.1 also hold for the reward* $\mathcal{R}^i_{U_p}(\theta^1_{T+1}, \ldots, \theta^N_{T+1}, f) = U^i(f(\theta^i_{T+1}), \ldots, f(\theta^N_{T+1})) - \sum_{t=1}^T C^U_t \|m^i_t - \frac{1}{N}\sum_j m^j_t\|^2 + \frac{1}{N-1}\sum_{k\neq i}\sum_{t=1}^T C^U_t \|m^k_t - \frac{1}{N}\sum_j m^j_t\|^2$ *for arbitrary* $l_1$*-Lipschitz* $U^i$ *with common Lipschitz constant* $L_U$. *Any bound on* $\alpha^i_t$ *can be achieved by setting* $C^U_t = L_U N C_t$ *for the* $C_t$ *achieving the same bound in E.2 or E.1 respectively.*

*Proof.* We first note that for any pointwise bound $\mathbb{E}|f(\theta^j_{T+1}) - f((\theta^j_{T+1}))'| \leq K$

$$\mathbb{E}|U^i(f(\theta^i_{T+1}), \ldots, f(\theta^N_{T+1})) - U^i(f((\theta^i_{T+1})'), \ldots, f((\theta^N_{T+1})'))|$$

$$\leq \mathbb{E}L_U \sum_j |f(\theta^j_{T+1}) - f((\theta^j_{T+1}))'|$$

$$= L_U \sum_j \mathbb{E}|f(\theta^j_{T+1}) - f((\theta^j_{T+1}))'|$$

$$\leq L_U N K.$$

This means that the gains $|U^i(f(\theta^i_{T+1}), \ldots, f(\theta^N_{T+1})) - U^i(f((\theta^i_{T+1})'), \ldots, f((\theta^N_{T+1})'))|$ in unpenalized reward player $i$ can achieve by using a given $\alpha^i_t$ at time $t$ instead of $\alpha^i_t = 0$ is multiplied by $L_U N$ compared to 28 and 34. Thus, for a given $C^U_t$, the bound on $\alpha^i_t$ for rational players is multiplied by $N L_U$ as well, as the quadratic formula solution for $\alpha^i_t$ is linear in the linear term. Correspondingly, we need to set $C^U_t = L_U N C_t$ to achieve the same bounds on $\alpha^i_t$ as in E.2 or E.1.

Now, for $\alpha^i_t \leq \epsilon$, the bound on the expected absolute change in $U^i(f(\theta^i_{T+1}), \ldots, f(\theta^N_{T+1}))$ is $L_U N$ times higher than for $f(\theta^j_{T+1})$ using the bounds above. Because these bounds are linear in $\epsilon$ for E.2, we can achieve a bound of $\delta$ for $U^i(f(\theta^i_{T+1}), \ldots, f(\theta^N_{T+1}))$ by ensuring $\alpha^i_t \leq \frac{\epsilon}{NL_U}$ for $\epsilon$ achieving a bound of $\delta$ for $f(\theta^i_T)$. In total, we thus need to multiply the corresponding $C_t$ by $NL_U$ twice to achieve a given bound on the gains in unpenalized reward from cheating: Once because we need a smaller bound on $\alpha^i_t$ to achieve the same bound on the unpenalized reward, and once to ensure that rational players are incentivized to use that smaller bound.

$\square$

**Discussion on the projection assumptions** We note that the assumption $P(\exists t \leq T : \Pi_W(\theta^s_t - \gamma_t \bar{m}_t) \neq \theta^s_t - \gamma_t \bar{m}_t) \in O(\frac{1}{NT})$ in particular holds if $W$ is chosen such that $\|w - \theta^s_0\| \in \Omega(T)$ for all $w$ in the boundary of $W$ while $\|\bar{m}_t\| \in O(\|\theta^s_t - \theta^s_0\|)$ with probability one for all $t \leq T$. In that case, the linearly decaying learning rate ensures that $(\|\theta^s_t - \theta^s_0\|)$ stays in $O(T)$ and thus in $W$ (for appropriately chosen constants) with probability one.

Similarly, Lemma 5 in Rakhlin et al. (2012) states that for a probability one bound on the gradient norm $\|\bar{m}_t\| \leq G, \forall t$, we have that $\|\theta^s_t - \theta^\star\| < \frac{2G}{m}$ with probability one, such that the iterates stay in $W$ without the need for any projections, as long as $G$ grows slower in $T$ than $\inf_{w \in \delta(W)} \|w - \theta^\star\|$. In particular, if $f$ grows quadratically in $\theta - \theta^\star$ and noise is proportional to the gradient norm, $G$ grows linearly in the distance between $\theta^\star$ and the boundary of $W$, such that the condition holds for the right proportionality constants.

