# OpenReview forum: "Incentivizing Honesty among Competitors in Collaborative Learning and Optimization"
_NeurIPS.cc/2023/Conference — NeurIPS 2023 poster_

### Official Review · Reviewer_hRji · 2023-07-04

**Soundness:** 3 good
**Presentation:** 3 good
**Contribution:** 3 good
**Rating:** 7
**Confidence:** 4

**Summary:**

The paper presents an incentive mechanism to encourage honest data reporting in the presence of spiteful behavior aiming to harm other participants.

**Strengths:**


The paper considers an interesting an novel setting.

It shows that incentive schemes can in principle induce cooperative behavior.

The incentive schemes show both budget-balance and individual rationality (ex ante).

**Weaknesses:**

While the reward scheme in the paper has truthful reporting as an equilibrium, it is well-known that peer-prediction schemes also admit uninformative equilibria; for example in this case all participants could report the same data without any penalty.
*** This was my most important worry and the authors have addressed this weakness in their rebuttal ****

The schemes requires that the participants observe IID data, which is usually not the case in federated/distributed learning.
*** This still remains to be improved ****

Only particular attack and defense strategies are considered.
*** The authors have convinced me that they have gone far enough at least for this paper. ***

There is no consideration of data privacy.
*** This remains future work. ***

**Questions:**


How can you scale the penalties without knowing the utilities of the model to the participants?

**Limitations:**

I think the fact that utilities have to be known needs to be stated more clearly.

---

> ### Author Rebuttal · Authors · 2023-08-09
>
> Thank you for the review. We are glad you find our setting novel and interesting. We aim to address your concerns in the following:
>
> **While the reward scheme in the paper has truthful reporting as an equilibrium, it is well-known that peer-prediction schemes also admit uninformative equilibria; for example in this case all participants could report the same data without any penalty.**
>
> Thank you for bringing this up. We agree that equilibria beyond the honest one are usually an issue in peer prediction. However, as discussed in the general response as well, we believe that these are not as problematic in our case. First, as in the general peer prediction setting, honesty offers a natural Schelling point and other equilibria are substantially harder to coordinate on. Second, as laid out in our general response, any non-honest equilibrium will have at least two players with $b^i \neq 0$, such that all players receive a larger MSE than at the honest equilibrium. Unlike in peer prediction, it is therefore unclear why players would coordinate on a non-honest equilibrium in our setting for the most natural parametrization, $\lambda_i>1$, where players prioritize more the quality of their own model than damaging others’ models.
>
> We will update our manuscript to better reflect this important point.
>
> **The schemes require that the participants observe IID data, which is usually not the case in federated/distributed learning.**
>
> We disagree that we cover IID data only.  The general formulation in Section 3 allows for dependence between the samples (line 107) and we explicitly model heterogeneity in the mean estimation case (line 169).
>
> While our SGD theorems currently assume IID data, we believe that they could be extended to the non-IID case in future work. In particular, the FeMNIST dataset contains heterogeneous data, and our experiments demonstrate the efficacy of our mechanism for SGD on that dataset.
>
> **Only particular attack and defense strategies are considered.**
>
> We would like to highlight that the attack strategies we consider in the mean estimation setting are very general: As discussed in line 200 of the paper and our general response, equation 3 essentially parameterizes arbitrary attacks. Correspondingly equation 4, which prevents unrealistic strategies like always sending the true mean $\mu$ despite only having access to samples, is the only restriction placed on the strategy space for mean estimation.
>
> While the strategy spaces are clearly more restricted in our SGD setting, we would like to point out that intertemporal dependencies make the analysis of that case highly nontrivial, even with our restrictions.
>
> We did not consider arbitrary defense strategies for two reasons: First, our mechanisms already fully incentivize honesty, such that there is no need for further defenses. Second, as defense strategies can be viewed as statistical inference procedures, analyzing arbitrary defense strategies would require a fundamental breakthrough in statistics: In particular, due to Stein’s paradox, the optimal defense strategy is currently not known for $d\geq 3$ dimensions, even in a single player setting with data distributions restricted to isotropic gaussians.
>
> **There is no consideration of data privacy.**
>
> We first note that the considered mechanisms all use gradient information only, so in this sense our approach is equally private/non-private as classic Federated Learning.
>
> Additionally, privacy concerns are reflected in the considered attack model: Indeed adding noise is a standard way of increasing the Differential Privacy of an algorithm and is often used in practice in Federated Learning, to increase users’ privacy. In that sense, our mechanism could be interpreted as a way to balance players’ privacy concerns with the degradation of the learnt model that is caused by actions taken to ensure privacy.
>
> Beyond that we indeed do not explicitly model privacy into the objectives, as it is an orthogonal aspect compared to competing incentives, which is the focus of this work.
>
> **How can you scale the penalties without knowing the utilities of the model to the participants?**
>
> It is generally impossible to use any penalty-based incentive scheme without some knowledge about the functional form of players’ utility functions, as rescaling the utility functions by a factor of $K$ would also increase the smallest penalty factor that achieves honesty by $K$.
>
> That said, a key advantage of our mechanism is that we only require limited information about players’ utilities ($\lambda_i$ in the Mean Estimation case and the smoothness, Lipschitz and convexity parameters in the SGD case) to correctly scale the penalty. In particular, we do not require any information about the true values of the unknown parameters $\mu$ or $\theta^*$.

---

> > ### Comment · Reviewer_hRji · 2023-08-10
> >
> > Thank you for your responses.
> >
> > Regarding multiple equilibria, I like the point that participants would have to coordinate and use side payments to compensate the losers. An even better result would be if you could show that the sum of the rewards, or at least the expectation of that sum, could not increase - then indeed any collusion would not be stable and it would greatly strengthen your paper.
> >
> > I agree that there has to be some restriction on the strategy spaces. If space permits, it would be useful to have some discussion on what strategies are allowed and what strategies are not considered.
> >
> > With regards to point 4., it would be good to explicitly point out in the paper what knowledge of participant utilities is required, as this is an important limitation of the work.

---

> > > ### Author Response · Authors · 2023-08-10
> > >
> > > Thank you for your timely comment! We are glad that our points were well-received.
> > >
> > > Thank you for the detailed suggestion regarding the result on multiple equilibria. What you suggested indeed holds (in expectation) as long as all players are more concerned about their own model’s quality than about others’ ($\lambda_i>1$).
> > >
> > > To demonstrate why, please note that the (unpenalized) expected reward for player $i$ equals $\frac{\sum_{j \neq i}||\theta^j - \mu||^2}{N-1} - \lambda_i ||\theta^i - \mu||^2$, such that the sum of all players’ rewards equals $\sum_j (1-\lambda_j) ||\theta^j - \mu||^2$. This is monotonously decreasing in all players’ MSE’s $||\theta^i - \mu||^2$ and, as discussed in our initial response, these MSEs will always be larger for all players at any non-honest equilibrium. Therefore, the sum of expected (unpenalized) rewards of all players is smaller at any non-honest equilibrium than at the honest one. Meanwhile, as discussed in the paper, the penalties paid by all players add up to zero in expectation. Therefore, the result also holds for the sum of penalized rewards.
> > >
> > > We will add this result to our paper and highlight its importance for equilibrium selection. We will also add an additional discussion on the strategy spaces, and aim to better highlight the knowledge required about players’ utility functions as a limitation.
> > >
> > > We hope these improvements address the reviewer’s original concerns and are happy to answer any further questions.

---

> > > ### Author Response · Authors · 2023-08-16
> > >
> > > Thank you again for your response. Since the discussion period is progressing, we wanted to check in whether the result from our last response successfully addresses your concern about multiple equilibria and affects your overall paper evaluation? We would be happy to provide more clarifications if needed.

---

### Official Review · Reviewer_hQqh · 2023-07-07

**Soundness:** 2 fair
**Presentation:** 2 fair
**Contribution:** 2 fair
**Rating:** 4
**Confidence:** 3

**Summary:**

The paper considers a federated learning setting with strategic data resources. The authors assume that the entities taking part in the learning process are selfish players incentivized to get the best model but benefit if their competitors receive inaccurate models. This selfish behavior pushes players to lie to the central learning mechanism in their reports.

The authors consider two cases: Mean estimation and a multi-round SGD on strongly convex objectives. They model the players' strategies as multiplicative/additive factors that could be added to the players' actual local computations. The authors show that even in the straightforward case of mean estimation, a PNE does not exist. They offer two remedies: Monetary payments (via peer-prediction techniques) and punishments (noisy model updates by the central mechanism). Then, they show that a PNE exists and characterize the form of payments/punishments required.

Finally, they conduct an experimental analysis demonstrating that their remedies positively affect the learning procedure.


**Strengths:**

1.	The paper deals with a practical issue that is somewhat under-explored.
2.	Despite the abundance of notations, the authors have done an excellent job in making the paper read smoothly.


**Weaknesses:**

1.	The paper adopts a game theoretic approach, but many modeling assumptions seem cumbersome and unjustified (see questions below).
2.	It is hard to assess this paper's technical contribution. Particularly, the novelty of the peer prediction-based mechanisms are well-studied ideas. The authors did not explain whether this paper adopts these ideas in a plug-and-play manner or presents new non-trivial derivations. The "our contribution" part addresses the paper's content but not its marginal contribution to the line of research, making the technical contribution hard to assess.


**Questions:**

1.	Strategy spaces: The assumption that players report their updates along with $\alpha^i \xi^i$, where $\alpha^i$ is player $ i$'s strategy and $\xi^i$ is a random variable seems completely arbitrary and unjustified. Typically, one makes assumptions about what players aim to do (e.g., maximize their payoff) and what they can do (e.g., bounded computation or memory, acting myopically, etc.). Explicit assumptions about players' strategies without proper grounding and reasoning are cumbersome and inconvenient. Could the authors justify what real-world scenarios the strategy spaces in Eq. (3) models? Could the authors present assumptions about player rationality\behavior that would recover their modeling?
2.	The assumption about $b^i$ in Eq. (4): What is the justification for this? Indeed, it facilitates the analysis but seems entirely out of context.
3.	Multi-round: The authors assume that in the multi-round case, players pick their strategies only once initially. What is the rationale for this modeling? Allowing players to change their strategies throughout the execution will be harder to analyze, but this compromise does not make much sense.
4.	Corollary 2: The authors show that a pure Nash equilibrium does not exist, concluding that "without modifications to the protocol, no player can benefit from collaborative learning." While this might be true, I do not see how the inexistence of Nash equilibrium implies collaboration is useless. To reach this conclusion, the authors must show that players are better off (i.e., their estimates are more accurate) without the protocol. Where is this analysis located in the paper? Further, I suspect that a mixed Nash equilibrium does exist, so arguing about whether players can benefit from collaboration should at least consider their payoff under some form of a solution concept (be it mixed equilibrium, sink equilibrium, or otherwise). My question: Could the authors describe why the inexistence of PNE suggests that collaboration is useless?
5.	The methods the authors adopt, e.g., scoring rules, seem to treat the most general case where players can report whatever they want, beyond limiting the structure of their message (Eq. (3) for the mean estimation case). This is at least true in the mean estimation case. The same thing applies to the payment case. What does this paper benefit from making the limiting (and, as I argued before, the highly unjustified) assumption of the structured strategy spaces?
6.	What is the technical modification of peer-prediction\noise communication required for this paper? How novel is the derivation needed for this paper, and how does it differ from previous papers? Answers to this question could facilitate the assessment of this paper's contribution.


Minor:

•	Why are the super scripts m and w needed in lines 198 and 211 (they also appear later in the paper)?

•	173: clients->players

---

> ### Author Rebuttal · Authors · 2023-08-09
>
> Thank you for the constructive and thoughtful review. We are glad you found our paper smooth to read. We aim to answer your questions in the following:
>
> **I suspect that a mixed Nash equilibrium does exist. Also, what does the existence Nash equilibria have to do with benefits from collaboration?**
>
> Thank you for pointing this out. Our result holds for mixed Nash equilibria with one important caveat: As explained in line 223, the players’ expected reward is monotonous in $\mathbb{E}(\alpha^j(x^j)^2)$ (with the expectation taken over both the sample $x^j$ and a player’s random strategy choice for mixed strategies) as long as any other player uses a fixed $\beta^i<1$. As $\beta^i$ and $\alpha^j$ are independent, this extends to random choices of $\beta^i$ with  $P(\beta^i<1)>0$. But the optimal $\beta^i$ never equals one for finite attacks, so no strategy profile with finite rewards can be stable. However, your comment made us carefully revisit the corollary, and we noticed that we do not explicitly rule out infinite values for $\mathbb{E}(\alpha^j(x^j)^2)$ or $\mathbb{E}(||b^j(x^j)||^2)$. For these infinite strategies, our monotonicity-based argument fails.
>
> The fact that the only equilibria are “at infinity” also explains why collaboration does not help: If one player uses an “infinite” strategy, all other players’ optimal $\beta^i$ equals one, such that they completely ignore others’ data, like in the non-collaborative case.
>
> We apologize for the confusion and will update our paper to better highlight our previously tacit assumption on the finiteness of strategies.
>
> **What do the strategy spaces in Eq. (3) model? What is the justification for assumption (4)?**
>
> As we discuss in more detail in line 200 of the paper and the general response, Equation 3 without the restrictions posed by Equation 4 essentially expresses the most general attack strategy possible. This is because any possible modification to the mean can be decomposed into a deterministic shift and adding zero-mean noise.
>
> Equation 4 prevents “non-general” strategies whose success depends on the precise value of the parameter $\mu$ to be estimated. In particular, without it strategies that cannot realistically be implemented without knowing the true parameter $\mu$, like always sending the true mean $\mu$, would be admissible. Behaviourally, the assumption can be interpreted as “players do not base their strategies on guesses about the real parameter $\mu$ that go beyond information obtained from their sample $x^i$."
>
> While we do agree that Equation 4 also prevents some strategies without this issue, we currently do not know of a weaker assumption that excludes unrealistic strategies while retaining the same mathematical simplicity.
>
> We will update our manuscript to make this point more clear.
>
> **Why is the strategy space in the multi-round setting restricted to predetermined strategies?**
>
> Our attack structure for SGD is inspired by data-hiding (which increases the variance of the gradient estimates) and Differential Privacy defenses (which add zero-mean noise to the gradient in order to increase the privacy of  local data). We opted for non-adaptive strategies, as adaptive strategies will lead to complex dependencies between consecutive SGD steps. In particular, this would make our already quite involved analysis of the SGD case even more difficult from a purely optimization perspective, since arbitrary dependencies between rounds are highly non-standard in usual gradient-based optimization proofs.
>
> **Why are assumptions made on the strategy spaces?**
>
> We believe that rigorous analysis of a (relatively broad class of) special cases is essential for progress whenever analyzing the most general case is not tractable. In particular, if we considered more general defenses in the mean estimation setting, even solving a single player version of our game would essentially amount to the classic statistical problem of finding an admissible estimator. Unfortunately, we are unaware of such results for $d \geq 3$ dimensions in the literature, due to Stein’s paradox. Please also refer to our general response for further discussion on these matters.
>
> **What is the technical modification of peer-prediction required for this paper? How novel is the derivation and how does it differ from previous papers?**
>
> To the best of our knowledge, our use of noise rather than explicit payments to implement a peer prediction mechanism (5.2) is completely novel. We achieve this using a new reduction of the noise-based case to our payment-based result.
>
> Similarly, we believe to be the first to use a peer prediction mechanism in a multi-round optimization scheme like SGD. Unlike in prior work, it is not possible to base the mechanism on the final output $\theta$, as this is a function of all players’ strategies over multiple time steps. This makes it impossible to apply standard arguments that relate the quality of a player’s estimate and how well their estimate predicts other players’ estimates.
>
> Instead, we employ a novel recursive bound for the squared norm of differences in SGD-iterates between a clean trajectory and a trajectory with time-varying gradient noise, that has to take into account ripple effects of noise added during early time steps. This allows us to bound the effect a player that uses a particular strategy can have on the final SGD iterate and thus the loss.
>
> Our first result (5.1) is closest to the existing literature, but there are still important differences: a) In peer prediction, the goal is to incentivize effort to produce good estimates, while our setting focuses on penalizing malicious manipulations. b) As far as we know, our redistribution scheme used to achieve zero expected payment for honest players has not previously been analyzed c) we consider estimates for n-dimensional vectors and arbitrary distributions.
>
> **What do superscripts m and w mean?**
>
> Thank you for pointing this out. We have removed these superscripts.

---

> > ### Comment · Reviewer_hQqh · 2023-08-18
> >
> > I thank the authors for their reply. I will take it into account during further discussion.

---

> > > ### Author Response · Authors · 2023-08-19
> > >
> > > Dear reviewer, thank for your response! Please do let us know about any remaining questions or concerns, so that we can address them until the end of the author-reviewer discussion period.

---

### Official Review · Reviewer_Mvyk · 2023-07-08

**Soundness:** 3 good
**Presentation:** 3 good
**Contribution:** 3 good
**Rating:** 6
**Confidence:** 4

**Summary:**

The paper studies a centralized collaborative learning problem. Authors provide theoretical guarantees for an attack method and a defense method. Further, the paper proposes two mechanisms to incentivize honesty: a method that uses an explicit side payment method and requires transferable utility, a centralized punishment mechanism where a central server adds noise to the estimates it sends to players that have sent suspicious updates. Simulation results are provided supporting the claims.

**Strengths:**

-- The paper is well written and easy to follow.

-- Authors provide a description of related work and background. The problem formulation considered in the paper is well positioned in the relevant literature.

-- Authors provide several theoretical results making novel technical contributions. Authors provide discussions around the implications of the theoretical results.

-- The paper provides simulation results supporting the theoretical claims.





**Weaknesses:**

-- Problem formulation is well positioned in the relevant literature. However, authors do not provide a discussion on how their results and methods compare to existing literature.

-- Authors provide numerical simulation results supporting their analysis. However, authors fail to compare their method with existing methods in the simulations.

**Questions:**

-- I encourage authors provide a comparison with existing methods.

-- Is it possible to add a proof sketch in the main paper highlighting the technical challenges addressed in the analysis?

-- Can these results be extended to the decentralized setting?

**Limitations:**

Authors do not provide a discussion on limitations of their method. Authors include a discussion on societal impacts in the Appendix.

---

> ### Author Rebuttal · Authors · 2023-08-09
>
> Thank you for your positive and thoughtful review. We are glad you found our paper easy to follow and our contributions novel. We aim to answer your questions and address your concerns in the following:
>
> **Can these results be extended to the decentralized setting?**
>
> In a decentralized setting where players publicly communicate their estimates to all other players, our payment-based mechanism could be implemented without a server, as the payment is a simple function of all players’ communicated estimates, so everyone’s payment can be computed publicly. However, this does not work for our noise-based mechanism, as players can just personally aggregate the public clean estimates when there is no payment-based penalty.
>
> **There is no discussion on how the methods/results compare to the existing literature. Is it possible to add a proof sketch in the main paper highlighting the technical challenges addressed in the analysis?**
>
> We appreciate this suggestion and plan to add further details on the key technical challenges in future versions of the paper with an extended page limit.
>
> As a brief summary, the first key technical insight is that ideas from peer prediction can be applied to our novel setting of competition in collaborative learning. With this insight, Theorem 5.1 can be derived with versions of techniques used in previous work on peer prediction, generalized to treat the d-dimensional case without strong distributional assumptions.
>
> Next, as far as we know, our noise-based mechanism is entirely novel and the analysis required a nontrivial reduction of that mechanism to the payment-based case. A key challenge in the analysis is that the magnitude of the added noise is correlated with a players’ sampling error, such that standard independence-based decompositions of the squared loss do not work.
>
> Lastly, our treatment of peer prediction for SGD is also entirely novel to the best of our knowledge. In this setting, it is not possible to base the mechanism on the final output as is usually done in peer prediction. Instead, our analysis bounds the effect a player can have on the final loss by manipulating an update at time t, which can be highly complicated due to ripple effects, using a novel recursive bound for the squared norm of differences in SGD-iterates between a clean trajectory and a trajectory with time-varying gradient noise. This allows us to connect the expected deviations between a player’s gradient and other gradients to the overall “damage” they cause, and bound that damage by penalizing gradient deviations.
>
> **There is no comparison to existing methods in the experiments.**
>
> Thank you for the suggestion. We would like to include comparisons to baseline methods from existing work, but since the formal setting we consider is entirely novel, there are no established methods to fairly compare to. While there are existing methods that aim to make collaborative learning robust to updates from dishonest players, our methods are orthogonal as they instead aim to prevent dishonesty in the first place.
>
> In order to demonstrate how our work complements existing robust collaborative learning methods, we provide an additional experiment using one such method: robust stochastic gradient descent using the median rather than the mean of players’ updates (Yin et al., ICML 2018). Please refer to the PDF response for the results. We present two plots, to demonstrate how the median compares to the standard mean-based aggregation in the presence of noise (Figure 4); and how our mechanism performs in combination with the median-based aggregation (Figure 3).  As can be seen from the first plot, while this median method increases robustness to noise, it still performs worse the more noise is added, such that incentivizing honesty remains important. From the second plot we see that, similarly to the experiment in the main text, the players are incentivized to not send noise as long as the constant C, that controls the strength of our penalties, is sufficiently large. Therefore, our mechanisms are effective in preventing attacks that would otherwise hurt the performance of the players models, even in cases when a robust FL defense is used.
>
> Yin, Dong, et al. "Byzantine-robust distributed learning: Towards optimal statistical rates". In: International Conference on Machine Learning (ICML), 2018.

---

### Official Review · Reviewer_seCF · 2023-07-10

**Soundness:** 3 good
**Presentation:** 3 good
**Contribution:** 3 good
**Rating:** 7
**Confidence:** 3

**Summary:**

The authors investigate the issue of  manipulation (in the form of falsifying data or model updates) among agents who mutually contribute to a shared model. Incentives for such behaviors arise when agents possess differing objectives with respect to the shared model. The authors first demonstrate that without external intervention, these incentives are essentially unavoidable. However, the authors propose two methods for inducing incentive compatibility in such settings; namely payments when utility is transferable, and noisy server messages  when utility is non-transferable. The authors derive these mechanisms and provide additional theoretical results for two settings of collaborative online learning, single shot mean estimation and multi shot shared gradient updates. Lastly the authors provide experimental results on the FeMNIST data set demonstrating that their mechanisms dissuade strategic behavior.


**Strengths:**

1. Distributed learning is a rapidly growing area and there is a real danger that strategic agents could disrupt the efficacy of these systems if their incentives are not properly accounted for. As such, the authors’ work is well motivated and helps to fill an important piece which is currently missing from the literature.

2. Manipulations in these types of settings are often framed as being adversarial. The authors model agents as being strategic rather than purely adversarial. As we have seen in areas like supervised learning the differences between adversarial and strategic agents can be highly consequential in terms of designing robust systems; considering both types of behavior is imperative (the former is already covered in prior work).

3. The need for such mechanisms is well motivated both from a narrative perspective and from a theoretical perspective (Corollary 4.2).

- Considering the case of non-transferable utility increases the applicability of the authors results, and the case of transferable utility provides the system more options to incentivize collaboration when payments are feasible.

4. The authors’ results are constructive, rather than simply existential. For example, rather than saying that there exists a $C$ and $\lambda$ such that players participate honestly, the authors provide specific ranges of these variables for which honest behavior is an equilibrium. This makes it easier for others to implement their methods in real-world scenarios.

5. While some strong assumptions are made for the theoretical results, such as convexity, the inclusion of the experimental results helps demonstrate that the authors’ approach is effective even when such assumptions do not hold.

6. The paper is well written and the authors take the time to outline the intuition and implications of their results.


**Weaknesses:**

1.  The mechanisms proposed only induce truthfulness as a Nash Eq, implying that other non-truthful equilibria exist. I understand that these types of results are standard throughout the literature, but when deploying these mechanisms in practice it is important to note that we have no guarantee which equilibrium agents will end up in. This is a far weaker result than truthfulness being a dominant strategy.

2. Similar to the last point, the mechanisms do not appear to be collusion proof. In particular one player could pretend to represent multiple clients (i.e., sending multiple updates each round). For example, in the case of single-shot mean estimation, if such an agent monopolizes a sufficiently large position of the data being submitted, they could force the other agents into submitting any desired value for large enough $C$ (since the deviation penalty will outweigh the other parts of their utility). If agents are willing to misreport data, they are probably also willing to collude. Not accounting for this possibility limits the scope of the work. With that said, the authors appear to be the first work to study robustness to strategic behavior in distributed systems, so perhaps asking for additional results on collusion is too much. However, this should be more clearly stated as a limitation of the work.

3. The experiments are somewhat limited. The primary contribution of the paper is theoretical and the point of these experiments is to show that the payment scheme works even when convexity does not hold, but this observation on a single dataset, for a single model, is a bit unconvincing. In particular, I would expect that the average reward received when increasing $\alpha_A$ would decrease more rapidly for larger $C$, however, it is not clear to me that the small amount of fines paid by honest players would hold across different scenarios.




### Comments and minor issues: no impact on score and need not be addressed in the author response.

1. Line 259: should this say “... at the honest equilibrium [when] ….”? In the supplement this is stated as an [and] rather than a [when].

2. Corollary 4.2 should probably be a Theorem. This is actually quite an interesting result and somewhat non-trivial based on the proofs. Although Theorem 4.1 is doing most of the heavy lifting here the corollary is actually the main results, while the Theorem feels more like a helping lemma.

3. Links to theorems and references are broken in the main body. Looks like this is the result of compiling the document with the supplement and then using a PDF editor to trim the supplement pages.

4. Figure 6 in the supplement takes up its own page.

5. The naming convention of Theorems is not consistent with the main body and makes it difficult to find a specific theorem within the supplement, unless using the reference provided in the main body.


**Questions:**

See above

**Limitations:**

See above

---

> ### Author Rebuttal · Authors · 2023-08-09
>
> We thank you for your positive and thoughtful review. We are glad you liked our paper and plan to incorporate your feedback in the next revision. In the following, we aim to address your concerns:
>
> **The mechanisms proposed only induce truthfulness as a Nash Eq, implying that other non-truthful equilibria exist.**
>
> Thank you for pointing out this important issue. We agree that honesty as a dominant strategy would be a more desirable result. As we discuss in more detail in the general response, we do not think that the other existent Nash equilibria would pose a large problem in practice. In particular, they are simultaneously more difficult to coordinate on and make all players’ receive larger MSEs than at the honest equilibrium, such that it is unlikely that players will choose to coordinate on these equilibria when they care more about their own models than others’ ($\lambda_i>1$).
>
> We will update our manuscript to better reflect this important point.
>
> **The mechanisms do not appear to be collusion proof**
>
> Thank you for bringing this up. Collusion in our framework is an important topic that would be interesting to analyze in future work, and we will highlight this limitation in an updated version of the paper.
>
> We do expect our mechanisms to be collusion-proof against small coalitions because an actor that pretends to be multiple actors will also have their penalty multiplied accordingly. However, we do agree that there is a problem once the colluding coalition significantly affects the mean estimate, as the shifted mean would reduce the penalty paid by each member of the coalition.
>
> **There are only experiments for a single dataset and model.**
>
> We reran our experiment on a subset of the LEAF Twitter sentiment analysis benchmark, training a 2-layer classifier on top of frozen BERT embeddings. These results are plotted in the response PDF in Figures 1 and 2. We observe similar trends as in the experiment in the main paper, in particular if $C$ (the scaling on the monetary penalties) is sufficiently large, players are strongly disincentivized from adding large noise.
>
> We found our model in the Twitter benchmark to be more sensitive to gradient noise than the CNN in our FeMNIST experiments, with a noise level of $\alpha_A=5$ degrading the loss by more than twice as much ($0.084$ vs $0.034$) as noise level $\alpha_A=9$ did for the CNN. Correspondingly, the penalties needed to achieve honesty were roughly 10 times larger than on the FeMNIST dataset, which leads to a similar increase of the 98th percentile of payments at the honest equilibrium (from $0.0031$ to $0.0243$) for the largest considered penalty $C = 0.002$. We would like to note that this outlier payment is still only a third of the damage caused to the loss at $\alpha_A=5$, and that a 4 times smaller penalty of $C = 0.0005$ still appears to be sufficient to incentivize honesty.
>
> Lastly, we would like to thank you for the additional feedback. We will incorporate it in the updated version of the paper.

---

> > ### Comment · Reviewer_seCF · 2023-08-14
> > **Response to authors**
> >
> > Thank you for detailed response.
> >
> > **Truthful EQ** I find the authors' point regarding the additional coordination required for non-truthful EQs to be convincing. Adding this as a remark would boost the usefulness of the results regarding truthful EQs.
> >
> > **Collusion** I agree that the mechanism is likely collusion proof when the level of collusion is small (or at least some similar mechanism). While collusion is not the main focus of this paper, providing a result regarding collusion would increase the paper's strength. However, even without such a result, the paper is a clear accept in my opinion.
> >
> > **Additional Experiments** Thank you for running these. The contrast in required payment between different datasets is quite interesting. There is a point to be made that perhaps in practice finding the the "correct" fine may be tricky as small fines may lead to undesirable strategic behavior, while large fines may disincentivize agents from participating in the mechanism.
> >
> >
> > **Restricted model** Other reviewers raised issues of unjustified or restrictive assumptions made by the authors. While I agree with some of these criticisms, I do not believe the authors' assumptions to be too restrictive, and find their response to reviewer @hQqh to be satisfactory.
> >
> >
> > After reading the other reviews and the authors' rebuttal I stand by original recommendation of Accept. This paper is very interesting and the model is highly relevant to the growing field of distributed learning.The authors' paper sets up a solid foundation through which future work can further analyze this problem (especially given its practicality).

---

> > > ### Author Response · Authors · 2023-08-15
> > > **Thank you for your response!**
> > >
> > > Thank you for your timely and detailed response! We appreciate your constructive and positive feedback regarding our paper.
> > >
> > > We are also happy that you find our modelling of the strategy spaces appropriate.
> > >
> > > We will incorporate your feedback into the next version of the manuscript. In particular, we will add discussions on the non-truthful EQs, how to find the correct penalty scaling and regarding collusion. We will also include and discuss the additional experiments.

---

### Author Rebuttal · Authors · 2023-08-09

We thank the reviewers for their valuable and constructive feedback. We are glad that the reviewers find our setting novel and interesting ($\color{blue}hRji$), well motivated ($\color{lime}seCF$), our technical contributions novel ($\color{red}Mvyk$) and our text smooth to read ($\color{cyan}hQqh$). We now address the most common questions brought up by the reviewers, and look forward to the reviewers’ replies.

**The mechanisms induce honesty as a Nash equilibrium, but there are other equilibria ($\color{lime}seCF$, $\color{blue}hRji$)**

While this is true, we would like to put it in context.

In the peer prediction setting, dishonest equilibria are usually preferable to all players (but the server), but this is not true for most natural instantiations of our setting. In the mean estimation setting, by the proof of theorem 5.1, $\mathbb{E}[b^i(x^i)]$ has to be nonzero for at least two players at non-honest equilibria such that all players receive at least one manipulated update and a worse model than at the fully honest equilibrium. Correspondingly, the sum of all players’ MSEs increases, such that at least some players would prefer the honest equilibrium as long as $\lambda_i>1$. This makes it unlikely for players to take on the difficult coordination task of playing a non-honest equilibrium. In the SGD setting, Theorem 6.1 guarantees that at any equilibrium the losses of the players are close to optimal.

We thank the reviewers for pointing this out. We will add a more detailed discussion of this in the final version.

**The considered strategy spaces are restricted ($\color{cyan}hQqh$, $\color{blue}hRji$)**

As noted in line 200, the attack strategies we consider for mean estimation are quite general: Equation (3) essentially parameterizes the most general attack space in a more interpretable way: Any random variable $m(x)$ that represents a message can be written as $\bar{x} + b(x)+\alpha(x)\xi$ where $b(x) =\mathbb{E}[m(x)|x]-\bar{x}$ and $\mathbb{E}[\xi|x]=0$.

Equation (4) effectively prevents $b^i$ from encoding knowledge about $\mu$. This restriction is needed to exclude “non-general” strategies, whose success depends on the true value of the parameter mu that is to be estimated, such as always sending a constant value $m^i$, so that $b(x)=m^i-\bar{x}^i$.

Our strategy spaces are more restricted in the complex SGD setting, where, however, the attack space has several natural interpretations, such as adding a noise-based differential privacy defense or hiding samples from the empirical estimates. That said, we would like to point out that the use of restricted strategy spaces/classes of estimators is common in both game theory and statistics. In particular, the optimal solution to even a single player version of our Mean Estimation game for an isotropic gaussian distribution is unknown for fully general strategy spaces in $d\geq 3$ dimensions, due to Stein’s paradox.

**Do meaningful mixed Nash equilibrium exist without our mechanisms ($\color{cyan}hQqh$)?**

Indeed, our result already includes mixed equilibria. The confusion might in part have been caused by an imprecision in the corollary statement: As our monotonicity-based argument for the nonexistence of equilibria only works as long as rewards are finite, the corollary should state: “... does not have any (mixed) Nash equilibrium for which $\mathbb{E}(\alpha^j(x^j)^2)$ and $\mathbb{E}(||b^j(x^j)||^2)$  are finite for all players.”

We apologize for the confusion and will update the paper to clarify this point by updating the corollary statement to the version presented above.

For more details, please consider the individual response to reviewer $\color{cyan}hQqh$.

**It is not clear from the manuscript whether our results are a straightforward application of existing peer prediction results or require novel technical methods ($\color{red}Mvyk$,$\color{cyan}hQqh$)**

We thank the reviewers for pointing this out and will aim to highlight the key technical challenges better in future versions of our manuscript. We respond to each reviewer individually and provide a summary below.

To our awareness, our work is the first to explicitly model competitive incentives between clients in collaborative learning. Our first key technical insight is that ideas from peer prediction, which instead focuses on conflicting interests between the server and clients, can be applied to this setting.

To the best of our knowledge, both the noise-based mechanism (Theorem 5.2) and our application of peer prediction to SGD (Theorem 6.1) are entirely novel. The analysis of the former required a nontrivial reduction to the payment-based case, while the analysis of the latter is based on a novel recursion for the squared norm of differences in SGD-iterates between a clean trajectory and a trajectory with time-varying gradient noise.

**Additional experiments ($\color{lime} seCF$,$\color{red} Mvyk$)**

We added further experiments, one using a different dataset and model; and one that compares our schemes to an existing method for robust FL. Please refer to the response PDF and the individual responses to reviewers $\color{lime} seCF$ and $\color{red} Mvyk$ for further details.

**The schemes require that the participants observe IID data ($\color{blue}hRji$)**

We disagree that we only cover IID data – Sections 3,4,5 and 7 explicitly analyze various non-IID settings. We will highlight this better in the next version of the manuscript. We refer to the individual response to Reviewer $\color{blue}hRji$ for more details.

---

### Decision · Program_Chairs · 2023-09-21

**Decision:**

Accept (poster)

**Comment:**

Three of the four reviewers are positive, the fourth reviewer feels that the assumptions and the technical novelty are not sufficiently unjustified. My own opinion (after reading the paper, the reviews and the rebuttals) is that it is not a clear accept, but the positives overweigh the negatives. I strongly encourage the authors to revise the paper and address the reviewers' concerns to the extent possible.